# ToEdit: How to Synthesize Text Data to Avoid Model Collapse?

## Abstract

We explore model collapse caused by synthetic data, where AI models trained on such data experience a gradual decline in performance. Our initial analysis examines language model pretraining on mixed human and synthetic data, highlighting performance degradation. Further statistical analysis reveals distributional shifts and an over-concentration of n-gram features caused by synthetic data. Inspired by these insights, we propose token-level editing on human data, to obtain semi-synthetic data instead of fully using model outputs. As a proof of concept, we theoretically demonstrate that token-level editing can prevent model collapse, as the test error is constrained by a finite upper bound. We conducted extensive experiments on pretraining, continual pretraining, and supervised fine-tuning of language models. The results validate our theoretical proof that token-level editing improves data quality and enhances model performance.

## 1 Introduction

As generative artificial intelligence (AI) such as ChatGPT (Achiam et al., 2023) and Stable Diffusion (Rombach et al., 2021) are now widely used in our daily lives, training next-generation language models within an ecosystem of synthetic and human data will be inevitable. How will synthetic data influence AI training? Recent studies have given rise to two opposing viewpoints: some argue that synthetic data is the future of AI training, while others claim it leads to model collapse. From a practical perspective, numerous synthetic datasets have been proved to boost the capabilities of language models, like mathematics (Trinh et al., 2024; LI et al., 2024), biomedicine (Zhang et al., 2024), alignment abilities (Ouyang et al., 2022; Cui et al., 2023) and so on. From a theoretical perspective, training models iteratively on their own synthetic outputs results in the continuous accumulation of errors, manifesting as a degenerative process for model learning (Shumailov et al., 2024), i.e., model collapse. Furthermore, model collapse leads to a breakdown of scaling laws, ultimately rendering the incremental computational effort ineffective (Dohmatob et al., 2024b).

There are two key questions that require further investigation: (1) Beyond the highly filtered synthetic data in post-training, what is the impact of general synthetic data on language model training, and how does it differ from human data? (2) How can we prevent model collapse when synthesizing data, thereby producing higher-quality data?

In this paper, we answer the first question through data mixture pre-training with synthetic and human data, which shows the non-iterative model collapse. Subsequent statistical analysis on distribution and features indicates coverage collapse and over-concentrates n-gram features of synthetic data. Based on the above insights, we answer the second question by proposing a token-level editing (ToEdit), which can avoid model collapse in theory and produce high-quality data across experiments, including pre-training, continual pre-training, and supervised fine-tuning in practical.

Remarkable recent works provide a solid foundation for our work. Shumailov et al. (2024); Dohmatob et al. (2024a) identify the model collapse phenomenon and provide the first theoretical framework based on linear regression. Gerstgrasser et al. (2024) demonstrated that if synthetic data is accumulated while retaining the initial real data, the test error will be bounded, thus breaking model collapse. Building on the above frameworks, we prove that our token-level editing can also avoid model collapse. Additionally, Dohmatob et al. (2024b) indicated missing long tails of synthetic data lead to scaling law cutoff, which motivated us to explore data mixture pretraining and statistical analysis.

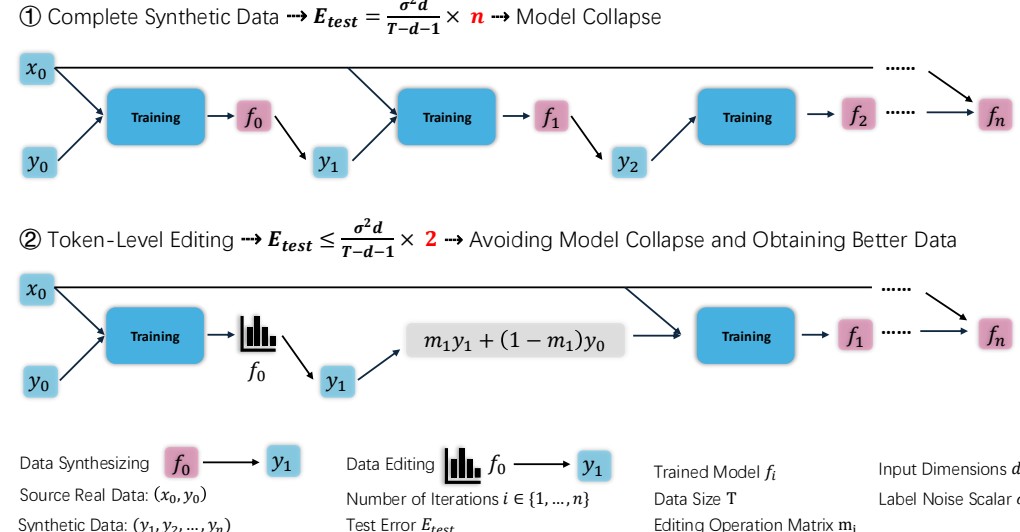

Figure 1: Model Collapse of Synthetic Data. ① the model continuously trains on its previously generated data, leading to a gradual decline in model performance, i.e., model collapse. ② We use the trained model for token-level editing rather than purely synthesizing data. In this case, we can preserve the distribution coverage, thereby avoiding model collapse and obtaining better data compared to the initial data. Specifically, ① starting from real data $(x_o, y_o)$, the test error $E_{test}$ increases as $f_0$ is iteratively trained on synthetic data $(y_1, y_2, \ldots, y_n)$. Our method, ② ToEdit, utilizes $f_0$ and an operation matrix $m_i$ to edit the data, achieving a fixed upper bound. Theoretical details are provided in § 3

**Contributions.** We summarize the key contributions of this work as follows:

- We discover non-iterative model collapse through pre-training GPT-2 on a mixture of synthetic and human data (§ 2.1). Specifically, we find that directly mixing general synthetic data, without iterative training, leads to performance degradation.
- We conduct distributional statistical analysis to uncover that synthetic data cause distribution coverage collapse and n-gram features over-concentrate. Further data selection struggled to correct the distribution(§ 2.2)
- We propose token-level editing, which can be proved to avoid model collapse (§ 3) and produce high-quality data across scenarios of pre-training, continual pre-training and supervised fine-tuning of language models (§ 4).

## 2 NON-ITERATIVE MODEL COLLAPSE

In this section, we investigate non-iterative synthetic data mixture training and explore the reasons behind non-iterative model collapse. Non-iterative refers to training a model directly on data synthesized by other models. Compared to previous iterative model collapse, non-iterative settings more closely reflect real-world model training scenarios.

### 2.1 HUMAN AND SYNTHETIC DATA MIXTURE PRE-TRAINING

*Setup* We define the mixing ratio between human and synthetic data as $\alpha$, where $0 \leq \alpha \leq 1$. The total amount of training data $D_{\text{total}}$ is expressed as a combination of human data $D_{\text{human}}$ and synthetic data $D_{\text{synthetic}}$, represented by the formula:

$$D_{\text{total}} = \alpha D_{\text{human}} + (1 - \alpha)D_{\text{synthetic}} \tag{1}$$

We use Dolma (Soldaini et al., 2024) as source human data. We use Cosmopedia (Ben Allal et al., 2024) as the source synthetic data, which is distilled from

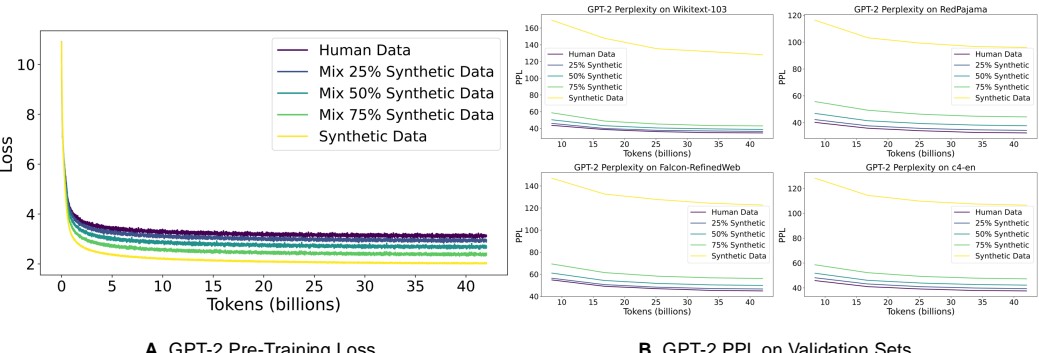

**A.** GPT-2 Pre-Training Loss        **B.** GPT-2 PPL on Validation Sets

Figure 2: Non-Iterative Model Collapse. Training language models from scratch on AI-synthesized data or a mixture of human and synthetic data leads to performance degradation. This degradation is positively correlated with the proportion of synthetic data used in training. **A.** We pretrain GPT-2 Small (124M) on human (Dolma (Soldaini et al., 2024)) and synthetic (Cosmopedia (Ben Allal et al., 2024)) data. As the proportion of synthetic data increases, the model's loss decreases. **B.** As the proportion of synthetic data increases, the PPL also rises. This trend remains consistent across different validation sets.

`Mixtral-8x7B-Instruct-v0.1` (Jiang et al., 2024). Using the data mixture of 50B tokens, we train two models from scratch, including GPT-2 (Radford et al., 2019) and OLMo (Groeneveld et al., 2024).

**Finding I: General synthetic data harm the language models pre-training.** Previous massive works have proved synthetic data can boost language models' capability, including instruction following (Wang et al., 2022a), reasoning (Zhu et al., 2023; Trinh et al., 2024), alignment (Cui et al., 2023), biomedicine (Zhang et al., 2024) and so on. However, as illustrated in Figure 2, the PPL of real-world validation sets is inversely proportional to the proportion of synthetic data. Compared with prior studies, we mix synthetic data in pre-training, not supervised fine-tuning and RLHF, which are downstream tasks. Before a language model reaches a certain level of learning, that is, when training from scratch, synthetic data is unlikely to help the model learn and may even hinder its learning. When synthetic data incorporates some human data into training data, the model collapse can be alleviated. Compared to previous works on iterative model collapse (Shumailov et al., 2024; Dohmatob et al., 2024a;b), the non-iterative damage caused by synthetic data is more concerning and relevant to the training of next-generation language models.

## 2.2 WHY DO SYNTHETIC DATA FAIL IN LANGUAGE MODEL PRE-TRAINING?

We conduct three statistical analyses: (1) sample-level distribution, (2) feature-based overlap, and (3) distribution-reference data selection. From the following experiments, we can summarize that compared with human data, synthetic data not only lacks long tails but also coverage collapse. It is hard to use human data as a reference to filter synthetic because the features in synthetic data are condensed heavily.

*Setup* We conducted statistical and feature-based analyses to explore why synthetic data fails in pre-training. (1) We leverage a prior distribution $P$ to estimate the human and synthetic data. We use Llama-3-8B (AI@Meta, 2024) and StableLM-Zephyr-3B (Bellagente et al., 2024). Different priors consistently yield the same results. (2) We analyze the n-gram features of human and synthetic data from a feature-based perspective, such as n-gram response values. (3) Based on the distribution of real data, we sample data from the synthetic dataset that closely matches the real data distribution in an attempt to filter the synthetic data.

**Finding II.i Synthetic data distribution not only misses long tails, but also causes coverage collapse.** Figure 3 and 9 illustrate that the PPL of synthetic data is confined to the lower 25% of the

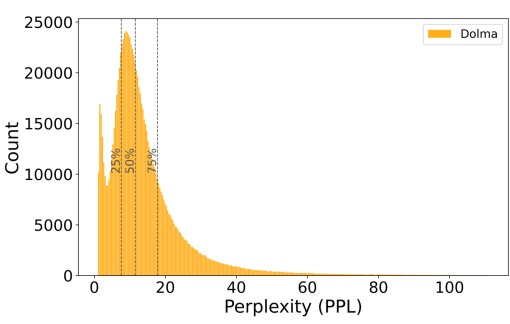 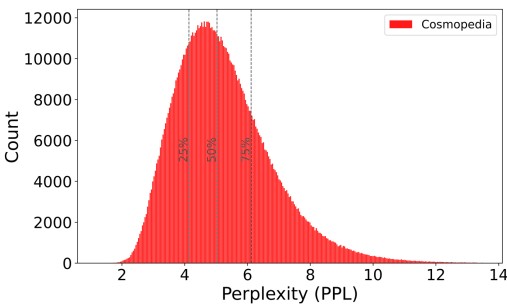

**A.** Human Data PPL Distribution Estimated by Llama-3-8B  **B.** Synthetic Data PPL Distribution Estimated by Llama-3-8B

Figure 3: PPL distribution of human and synthetic data estimated by Llama-3-8B. The synthetic data lacks the long tail of the human data and is also concentrated within the first 25% of the human data distribution. **A.** Distribution of human data is sharp with a long tail, spanning a wide range from 0 to over 100. **B.** The values are concentrated within a much narrower range, mostly between 0 and 12. The experiment uses Dolma v6 and Cosmopedia as human and synthetic data, each with sampled 6B tokens. More results in Figure 9.

human data, failing to capture the full range and complexity of real data distributions. Specifically, as illustrated in Figure 3A, human data exhibit a wide distribution in the range $[1, 100+]$, characterized by a sharp peak and a pronounced long tail. In contrast, as shown in Figure 3B, the synthetic data is confined to a narrower range of $[0, 14]$, displaying a smoother distribution. Further results of StabLM are shown in Figure 9. While the absolute PPL ranges estimated by different models may vary, the relative shapes and proportional ranges of these two distributions remain consistent. This phenomenon provides evidence that when scaling up to larger synthetic datasets, there is a notable absence of the long tail. Furthermore, we also observe a more severe coverage collapse. This limited coverage reduces the data's ability to generalize well and may contribute to model collapse in Figure 2.

**Finding II.ii Synthetic data over-concentrates N-gram features.** Based on the above distribution estimate, we further analyze why synthetic data fails at the feature level. Figure 10 and 11 demonstrate that synthetic data exhibits higher frequencies in certain bi-grams than human data. To further examine feature-level differences, we hash unigram and bigram features into 10,000 hash buckets. As illustrated in Figure 4, the response range of human data is noticeably broader, while the features of synthetic data are primarily concentrated in a few specific buckets. This indirectly supports our earlier observation of feature over-concentration. We then expanded the hash bucket range to $1,000 \times 20,000$ buckets and used a locality-sensitive hashing method to differentiate the features more precisely. However, the results remained similar. As shown in Figure 12, the majority of the response values are close to zero. The features of synthetic data are difficult to distinguish.

**Finding II.iii Distribution shifting cannot be mitigated through data selection.** Inspired by recent data selection works (Xie et al., 2023; Albalak et al., 2024), we try to leverage the human data features as reference distribution to select synthetic samples. We implement importance sampling in DSIR (Xie

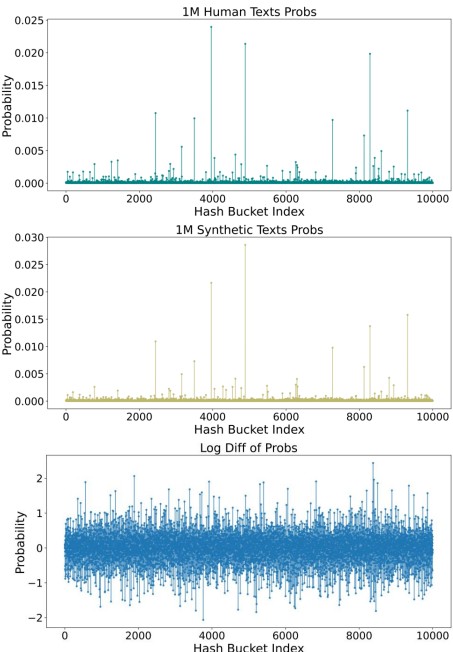

Figure 4: Uni/Bi-gram feature distribution across 10,000 hash buckets.

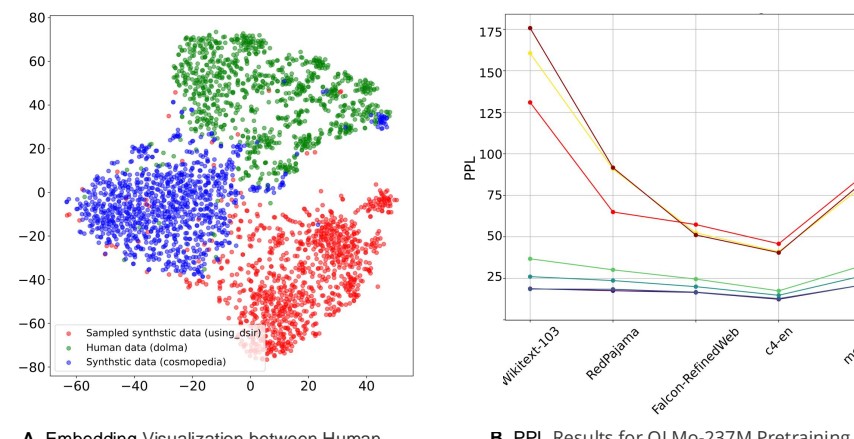

**A.** Embedding Visualization between Human, Synthetic, and DSIR-Selected Data using t-SNE

**B.** PPL Results for OLMo-237M Pretraining on Selected Synthetic Data and Data Mixtures

Figure 5: **A.** Embedding visualization using t-SNE and sentence-transformers. **B.** pretraining results for selected synthetic data and other data mixtures.

et al., 2023) to filter synthetic data. As shown in Figure 5A, the sampled data still fails to align with real data in the embedding space, even at the boundary regions of the synthetic data. As illustrated in Figure 5B, the training results of selected synthetic samples still fluctuates around the original performance of the synthetic data, indicating that even biased sampling cannot correct the distributional shift.

### 2.3 PROPOSED STRATEGY

Following these lessons so far, due to the properties of coverage collapse and feature overconcentration of synthetic data, our best option is to use totally human data and avoid the inclusion of synthetic data. However, we are still wondering how we can use synthetic data to improve human data. We arrive at a general guideline for synthetic data: full synthetic data will result in model collapse, so we need to keep the main human data distribution. In that case, we propose token-level editing, which leverages a prior distribution to edit the data. Our method can maintain the source distribution and improve the source data, called semi-synthetic data.

## 3 TOKEN-LEVEL DATA EDITING

In this section, we introduce token-level data editing to obtain semi-synthetic data. Furthermore, we provide theoretical analysis and proof that our method's test squared error has a finite upper bound, independent of the number of iterations. In this case, our method not only avoids model collapse but also obtains better performance.

### 3.1 METHOD

We formulate data synthesis as follows: assuming $P$ is a prior distribution, given a sequence of tokens $\mathbf{x} = (x_1, \ldots, x_t)$, the full synthetic data is $\mathbf{y} = (y_1, \ldots, y_n)$. The synthesis process is derived as:

$$P(y_1, \ldots, y_n \mid x_1, \ldots, x_t) = \prod_{i=1}^{n} P(y_i \mid y_1, \ldots, y_{i-1}, x_1, \ldots, x_t). \tag{2}$$

This conditional probability formulation outlines the generation of synthetic data conditioned on the given token sequence. Then the synthetic data is used to train models.

Inspired by previous studies of data selection (Mindermann et al., 2022; Ankner et al., 2024; Lin et al., 2024), prior distribution can be a pointer to indicate the useless or learnable samples. In

this case, we use a pre-trained language model to infer the pretraining corpus. As illustrated in Figure 6, even a model pre-trained on trillions of tokens can not fit the pretraining corpus perfectly. Specifically, 75% is under 0.6 probability. The tokens with both high and low probabilities are the most concentrated, suggesting the potential for data filtering. We leverage this U-shape distribution as a pointer to resample tokens. Specifically, we use a language model as prior distribution to compute each token's conditional probability $P(\cdot|\mathbf{x})$ if the probability exceeds a certain threshold $P(\cdot|\mathbf{x}) \geq p$, it indicates that this token is easy to learn, and we proceed with resampling at that point.

Token-level Editing doesn't generate the whole sequence but leverages conditional probability $P(x_i \mid x_1, \ldots, x_{i-1})$ to revise the input sequence. In this way, we can avoid using purely synthetic data while modifying the dataset to preserve human long-tail features, aiming to obtain higher-quality semi-synthetic data. Token-level Editing can be formulated as follows:

$$x_i' = \begin{cases} x_i, & \text{if } P(x_i \mid x_1, \ldots, x_{i-1}) < p \\ \tilde{x}i, & \text{if } P(x_i \mid x_1, \ldots, xi - 1) \geq p \end{cases} \tag{3}$$

Where $x_i'$ is the final token in the edited sequence. $\tilde{x}_i$ is a token resampled from a prior distribution. We can control the size of p that balances between retaining the structure of human data and avoiding overfitting to the synthetic data.

---

**Algorithm 1** Token-level Editing

1: **Input:** Sequence of tokens $\mathbf{x} = (x_1, \ldots, x_t)$, prior distribution $P$, probability threshold $p$
2: **Output:** Edited sequence **x'** $= (x_1', \ldots, x_t')$
3: **for** each token $x_i$ in sequence **x do**
4:     Compute conditional probability $P(x_i \mid x_1, \ldots, x_{i-1})$
5:     **if** $P(x_i \mid x_1, \ldots, x_{i-1}) \geq p$ **then**
6:         Resample token $\tilde{x}_i$ from prior distribution
7:         Set $x_i' \leftarrow \tilde{x}_i$
8:     **else**
9:         Set $x_i' \leftarrow x_i$
10:     **end if**
11: **end for**
12: **Return:** Edited sequence **x'** $= (x_1', \ldots, x_t')$

---

### 3.2 THEORETICAL ANALYSIS

To investigate more mathematical insights, we utilize an analytical framework of the linear model and adopt notions in prior research (Mobahi et al., 2020; Dohmatob et al., 2024a; Gerstgrasser et al., 2024). This theoretical framework primarily considers a linear model that iteratively trains on its own generated data, similar to pipelines like self-play and self-distillation, but without complex constraints. It simply involves training continuously on the data generated by the previous generation of the model. Dohmatob et al. (2024a) point out that with iterative training, test errors accumulate progressively, eventually leading to model collapse. Based on this theoretical framework, we incorporate our proposed token-level editing into the framework and analyze whether our method can prevent model collapse.

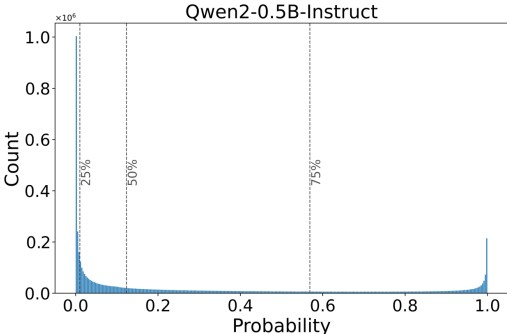

Figure 6: U-shape token probability distribution of Dolma-sampled V6 estimated by Qwen-0.5B-Instruct (qwe, 2024).

**Notation and Preliminaries** For a given distribution $P_{\Sigma,w,\sigma^2}$, the data $(x, y) \sim P_{\Sigma,w,\sigma^2}$ on $\mathbb{R}^d \times \mathbb{R}$, where $x$ is drawn from a multivariate normal

distribution $x \sim \mathcal{N}(0, \Sigma)$, $\epsilon$ is an independent noise term sampled from $\mathcal{N}(0, \sigma^2)$, and the label $y$ is given by the linear model $y = x \cdot w^* + \epsilon$.

**Iterative Data Editing Process** We utilize the model obtained from the previous round of training to make a limited number of modifications. Specifically, we resample and replace data points with relatively high confidence. The editing operations are defined by the matrices $\{M_1, M_2, \ldots, M_n\}$. The iterative data synthesis and model-fitting process can be formalized as follows:

$$P_{\Sigma, w^*, \sigma^2} \to P_{\Sigma, \hat{w}_1, \sigma^2} \to \ldots \to P_{\Sigma, \hat{w}_n, \sigma^2},$$

where $n$ is the number of iterations. The detailed process of data editing and iterations is described as follows:

For $n = 1$, we begin by initializing the covariates/features as $\tilde{X}_1 = X$. The target values are defined as $\tilde{Y}_1 = \hat{Y}_1 = Xw^* + E_1$, where $E_1 \sim \mathcal{N}(0, \sigma^2 I_T)$. The linear model is then fitted by solving for $\hat{w}_1 = \tilde{X}_1^\dagger \tilde{Y}_1$. To proceed to the next iteration, we resample the data, obtaining $\hat{Y}_2 = X\hat{w}_1 + E_2$, with $E_2 \sim \mathcal{N}(0, \sigma^2 I_T)$.

For $n \geq 2$, the input covariates/features remain as $\tilde{X}_n^\top = X$, while the target values are updated using the edited targets, following the equation $\tilde{Y}_n^\top = M_{n-1}\hat{Y}_n + (1 - M_{n-1})\tilde{Y}_{n-1}$. The linear model is then fitted by computing $\hat{w}_n = \tilde{X}_n^\dagger \tilde{Y}_n$. Finally, the data is resampled for the next iteration, yielding $\hat{Y}_{n+1} = X\hat{w}_n + E_{n+1}$, where $E_{n+1} \sim \mathcal{N}(0, \sigma^2 I_T)$.

The matrix $M_n$ is a diagonal matrix, where some elements on the diagonal are 1 and others are 0. The multiplication by M can be interpreted as an operation that selectively modifies certain data points (those corresponding to 1s) while retaining others (those corresponding to 0s). Then, the data editing process can be formulated as follows:

$$\tilde{Y}_n^\top = M_{n-1}\hat{Y}_n + (1 - M_{n-1})\tilde{Y}_{n-1} \tag{4}$$

where $\tilde{Y}_{n-1}$ is the data after editing in the $n-1$ generation, and $\hat{Y}_n$ is the synthetic data from the $n$-th generation. This process can be described as: firstly, synthesizing labels for all inputs; secondly, the $M$ matrix determining which data is edited and which is retained. For a matrix $A$ with full column rank, its Moore-Penrose pseudo-inverse is $A^+ = (A^\top A)^{-1}A^\top$. The noise terms $E_1, E_2, \ldots, E_n$ are independent of each other and the covariates/features. Since $X$ has full column rank, $\tilde{X}_n$ retains this property for all $n \geq 1$.

**Test Error** Model collapse is ultimately reflected through test error, and here we follow previous work (Gerstgrasser et al., 2024) to define the standard test error. For any linear estimator $\hat{w}$ derived from the training data, we evaluate the test error using the standard method as follows:

$$E_{test}(w) \overset{\text{def}}{=} \mathbb{E}\left[(x_{test}^T w - y_{test})^2\right] - \sigma^2 = \mathbb{E}[\|w - w^*\|_\Sigma^2] \tag{5}$$

where the expectation is computed with respect to the training data, while the test pair $(x_{\text{test}}, y_{\text{test}})$ is sampled from $P_{\Sigma, w^*, \sigma^2}$ independently of the training set.

### 3.3 Test Error Under Data Editing

Our goal is to derive an analytical expression for the test error of the $n$-th model in the data editing setting. As indicated by the test error in Eq. 5, this requires two steps: (1) establishing the relationship between the fitted linear parameters $w_n$ and the true parameters $w^*$, and (2) simplifying the test error expression. We start by establishing the formulation between $w_n$ and $w^*$. Proofs are detailed in App. B.

**Theorem 1** *In the data editing setting, $\forall n \geq 1$, the fitted linear parameters $\hat{w}_{n+1}$ can be derived as:*

$$\hat{w}_{n+1} = w^* + (X^\top X)^{-1}X^\top \left(E_1 + \sum_{i=1}^{n} M_i E_{i+1}\right) \tag{6}$$

*where, $w^*$ is the true parameter, $X$ is the original design matrix, $E_i$ is the extra noise added at the $i$'th iteration, and $M_i$ is an idempotent diagonal matrix, defining the editing operation.*

**Theorem 2** *Consider an $n+1$ fold data editing process with $T \geq d+2$ samples per iteration and isotropic features ($\Sigma \stackrel{def}{=} I_d$), the test error for the ridgeless linear model $\hat{w}_n$ learned on the edited data up to iteration $n+1$, is bounded by:*

$$E_{test}(\hat{w}_{n+1}) \leq \frac{2\sigma^2 d}{T - d - 1} \tag{7}$$

*Furthermore, assuming the editing operation satisfies $||M_i|| = ||M_{i-1}||\eta$ with $\eta \in (0, 1)$, the test error can be further bounded by:*

$$E_{test}(\hat{w}_{n+1}) \leq \frac{\sigma^2 d}{T - d - 1} + \sigma^2 \sqrt{\mathbb{E}\left[tr\left((X^\top X)^{-2}\right)\right]} \cdot \frac{\sqrt{\mathbb{E}\left[tr(M_1)\right]}}{1 - \eta} \tag{8}$$

Recalling that the cause of model collapse (Dohmatob et al., 2024a): training iteratively on synthetic data leads to an accumulation of error over iterations, as shown in the following equation:

$$E_{test}^{collapse}(\hat{w}_n) = \frac{\sigma^2 d}{T - d - 1} \times n \tag{9}$$

Compared Eq. 7 with Eq. 9, the error in data editing is bounded by a fixed value, preventing continuous error accumulation and thus avoiding model collapse. Combining the above theoretical derivations and statistical analysis of synthetic data (§ 2.1), the underlying reason is that our approach retains the coverage of the initial distribution. We move away from pure data synthesis toward token-level data editing, which allows us to obtain better data while avoiding model collapse. Moreover, remarkable previous studies (Dohmatob et al., 2024b; Gerstgrasser et al., 2024) pointed out similar conclusions. They indicated mixing real data with synthetic data will break model collapse and provide an upper bound under data accumulation. Different from their work, our data editing aims to yield better data, enabling synthetic data to perform well both in theory and practice, not only avoiding model collapse.

## 4 EXPERIMENTS

To validate our proposed method, we conduct experiments across three stages of language model training including: pre-training, continual pre-training (CPT) and supervised fine-tuning (SFT).

### 4.1 IMPLEMENTATION

We use the Llama-3-8B (AI@Meta, 2024) as a prior distribution to estimate the token distribution in each text sample. The modification probability is set to $p = 0.99$. This means that we resample tokens in positions where the probability exceeds $p$, and the resampling is based on the conditional probability given the preceding context. The entire process of our method requires only a single forward pass, without auto-regressive generation. We integrate the fast inference engine vLLM (Kwon et al., 2023), allowing the entire data editing process to be completed on a single 4090 GPU. After completing the data editing, we compared the original data and the edited data on language model training performance across pre-training, CPT, and SFT. Here, we used top-k as the sampling strategy with $k = 8$. We also experimented with top-p and rejection sampling, which produced similar results.

### 4.2 DATASETS AND MODELS

Here, we provide an overview of our experimental setup. More training details are presented in Appendix D. **As for pre-training**, we pre-train the 1B OLMo model (Groeneveld et al., 2024) from scratch, using Dolma-sampled V6 (6B tokens) as the pre-training corpus. Dolma (Soldaini

et al., 2024) is the largest open-source pre-training corpus available. We use 8 general tasks in lm-evaluation-harness (Gao et al., 2024) to evaluate for pre-training models. **As for continual pre-training**, we follow Cheng et al. (2024b) to continual pre-train the OLMo-1B (Groeneveld et al., 2024) and Llama-3-8B (AI@Meta, 2024) on Biomedicine, Finance and Math. Each domain corpus contains 1B tokens. Correspondingly, we evaluate the continual pre-training models using 15 downstream tasks, with 5 tasks from each domain. **As for supervised fine-tuning**, we fine-tune Llama-3-8B on instruction tuning tasks. We use natural-instructions (Wang et al., 2022b), as fine-tuning data, which consists of over 1500 tasks. We evaluate the SFT models using 5 downstream tasks designed to measure instruction-following capabilities. All Llama-3-8B experiments use LoRA (Hu et al., 2021), while the OLMo-1B model is trained with full parameters.

Table 1: Domain specific tasks performance for continual pretraining models. CPT indicates continual pre-training. $\Delta$ indicates training with our edited data. Our method shows consistent improvements across three domains on OLMo-1B and Llama-3-8B.

| Biomedicine | | | | | | |
|---|---|---|---|---|---|---|
| Models | MQP | ChemProt | PubMedQA | RCT | USMLE | Average |
| OLMo-1B | 52.59 | 17.2 | 51.40 | 32.70 | 28.90 | 36.63 |
| CPT | 52.29 | 21.00 | 58.50 | 34.90 | 27.49 | 38.83 |
| $\Delta$ ToEdit | 54.59 | 22.40 | 65.00 | 34.50 | 27.96 | **40.89** |
| LLama-3-8B | 66.80 | 28.59 | 60.8 | 73.85 | 40.61 | 54.13 |
| CPT | 72.29 | 29.4 | 69.1 | 72.65 | 36.76 | 56.04 |
| $\Delta$ ToEdit | 76.39 | 30.2 | 65.3 | 73.30 | 37.23 | **56.48** |
| Finance | | | | | | |
| Models | HeadLine | FPB | FiQA_SA | ConvFinQA | NER | Average |
| OLMo-1B | 69.00 | 47.03 | 48.05 | 4.83 | 62.19 | 46.22 |
| CPT | 70.31 | 49.78 | 40.36 | 18.72 | 60.44 | 47.92 |
| $\Delta$ ToEdit | 71.77 | 51.39 | 46.06 | 18.85 | 62.97 | **50.21** |
| LLama-3-8B | 81.28 | 63.58 | 81.60 | 52.88 | 72.53 | 70.37 |
| CPT | 85.68 | 54.22 | 81.88 | 67.78 | 67.43 | 71.40 |
| $\Delta$ ToEdit | 83.83 | 61.61 | 80.82 | 67.31 | 67.62 | **72.24** |
| Math | | | | | | |
| Models | Arc-Challenge | GPQA | GSM8K | MATH | MMLU | Average |
| OLMo-1B | 28.67 | 24.23 | 1.67 | 0.00 | 26.56 | 16.23 |
| CPT | 28.41 | 24.03 | 1.52 | 0.10 | 27.23 | 16.26 |
| $\Delta$ ToEdit | 28.92 | 28.12 | 2.20 | 0.10 | 23.63 | **16.59** |

## 4.3 RESULTS

Table 1, 2, and 3 respectively demonstrate the effectiveness of our method in continual pre-training, pre-training, and fine-tuning tasks. Across these three stages of language model training, our method enhances the model's performance on downstream tasks without increasing the data size. Our method further taps into the potential of existing data, also demonstrating that semi-synthetic data is a viable path to obtaining higher-quality data.

Specifically, as shown in Table 1, our method shows consistent improvements over the source data across OLMo-1B and LLaMA-3-8B. For instance, in the Biomedicine domain, the average score for OLMo-1B increased from 36.63 to 40.89 with ToEdit, while LLaMA-3-8B saw an increase from 54.13 to 56.48. Table 2 further supports the effectiveness of our approach in pre-training. The average performance of OLMo-1B increases from 32.75 to 33.11, reflecting improved generalization capabilities. While the improvement is modest, the consistent trend across tasks like PIQA, BoolQ, and ARC-c highlights the broader applicability of our method.

As for SFT results in Table 3, using both the original and edited data, the results indicate a small but consistent improvement. Specifically, ToEdit improves orignal FLAN V2, with average performance increasing from 70.18 to 70.65. As for Natural Instructions, the average performance of

LLaMA-3-8B improves from 69.34 to 69.70, with gains in tasks like Winogrande and SIQA. These improvements, demonstrate the adaptability of our method to instruction-tuning tasks. For code-related tasks, the improvements are particularly evident in ARC-Challenge and GPQA, indicating better reasoning and code comprehension.

In summary, experiments on pretraining, continual pretraining, and SFT validate the effectiveness and versatility of our method. More ablation studies and discussions can be found Appendix F and E.

Table 2: General performance of the pre-trained base models. PT indicates we pre-train OLMo-1B from scratch. Experimental results demonstrate that our method can also enhance the effectiveness of pre-training.

| | PIQA | BoolQ | OBQA | ARC-c | ARC-e | HellaSwag | SIQA | Winogrande | Average |
|---|---|---|---|---|---|---|---|---|---|
| OLMo-1B (PT) | 53.97 | 38.26 | 12.20 | 17.23 | 28.36 | 26.02 | 34.80 | 51.14 | 32.75 |
| Δ ToEdit | 54.13 | 38.65 | 12.80 | 18.43 | 27.48 | 25.94 | 34.95 | 52.49 | **33.11** |

Table 3: Performance of the SFT models. We fine-tune LLaMA-3-8B using instruction tuning and code reasoning tasks, comparing performance with the edited version produced by our method. The experimental results indicate that our approach can enhance the data for instruction-tuning and code reasoning tasks.

| | Models | PIQA | BoolQ | HellaSwag | SIQA | Winogrande | Average |
|---|---|---|---|---|---|---|---|
| *Instruction Tuning* | | | | | | | |
| *Natural Instructions* | Llama-3-8B | 79.82 | 87.06 | 58.32 | 46.83 | 74.66 | 69.34 |
| | Δ ToEdit | 80.58 | 87.80 | 58.27 | 46.93 | 74.90 | **69.70** |
| *CoT* | Llama-3-8B | 79.87 | 81.28 | 59.72 | 49.69 | 74.51 | 69.01 |
| | Δ ToEdit | 80.25 | 81.16 | 59.74 | 50.56 | 74.59 | **69.26** |
| *FLAN V2* | Llama-3-8B | 80.79 | 84.04 | 59.98 | 51.43 | 74.66 | 70.18 |
| | Δ ToEdit | 80.69 | 85.20 | 59.99 | 52.00 | 75.37 | **70.65** |
| *Open Assistant 1* | Llama-3-8B | 79.65 | 83.18 | 60.51 | 48.52 | 74.11 | 69.19 |
| | Δ ToEdit | 79.98 | 83.91 | 60.34 | 48.31 | 74.66 | **69.44** |

| | Models | ARC-c | GPQA | GSM8K | MMLU | Average |
|---|---|---|---|---|---|---|
| *Code Reasoning* | | | | | | |
| *OSS-Instruct-75K* | Llama-3-8B | 51.28 | 27.46 | 49.58 | 62.14 | 45.76 |
| | Δ ToEdit | 51.79 | 28.79 | 49.36 | 62.04 | **46.13** |
| *Evol-Instruct-110K* | Llama-3-8B | 52.90 | 27.90 | 50.87 | 62.40 | 46.62 |
| | Δ ToEdit | 52.22 | 29.69 | 50.87 | 62.60 | **46.92** |

## 5 CONCLUSION

With the growing prevalence of generative AI models like ChatGPT (Achiam et al., 2023) and Stable Diffusion (Rombach et al., 2021), when training next-generation AI models, it will be inevitable to use a mixture of synthetic data and human data. Therefore, we focus on two key questions: (1) What is the impact of synthetic data on language model pre-training, and what are the underlying causes? (2) How can we prevent model collapse and synthesize high-quality data? We found that synthetic data can impair the effectiveness of pre-training when mixed with human data, leading to non-iterative model collapse. Statistical analysis reveals that synthetic data suffers from significant distribution gaps and overly concentrated n-gram features. Based on this, we propose token-level editing instead of relying purely on synthetic data. Specifically, we perform token resampling guided by a trained prior. Moreover, our method can theoretically prevent model collapse. Experimentally, our approach shows improvements over the source data across pre-training, continual pre-training, and supervised fine-tuning.

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

## A  RELATED WORK

**Model collapse**  Shumailov et al. (2024); Dohmatob et al. (2024a;b) demonstrate AI models trained recursively on data generated by earlier versions of themselves over time can result in performance degradation, ultimately rendering the AI model completely useless. This process can be formulated as follows:

$$E_{test}(\hat{w}_{n+1}) = \frac{\sigma^2 d}{T - d - 1} \times n$$

This indicates that the error will continuously increase with the number of iterations $n$. Dohmatob et al. (2024b) further pointed out that synthetic data also contribute to a truncation of the scaling law. This phenomenon stems from the sampling strategy (e.g., Top-p) used during the language model's generation process. Gerstgrasser et al. (2024) further adjusted the data iteration setting by replacing data replacement with data accumulation during the iterative process. They demonstrated that data accumulation can prevent model collapse. Inspired by the above work, we believe that training language models on synthetic datasets will be inevitable in the future. Therefore, it is crucial to theoretically discuss how to prevent model collapse. Building on the above theoretical framework, we proved that token-level editing establishes an upper bound during the iterative process, thereby preventing the continuous accumulation of errors.

**Synthetic Data**  Phi-1/2 (Gunasekar et al., 2023) demonstrated the synthetic data boost training efficiency and performance compared with raw data in language model pre-training. Liu et al. (2024) highlighted that synthetic data will play a crucial role in the development of AI. For example, synthetic data can be used to construct highly specialized datasets, enhancing the performance of downstream tasks. Trinh et al. (2024) utilized synthetic math data to train a 125M language model, which successfully solved 25 out of 30 selected problems from the International Mathematical Olympiad (IMO) problem set. Zhang et al. (2024) developed a biomedical instruction dataset that was used to train specialized bio-models, enabling them to excel in answering questions related to medical exams and clinical scenarios. Eldan & Li (2023) introduced a novel synthetic dataset and evaluation paradigm that enables small language models to generate coherent, diverse, and grammatically sound stories. As outlined above, in the post-training stages of LLMs, synthetic data enhances the ability of downstream tasks and aligns foundation models with humans. And Maini et al. (2024) proposed rephrasing the pre-training data into a Wikipedia or Q/A style to achieve better alignment with downstream tasks. Synthetic data is a powerful tool for training. Our approach is also based on synthetic data methods. Instead of sampling data solely based on this prior, we modify the data using the prior as a guide.

## B  PROOF

### B.1  PROOF OF THEOREM 1

For $n = 1$, we have:

$$\hat{w}_1 = \tilde{X}_1^\dagger \tilde{Y}_1 = (X^\top X)^{-1} X^\top (X w^* + E_1) = w^* + (X^\top X)^{-1} X^\top E_1$$

For $n \geq 1$, we have:

$$\begin{aligned}
\hat{w}_{n+1} &= \tilde{X}_{n+1}^\dagger \tilde{Y}_{n+1} \\
&= (\tilde{X}_{n+1}^\top \tilde{X}_{n+1})^{-1} \tilde{X}_{n+1}^\top \tilde{Y}_{n+1} \\
&= (X^\top X)^{-1} X^\top \tilde{Y}_{n+1}
\end{aligned}$$

Recalling that:

$$\tilde{Y}_i = \begin{cases} X w^* + E_1, & \text{if } i = 1 \\ M_{i-1}(X \hat{w}_{i-1} + E_i) + (1 - M_{i-1})\tilde{Y}_{i-1}, & \text{if } 2 \leq i \leq n+1 \end{cases}$$

Substituting this $\tilde{Y}_i$ into the expression for $\hat{w}_{n+1}$:

We begin the data editing data process:

$$\tilde{Y}_2 = M_1(X\hat{w}_1 + E_2) + (1 - M_1)\tilde{Y}_1 \tag{10}$$

Then:

$$\tilde{Y}_3 = M_2(X\hat{w}_2 + E_3) + (1 - M_2)\tilde{Y}_2 \tag{11}$$

We have:

$$\tilde{Y}_3 = M_2(X\hat{w}_2 + E_3) + (1 - M_2)\left(M_1(X\hat{w}_1 + E_2) + (1 - M_1)\tilde{Y}_1\right)$$

$$= M_2(X\hat{w}_2 + E_3) + (1 - M_2)M_1(X\hat{w}_1 + E_2) + (1 - M_2)(1 - M_1)\tilde{Y}_1$$

We can expand $\tilde{Y}_{n+1}$ by recursively substituting the previous expressions:

$$\tilde{Y}_{n+1} = M_n(X\hat{w}_n + E_{n+1}) + (1 - M_n)\tilde{Y}_n \tag{12}$$

$$= M_n(X\hat{w}_n + E_{n+1}) + (1 - M_n)\left[M_{n-1}(X\hat{w}_{n-1} + E_n) + (1 - M_{n-1})\tilde{Y}_{n-1}\right] \tag{13}$$

$$= M_n(X\hat{w}_n + E_{n+1}) + (1 - M_n)M_{n-1}(X\hat{w}_{n-1} + E_n) + (1 - M_n)(1 - M_{n-1})\tilde{Y}_{n-1} \tag{14}$$

$$\vdots \tag{15}$$

$$= \sum_{i=1}^{n}\left[\left(\prod_{j=i+1}^{n}(1 - M_j)\right)M_i(X\hat{w}_i + E_{i+1})\right] + \left(\prod_{j=1}^{n}(1 - M_j)\right)\tilde{Y}_1 \tag{16}$$

Recalling properties of $M_i$:

$$M_i(1 - M_i) = 0 \quad \text{and} \quad (1 - M_i)M_i = 0 \tag{17}$$

$$M_iM_j = 0 \quad \text{for} \quad i \neq j \tag{18}$$

$$(1 - M_i)(1 - M_j) = 1 - M_i - M_j \quad \text{for} \quad i \neq j \tag{19}$$

$$\tag{20}$$

Then we have:

$$\tilde{Y}_{n+1} = \sum_{i=1}^{n} M_i(X\hat{w}_i + E_{i+1}) + \left(1 - \sum_{i=1}^{n} M_i\right)\tilde{Y}_1 \tag{21}$$

$$= \sum_{i=1}^{n} M_i(X\hat{w}_i + E_{i+1}) + \left(1 - \sum_{i=1}^{n} M_i\right)(Xw^* + E_1) \tag{22}$$

$$= Xw^* + E_1 + \sum_{i=1}^{n} M_i\left(X(\hat{w}_i - w^*) + (E_{i+1} - E_1)\right) \tag{23}$$

Substituting this back into the expression for $\hat{w}_{n+1}$:

$$\hat{w}_{n+1} = (X^\top X)^{-1}X^\top\left[Xw^* + E_1 + \sum_{i=1}^{n} M_i\left(X(\hat{w}_i - w^*) + (E_{i+1} - E_1)\right)\right] \tag{24}$$

$$= w^* + (X^\top X)^{-1}X^\top\left[E_1 + \sum_{i=1}^{n} M_iX(\hat{w}_i - w^*) + \sum_{i=1}^{n} M_i(E_{i+1} - E_1)\right] \tag{25}$$

We can observe:

$$\hat{w}_1 = (X^\top X)^{-1} X^\top (Xw^* + E_1) = w^* + (X^\top X)^{-1} X^\top E_1 \tag{26}$$

$$\hat{w}_2 = w^* + (X^\top X)^{-1} X^\top \left( M_1 X (X^\top X)^{-1} X^\top E_1 + M_1 E_2 + (1 - M_1) E_1 \right) \tag{27}$$

$$= w^* + (X^\top X)^{-1} X^\top (E_1 + M_1 E_2) \tag{28}$$

We prove this Theorem 1 by induction.

**Inductive Step:**  Assume the formula holds for $n$, we have:

$$\hat{w}_{n+1} = w^* + (X^\top X)^{-1} X^\top (E_1 + M_1 E_2 + M_2 E_3 + \cdots + M_n E_{n+1}) \tag{29}$$

$$= w^* + (X^\top X)^{-1} X^\top \left( E_1 + \sum_{i=1}^{n} M_i E_{i+1} \right) \tag{30}$$

Substitute $\hat{w}_i$ into $\hat{w}_{n+1}$:

Then we can get:

$$\hat{w}_{n+1} = w^* + (X^\top X)^{-1} X^\top \left[ E_1 + \sum_{i=1}^{n} M_i P \left( E_1 + \sum_{j=1}^{i-1} M_j E_{j+1} \right) + \sum_{i=1}^{n} M_i (E_{i+1} - E_1) \right] \tag{31}$$

$$= w^* + (X^\top X)^{-1} X^\top \left[ E_1 + \sum_{i=1}^{n} M_i \left( E_{i+1} + \sum_{j=1}^{i-1} M_j E_{j+1} \right) \right] \tag{32}$$

$$= w^* + (X^\top X)^{-1} X^\top \left( E_1 + \sum_{i=1}^{n} M_i E_{i+1} \right) \tag{33}$$

$$\text{where} \quad P = X (X^\top X)^{-1} X^\top, \tag{34}$$

The above derivation aligns with Theorem 1, and the proof is complete.

## B.2  PROOF OF THEOREM 2

We substitute the Eq. 30 into Test Error Eq. 5:

$$E_{test}(\hat{w}_{n+1}) = \mathbb{E} \left[ \left\| (X^\top X)^{-1} X^\top \left( E_1 + \sum_{i=1}^{n} M_i E_{i+1} \right) \right\|_\Sigma^2 \right] \tag{35}$$

$$= \mathbb{E} \left[ \left( E_1 + \sum_{i=1}^{n} M_i E_{i+1} \right)^\top X (X^\top X)^{-2} X^\top \left( E_1 + \sum_{i=1}^{n} M_i E_{i+1} \right) \right] \tag{36}$$

$$= \sigma^2 \mathbb{E} \left[ \text{tr} \left( (X^\top X)^{-1} \right) \right] + \sigma^2 \sum_{i=1}^{n} \mathbb{E} \left[ \text{tr} \left( M_i (X^\top X)^{-1} M_i \right) \right] \tag{37}$$

$$= \sigma^2 \mathbb{E} \left[ \text{tr} \left( (X^\top X)^{-1} \right) \right] + \sigma^2 \sum_{i=1}^{n} \mathbb{E} \left[ \text{tr} \left( (X^\top X)^{-1} M_i \right) \right] \tag{38}$$

Further, by applying the Cauchy-Schwarz inequality (Rudin, 1976), we obtain:

$$E_{test}(\hat{w}_{n+1}) \le \sigma^2 \mathbb{E} \left[ \text{tr} \left( (X^\top X)^{-1} \right) \right] + \sigma^2 \sqrt{\mathbb{E} \left[ \text{tr} \left( (X^\top X)^{-2} \right) \right]} \cdot \sum_{i=1}^{n} \sqrt{\mathbb{E} \left[ \text{tr}(M_i) \right]} \tag{39}$$

We refer to the following lemma (Dohmatob et al., 2024a), which is essential for proving Theorem 2:

**Lemma 3** *Let $T$ and $d$ be positive integers with $T \geq d+2$, and let $X \in \mathbb{R}^{T \times d}$ be a random matrix with i.i.d. rows from $\mathcal{N}(0, \Sigma)$ with $\Sigma$ positive definite. Then, $X$ has full rank a.s. Moreover, it holds that:*

$$\mathbb{E}_X \left[ (X^\top X)^{-1} \right] = \frac{1}{T - d - 1} \Sigma^{-1}. \tag{40}$$

Using Lemma 3, we have:

$$E_{test} \left[ \text{tr} \left( (X^\top X)^{-1} \right) \right] = \frac{d}{T - d - 1} \tag{41}$$

Then, we have:

$$E_{test}(\hat{w}_{n+1}) = \sigma^2 \mathbb{E} \left[ \text{tr} \left( (X^\top X)^{-1} \right) \right] + \sigma^2 \sum_{i=1}^n \mathbb{E} \left[ \text{tr} \left( (X^\top X)^{-1} M_i \right) \right] \tag{42}$$

$$\leq \frac{\sigma^2 d}{T - d - 1} + \sigma^2 \sqrt{\mathbb{E} \left[ \text{tr} \left( (X^\top X)^{-2} \right) \right]} \cdot \sum_{i=1}^n \sqrt{\mathbb{E} \left[ \text{tr}(M_i) \right]} \tag{43}$$

In our setting, the data is incrementally modified over iterations and modifications decreases progressively. This behavior can be modeled by the sum of a geometric series, where the amount of modified data decreases by a fixed ratio $\eta$ with each iteration. Then, we assume the editing operation as $||M_i|| = ||M_{i-1}|| \eta$, for $i = 1, 2, \ldots, n$. Therefore, the test error for data editing can be bounded:

$$E_{test}(\hat{w}_{n+1}) \leq \frac{\sigma^2 d}{T - d - 1} + \sigma^2 \sqrt{\mathbb{E} \left[ \text{tr} \left( (X^\top X)^{-2} \right) \right]} \cdot \frac{\sqrt{\mathbb{E} \left[ \text{tr}(M_1) \right]}}{1 - \eta} \tag{44}$$

Additionally, since $M_i$ is not full-rank, as seen from Eq. 38, we can apply a more relaxed and simplified bound, as follows:

$$E_{test}(\hat{w}_{n+1}) \leq \frac{2\sigma^2 d}{T - d - 1} \tag{45}$$

Thus, the above derivation satisfies the Theorem 2.

## C    MORE RESULTS OF HUMAN AND SYNTHETIC DATA MIXTURE TRAINING

We provide more training results for the human and synthetic data mixture. The main results and analysis can be found in Sec 2.1. Except for GPT-2 pretraining, we also use the OLMo models (Groeneveld et al., 2024) for further experiments.

As shown in Figure 8, the training loss continues to decrease as the amount of synthetic data increases, which is consistent with GPT-2 pretraiing in Figure 2. More synthetic data can lead to better fitting. However, a lower loss does not necessarily mean a better model. As illustrated in Figure 2B and 7, models that fits better perform worse in real world tasks.

Furthermore we follow Maini et al. (2024) to conduct more experiments including PPL results on 22 validation sets of Pile (Gao et al., 2020) and general understanding tasks. The additional results in Table 5 and 6 are consistent with our findings. Specifically, the PPL increases as the proportion of purely synthetic data grows, while the performance on downstream tasks similarly exhibits a gradual decline with the increase in synthetic data.

## D    DETAILED EXPERIMENT SETTINGS

In this section, we describe our experiments settings detailed.

## D.1 Training

**Pre-training** We utilized both GPT-2 and OLMo models. The pre-training datasets included Dolma, representing real data, and Cosmopedia, representing synthetic data. For GPT-2, we employed the official FSDP (Fully Sharded Data Parallel) framework provided by Torch for training. For OLMo[1], we used the official open-source computational code, which also incorporates the FSDP framework alongside Flash Attention for acceleration.

**Continual Pre-training** We follow Cheng et al. (2024b) to conduct continual pre-training on Bio, Finance, and Math domains. Specifically, PubMed Abstracts from the Pile are utilized as the pre-training corpora for the biomedicine domain. For the finance domain, financial news data covering over 7,000 stocks from May 2022 to May 2023 is collected using the FinGPT framework. We continue pre-training OLMo-1B and LLaMA-3-8B on each domain. For implementation, we utilized the official training framework for OLMo-1B, leveraging Fully Sharded Data Parallel (FSDP) for continual pre-training. For LLaMA, we adopted the LLaMA-Factory framework to carry out the continual pretraining process.

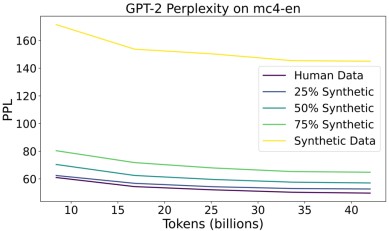

Figure 7: GPT-2 perplexity (PPL) on validation sets, trained from scratch.

Our experiments was primarily conducted on OLMo-1B and Llama-3-8B models, with Llama-3-8B utilizing LoRA (Low-Rank Adaptation) for parameter-efficient fine-tuning. The data and evaluation are given in this repo[2]. We conducted the continual pretraining on a total of 1B tokens.

**Supervised Fine-tuning** We used the Llama-Factory (Zheng et al., 2024) framework to fine-tune Llama-3-8B. As for general instruction tuning tasks, we adopt instruction tuning datasets from (Xia et al., 2024) [3], including CoT (Wei et al., 2022) , FLAN V2 (Longpre et al., 2023), and Open Assistant 1 (Kopf et al., 2023). As for code-related reasoning tasks, we utilize OSS-Instruct-75K [4] and Evol-Instruct-110K [5]. These datasets provide sufficient diversity for verification on fine-tuning.

## D.2 Evaluation

**Pre-training** We use PPL and downstream tasks to conduct analysis and performance test. As for PPL, it stands for perplexity, a commonly used metric in NLP to evaluate the quality of language models. It measures how well a probabilistic model predicts a given dataset, with lower values indicating better performance. Formally, the perplexity of a language model is calculated as:

$$PPL = 2^{-\frac{1}{N}\sum_{i=1}^{N}\log_2 P(x_i)}$$

Alternatively, it can also be expressed as:

$$PPL = \exp\left(-\frac{1}{N}\sum_{i=1}^{N}\log P(x_i)\right)$$

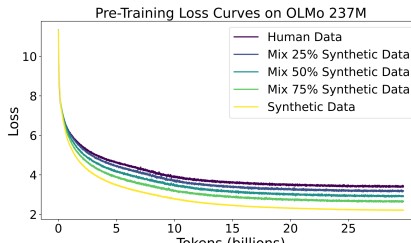

Figure 8: OLMo-237M pretraining with mixed human and synthetic data proportions. We pretrain the OLMo-237M model using a mixture of human data (Dolma (Soldaini et al., 2024)) and synthetic data (Cosmopedia (Ben Allal et al., 2024)).

---

[1] https://github.com/allenai/OLMo
[2] https://github.com/microsoft/LMOps/tree/main/adaptllm
[3] https://huggingface.co/datasets/princeton-nlp/less_data
[4] https://huggingface.co/datasets/ise-uiuc/Magicoder-OSS-Instruct-75K
[5] https://huggingface.co/datasets/ise-uiuc/Magicoder-Evol-Instruct-110K

Where $N$ is the number of tokens in the dataset, and $P(x_i)$ is the predicted probability of the $i$-th token. Perplexity essentially represents the exponential of the average negative log-likelihood of the predicted tokens, indicating how "perplexed" the model is when making predictions.

As for downstream tasks, we use general understanding tasks in (Maini et al., 2024) to analyze model collapse in Table 5 and general test tasks in (Cheng et al., 2024a) to test our methods in Table 2. All downstream tasks we used can be found in (Gao et al., 2024)[6].

**Continual Pre-training**    We use the domain specific task in (Cheng et al., 2024b) to test domain CPT performance. The test data and code can be found in here[7].

**Supervised Fine-tuning**    We utilize the general downstream tasks from (Cheng et al., 2024a) to evaluate instruction-tuning performance and reasoning tasks to assess reasoning capabilities. All downstream tasks we used can be found in (Gao et al., 2024)[8].

Table 4: Performance impact of different $p$ in BioMed.

| Criteria | PubMedqa | MQP | RCT | USMLE | ChemProt | Avg |
|---|---|---|---|---|---|---|
| Resampled Tokens $p \geq 0.99$ | 64.5 | 55.73 | 30.95 | 27.65 | 14.6 | 38.686 |
| Resampled Tokens $p \geq 0.999$ | 63.6 | 55.4 | 29.09 | 28.12 | 16.2 | 38.482 |
| Resampled Tokens $p \leq 0.1$ | 62.4 | 51.47 | 25.6 | 29.14 | 10.0 | 35.722 |
| Resampled Tokens $p \leq 0.01$ | 65.4 | 54.91 | 28.19 | 27.80 | 11.0 | 37.46 |
| Resampled Tokens $p \leq 0.001$ | 64.2 | 56.39 | 35.0 | 27.80 | 12.4 | 39.158 |

# E    ABLATION STUDIES ON THE HYPER-PARAMETER $p$

We supplement 4 experiments on hyper-parameter $p$, including: (1) ablation studies of values , (2) token percentage statistics, (3) comparisons of sampling strategies, and (4) an ablation study on sampling size.

As Table 4 shows different $p$ influences on BioMed, different values lead to fluctuations in data performance. The Table 9 presents the distribution percentages across different probability value ranges. As mentioned above, we need to refine the data while preserving mainly source distribution. As shown in Figure 6, a larger $p$ indicates fewer tokens will be resampled, while a smaller $p$ results in more tokens being resampled. Balancing performance and the preservation of data distribution, we set $p = 0.99$ as threshold for our experiments. The Table 8 shows the results of different sampling strategies. Specifically, to control variables, we set $k = 8$ for top-k sampling and $p = 0.99$ for top-p sampling. We use reject sampling implementation in Liu et al. (2023). The results of reject sampling, top-p, and top-k are comparable. However, top-p involves a dynamic sampling range, and reject sampling requires multiple rounds of computation, leading to increased overhead. Considering computational efficiency, we chose top-k for sampling. This aligns with our original objective of maintaining minimal computational overhead. This aligns with our initial objective of minimizing computational overhead as much as possible. The Table 7 shows the ablation study on sampling size of top-k. The improvement achieved with larger values is relatively small. Therefore, we chose $k = 8$ in our experiments.

# F    DISCUSSION

## F.1    WHAT IS THE DIFFERENCE BETWEEN NON-ITERATIVE AND ITERATIVE MODEL COLLAPSE?

We define 'non-iterative model collapse' as the performance degradation caused by directly mixing general synthetic data with real data, without iterative training. Theoretically, without additional regularization constraints to guide data generation, the variance of the model-generated data gradually

---

[6] https://github.com/EleutherAI/lm-evaluation-harness
[7] https://github.com/microsoft/LMOps/tree/main/adaptllm
[8] https://github.com/EleutherAI/lm-evaluation-harness

Table 5: Comparison of human and synthetic data performance across downstream tasks in (Maini et al., 2024).

|  | TruthfulQA | LogiQA | Wino. | PIQA | ARC-E | BoolQ | OBQA | Avg |
|---|---|---|---|---|---|---|---|---|
| Human Data | 32.68 | 23.03 | 51.3 | 64.42 | 44.4 | 60.98 | 15 | 41.69 |
| 25% Synthetic Data | 27.91 | 21.37 | 50.12 | 63.93 | 43.94 | 62.29 | 15.4 | 40.71 |
| 50% Synthetic Data | 30.84 | 22.58 | 52.41 | 63.33 | 44.02 | 62.14 | 16 | 41.62 |
| 75% Synthetic Data | 29.5 | 22.65 | 49.8 | 63.44 | 44.53 | 61.56 | 17.2 | 41.24 |
| Synthetic Data | 28.89 | 22.58 | 49.72 | 63 | 46.3 | 54.53 | 16.8 | 40.26 |

Table 6: PPL evaluation results on 22 vaildation using the testing framework in (Maini et al., 2024). The PPL increases as the proportion of purely synthetic data grows.

|  | ArXiv | BookCorpus2 | Books3 | DM_Mathematics | Enron_Emails | EuroParl | FreeLaw | GitHub | Gutenberg_(PG-19) | HackerNews | NIH_ExPorter |  |
|---|---|---|---|---|---|---|---|---|---|---|---|---|
| Human Data | 22.26 | 25.39 | 22.87 | 10.84 | 23.50 | 30.73 | 12.04 | 4.15 | 16.88 | 32.54 | 23.53 | |
| 25% Synthetic Data | 21.86 | 26.32 | 23.87 | 11.05 | 24.85 | 35.02 | 12.84 | 4.35 | 17.99 | 33.80 | 23.76 | |
| 50% Synthetic Data | 22.50 | 28.01 | 25.75 | 10.84 | 26.56 | 41.99 | 14.02 | 4.67 | 19.70 | 36.12 | 24.61 | |
| 75% Synthetic Data | 24.35 | 31.19 | 28.98 | 11.81 | 30.30 | 56.32 | 16.03 | 5.30 | 22.75 | 40.44 | 26.19 | |
| Synthetic Data | 35.60 | 43.72 | 47.72 | 17.25 | 66.97 | 129.75 | 29.62 | 12.00 | 50.14 | 87.95 | 39.48 | |

|  | OpenSubtitles | OpenWebText2 | PhilPapers | Pile-CC | PubMed_Abstracts | PubMed_Central | StackExchange | Ubuntu_IRC | USPTO_Backgrounds | Wikipedia_(en) | YoutubeSubtitles | Avg |
|---|---|---|---|---|---|---|---|---|---|---|---|---|
| Human Data | 28.08 | 25.77 | 33.56 | 26.78 | 18.97 | 15.49 | 10.81 | 20.86 | 19.32 | 24.31 | 21.54 | 21.37 |
| 25% Synthetic Data | 29.25 | 26.94 | 34.63 | 27.83 | 19.55 | 15.38 | 11.03 | 22.32 | 19.58 | 25.88 | 22.63 | 22.31 |
| 50% Synthetic Data | 31.00 | 28.76 | 37.48 | 29.36 | 20.51 | 15.89 | 11.54 | 23.53 | 20.51 | 27.57 | 24.91 | 23.90 |
| 75% Synthetic Data | 34.18 | 32.04 | 42.39 | 32.17 | 22.33 | 16.92 | 12.55 | 26.54 | 22.21 | 30.68 | 28.98 | 27.03 |
| Synthetic Data | 57.83 | 53.94 | 78.18 | 54.69 | 34.82 | 23.87 | 20.47 | 51.78 | 37.24 | 46.12 | 65.49 | 49.30 |

decreases during this process. The diversity of the generated data diminishes over time, ultimately leading to the collapse of the model itself.

**From a setting perspective:** The difference between the two lies in their scope. Non-iterative model collapse is not confined to training on self-generated data, which allows it to uncover broader properties of synthetic data.

Table 7: Ablation study on sampling size $k$ for top-k.

| Sampling Size ($k$) | PubMedQA | MedMCQA | MedQA (4 options) |
|---|---|---|---|
| $k = 8$ | 64.5 | 26.13 | 24.82 |
| $k = 64$ | 63.8 | 28.14 | 27.34 |

For instance, in our experiments, we train GPT-2 on the Cosmopedia dataset in a single generation, which was generated by Mixtral-8x7B-Instruct-v0.1. In contrast, iterative model collapse focuses on training the model over multiple generations using self-generated data.

**From a property perspective:** The non-iterative model collapse emphasizes the gap between human data and general purely synthetic data, particularly regarding distributional properties and n-gram features. In contrast, the iter-

Table 8: Results of different sampling strategies.

| Sampling Strategy | PubMedQA | MedMCQA | MedQA (4 options) |
|---|---|---|---|
| Top-k | 64.5 | 26.13 | 24.82 |
| Top-p | 63.8 | 27.11 | 25.61 |
| Reject Sampling | 64.5 | 28.90 | 28.20 |

ative model collapse illustrates the iterative evolution of the model, resembling a self-play process. This process illustrates the gradual evolution of self-generated data. It does not involve an analysis of the differences in nature between self-generated and real data.

They both ultimately lead to model collapse, driven by the same underlying cause—synthetic data, though they investigate different aspects of synthetic data.

The most common setting is training a model on a mixture of human and synthetic data, where the synthetic data is not generated by the model itself, and its exact origin may be unknown. Moreover, there are already numerous popular datasets, such as UltraChat and OpenOrca, that combine synthetic and real data to improve training diversity and robustness. Therefore, studying synthetic data in the context of non-iterative model collapse is more realistic.

### F.2 WHAT IS COVERAGE COLLAPSE?

'Coverage collapse' refers to a phenomenon in which the distribution of synthetic data covers a significantly narrower range of values compared to human data, even when the data sizes are identical. For instance, as shown in Figure 3, the PPL range of synthetic data is limited to $[0, 14]$, whereas the PPL range of human data extends from $[0, 100]$. Despite this disparity, the overall coverage, represented by the area under the distribution curves, remains the same. This significant distribution gap is what we define as 'coverage collapse.'

### F.3 HOW DOES THE DSIR WORK?

DSIR (Xie et al., 2023) works by estimating importance weights for each data sample to measure its relevance to the target distribution. This involves three main steps: first, we leverage n-gram models to estimate two distributions of human and synthetic data, $q_{feat}$ and $p_{feat}$, which represent the target and raw distributions, respectively. We use them to compute the likelihood ratio for each sample. Next, we calculate the importance weight for each sample $z_i$ as $w_i = \frac{\hat{p}_{\text{feat}}(z_i)}{\hat{q}_{\text{feat}}(z_i)}$. The weight $w_i$ quantifies how well the sample aligns with the target distribution. Finally, we perform importance-weighted sampling without replacement to select examples, ensuring that the selected data is more representative of the target distribution.

Table 9: Token distribution across different probability ranges in BioMed dataset.

| Probability Range | Percentage | Token Count |
|---|---|---|
| 0.0-0.1 | 34.7% | 388,626,330 |
| 0.1-0.2 | 8.1% | 90,716,809 |
| 0.2-0.3 | 5.4% | 60,477,872 |
| 0.3-0.4 | 4.4% | 49,278,266 |
| 0.4-0.5 | 3.8% | 42,558,503 |
| 0.5-0.6 | 3.6% | 40,318,546 |
| 0.6-0.7 | 3.7% | 41,438,924 |
| 0.7-0.8 | 4.0% | 44,798,424 |
| 0.8-0.9 | 5.2% | 58,238,944 |
| 0.9-1.0 | 27.1% | 303,543,988 |

We use DSIR in our data analysis as it allows for principled and computationally efficient selection of synthetic data points that align with the target distribution. Moreover, the importance weight also reflects the alignment between the n-gram features of synthetic data and human data. Using DSIR, we can analyze the differences between synthetic and human data across n-gram feature distributions and data matching. As shown in Figure 4, it is challenging to select synthetic data that matches human data characteristics under the significant distribution difference. To obtain high-quality synthetic data, it is essential to focus on improving the data synthesis methods.

Table 10: Comparison of different synthetic data methods.

| Method | Data Type | Approach | Result |
|---|---|---|---|
| Cosmopedia (Ben Allal et al., 2024) | Pure synthetic | Using a prompt to induce data from LLMs. | Reveal non-iterative model collapse. |
| Rephrasing the Web (Maini et al., 2024) | Semi-synthetic | Using a prompt and source content to guide LLMs to reformat source content. | Improve training performance. |
| ToEdit (Ours) | Semi-synthetic | Using the distribution of source content estimated by LLMs (single forward pass) to replace tokens. | Improve training performance. |

### F.4 WHY DOES THE OBSERVED PROBABILITY DISTRIBUTION EXHIBIT FILTERING POTENTIAL?

**From the perspective of information theory,** we can analyze the filtering potential of the U-shape distribution as follows: We utilize the U-shape distribution in Figure 6 to re-sample tokens in the high-probability region, aiming to adjust the U-shaped distribution toward a uniform distribution. By doing so, we can maximize the information entropy. According to information theory, maximizing information entropy is achieved when the distribution is uniform.

**Lemma 1**: Let $X$ be a discrete random variable with $n$ possible outcomes. If the probability of each outcome is uniform, i.e., $P(x_i) = \frac{1}{n}$ for all $i \in \{1, 2, \ldots, n\}$, the Shannon entropy is maximized, given by:

$$H(X) = -\sum_{i=1}^{n} \frac{1}{n} \log \frac{1}{n} = \log n.$$

This represents the maximum uncertainty achievable, implying that the dataset carries the maximum possible information content. Thus, the uniform distribution, which assigns equal probability to all outcomes, possesses the maximum information entropy. To leverage this property, we utilize the U-shape distribution to re-sample tokens in the high-probability region, adjusting the U-shaped distribution toward a uniform distribution. By doing so, we can maximize the information entropy.

**From the perspective of language model learning,** our method emphasizes the importance of poorly learned data. Specifically, we resample easy tokens and encourage the model to focus on learning more challenging ones. Our method can enhance the learning of underrepresented data by resampling high-probability tokens.

### F.5 NON-AUTOREGRESSIVE TOKEN REPLACEMENT MAY COMPROMISE TEXT COHERENCE.

When designing data synthesis algorithms, we must balance synthesis efficiency and effectiveness, considering both autoregressive and non-autoregressive approaches. Autoregressive methods leverage the inherent capabilities of language models to generate coherent text sequentially. In contrast, non-autoregressive methods resample individual tokens based on

Table 11: Percentage of tokens requiring edits in the Natural-Instructions dataset. The total number of tokens is 4,671,834.

|  | Generation 1 (source) | Generation 2 | Generation 3 |
|---|---|---|---|
| **Tokens** ($p > 0.99$) | 584,103 | 549,519 | 517,433 |
| **Percentage** | 12.5% | 11.76% | 11.08% |

their probability distributions. Since data synthesis is a prerequisite for model training, we aim to ensure that the cost of data synthesis does not exceed the cost of training itself.

Specifically, our ToEdit modifies data using the probability distribution in a single forward pass. For instance, if the generated sequence length is 1024, the computational cost of autoregressive methods would be 1024 times higher than ours. This efficiency advantage is why our method can run effectively on GPUs like the 3090 or 4090 series.

However, this efficiency may come at the cost of coherence, as resampled tokens may not fit seamlessly into a given sentence. To address this issue, we introduce a hyperparameter, resampling probability $p$, to control the resampling threshold. We perform sampling in high-probability regions, focusing on tokens that are relatively easier to predict. We manually verify and tune on a small validation set before applying it across all experiments. In our experiments, we set $p = 0.99$.

Additionally, we supplement more experiments and discussion about hyper-parameter $p$. As Table 4 shows, different values of $p$ influence BioMed performance, leading to fluctuations in data quality. Table 9 presents the distribution percentages of the token probabilities across different value ranges. We need to refine the data while primarily preserving the source distribution. A larger $p$ indicates fewer tokens will be resampled, while a smaller $p$ results in more tokens being resampled. Balancing performance and the preservation of data distribution, we set $p = 0.99$ as the threshold for our experiments.

### F.6 Gradual Decline in Editing

We present the percentage statistics of edited tokens in Table 11, demonstrating that the edited tokens indeed exhibit a progressive decrease. Specifically, We observe that the percentage of edited tokens (above the threshold $p > 0.99$) decreases as the generation number increases. Theoretically, this is a process of distribution shifting. When tokens ($p > 0.99$) are resampled, randomness is introduced. The sampling process can select tokens with lower probabilities. Then, tokens ($p > 0.99$) is replaced, leading to a reduction of edited tokens in subsequent generations. The Table 11 provides real-world evidences for this pattern of decay.

### F.7 Comparison with Pure Synthetic Data and Reformat Methods

Specifically, both *Rephrasing the Web* (Maini et al., 2024) and our token-level editing aim to refine data while preserving the original distribution, producing semi-synthetic data. In contrast, purely synthetic data in Cosmopedia lacks the long-tail distribution and overly concentrates on n-gram features. Ultimately, semi-synthetic data enhances training performance, whereas purely synthetic data results in model collapse. Moreover, replacing a whole real sample with synthetic data can damage the performance.

The primary distinction between Cosmopedia, Rephrasing the Web (Maini et al., 2024), and our approach lies in how much of the original human data distribution is preserved. We provide a detailed comparison of these synthetic methods in Table 10.

### F.8 Must We Assume the Data is 100% Human-authored?

We do not need to assume that the data is 100% human authored; In experimental settings, some datasets used in our experiments include partially synthetic data:

- Datasets used in continual pretraining (e.g., Biomed, Finance) include partially synthetic data, which has been reformatted into a reading comprehension structure (Cheng et al., 2024b).

- OSS-Instruct-75K and Evol-Instruct-110K also contain samples synthesized by ChatGPT.

In the theoretical framework, synthetic data is generated iteratively through an $n$-generation process. (1) If the starting point is a real distribution, our method preserves most of the initial distribution to generate higher-quality data. (2) If the starting point is a mixture of synthetic and real data, the modifications are minimal, ensuring the original distribution remains largely unaffected. Therefore, applying our method in any generation $i$, we can further avoid issues, such as reduced variance and diminished diversity, which are key factors contributing to model collapse.

In other words, whether the current data is fully real or a mix of real and synthetic, using it as anchor data to synthesize data, our method builds upon the current data distribution to achieve improvements, rather than causing model collapse.

In summary, we aim to improve the data synthesis method, specifically focusing on how to obtain higher-quality data from the existing datasets. We do not need to assume that the data at hand is 100% human-generated. Our algorithm is designed to minimize excessive distribution truncation of the original data.

## G POTENTIAL APPLICATIONS AND FUTURE WORK

Based on the above discussion, our approach can be applied to optimize the current data, even if it is a mixture of real and synthetic data. From the findings and proposed method in our paper, we can influence future research in the following aspects:

**Potential applications of our work:** (1) Data optimizations. We can quickly modify and optimize the current data, using a trained language model with a single forward pass. (2) Regularization in the data synthesizing process. When synthetic data becomes excessive, we can introduce real data as an anchor to balance the issues of excessive homogeneity and tail distribution cut-off in synthetic data, thereby preventing mode collapse.

**Lessons from our work:** The key to improving the quality of synthetic data lies in balancing long-tail distribution preservation and optimizing synthetic data approaches. In other words, we should focus on two questions: how to generate more informative synthetic data and how to integrate it with real data effectively. Building on this foundation, future improvements can focus on two aspects: first, obtaining more information gain by designing more efficient generation mechanisms to inject valuable information into the synthetic data; and second, optimizing methods to reduce noise during the synthesis process. This approach ensures that synthetic data retains its authenticity while enhancing its utility in practical tasks.

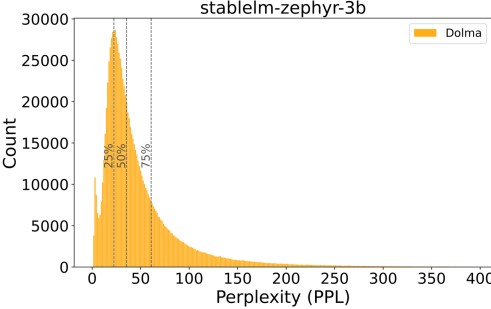
**A.** Human Data PPL Distribution Estimated by StableLM-3B

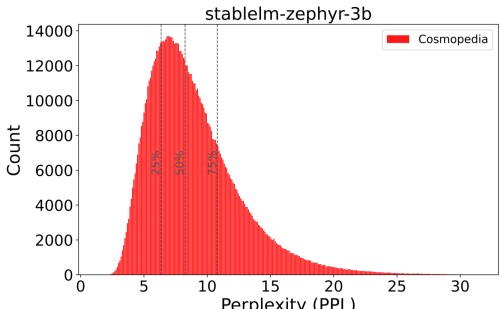
**B.** Synthetic Data PPL Distribution Estimated by StableLM-3B

Figure 9: PPL distribution of human and synthetic data estimated by StabLM-Zephyr-3B. This indicates that different prior distributions yielded the same result, which is consistent with Figure 3. The synthetic data lacks a long tail and is concentrated within a narrow portion of the distribution.

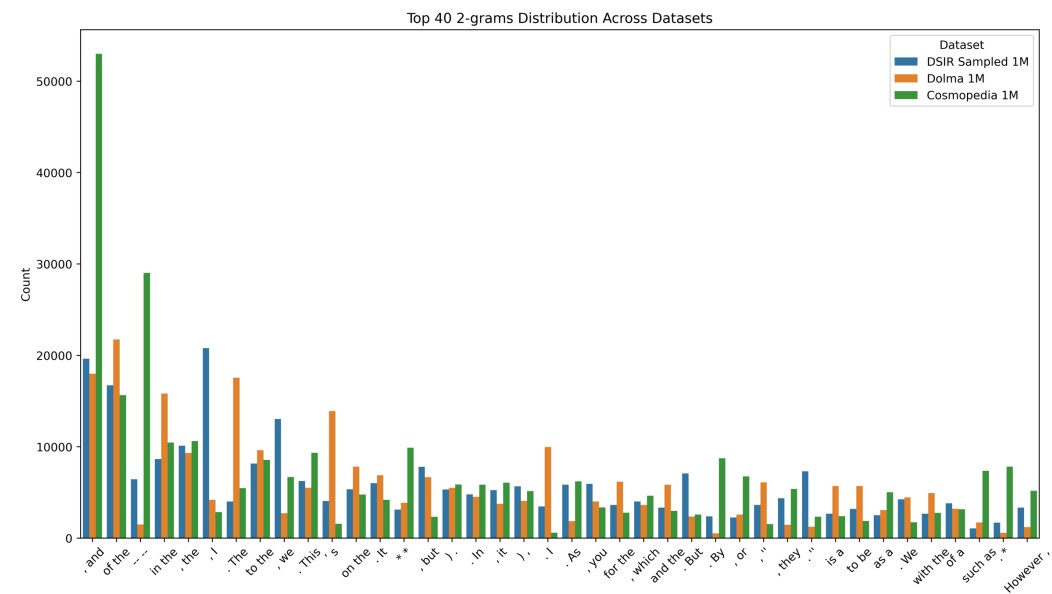

Figure 10: The top 40 bi-grams from separately sampled 1M subsets of Dolma, Cosmopedia, and DSIR-selected datasets.

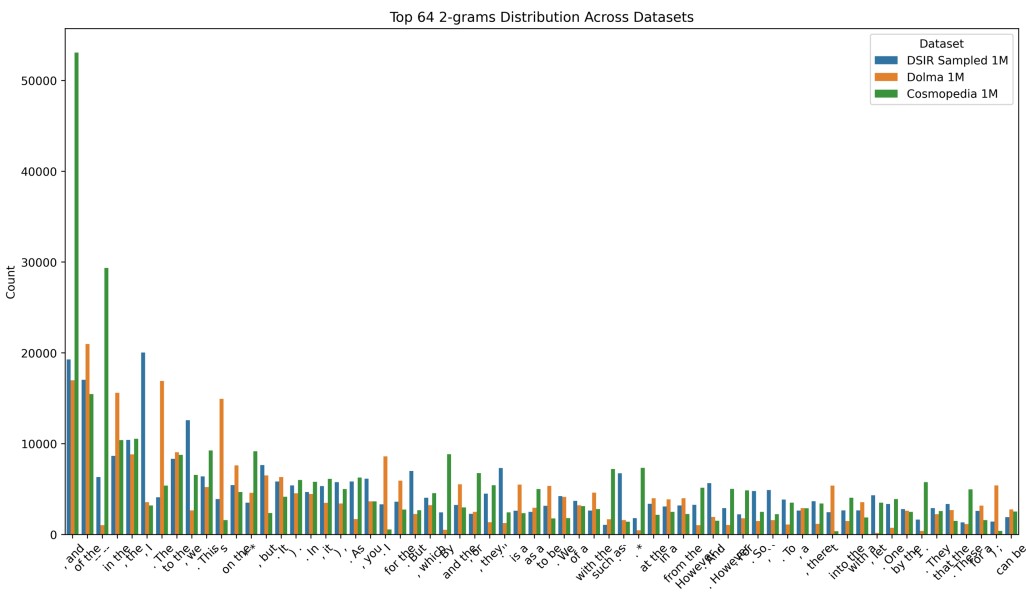

Figure 11: The top 64 bi-grams from separately sampled 1M subsets of Dolma, Cosmopedia, and DSIR-selected datasets.

Table 12: PPL results of GPT-2 124M pretraining on pure Human or Synthetic data.

| Data Type | Human Data (Dolma) | | | | | Synthetic Data (Cosmopedia) | | | | |
|---|---|---|---|---|---|---|---|---|---|---|
| Tokens Size | 8.4B | 16.8B | 25.2B | 33.6B | 42B | 8.4B | 16.8B | 25.2B | 33.6B | 42B |
| Epochs | 1 | 2 | 3 | 4 | 5 | 1 | 2 | 3 | 4 | 5 |
| Wikitext-103 | 43.62 | 38.57 | 36.11 | 34.89 | 34.55 | 169.38 | 147.73 | 135.23 | 131.78 | 128.05 |
| RedPajama | 40.18 | 35.84 | 33.97 | 32.74 | 32.34 | 116.37 | 103.25 | 99.27 | 96.81 | 96.03 |
| Falcon-RefinedWeb | 54.85 | 49.10 | 46.93 | 45.43 | 44.90 | 146.97 | 132.60 | 127.68 | 124.32 | 122.69 |
| c4-en | 45.87 | 41.00 | 39.10 | 37.95 | 37.56 | 128.25 | 114.41 | 109.73 | 107.53 | 106.55 |
| mc4-en | 61.00 | 54.44 | 52.11 | 50.38 | 49.74 | 171.44 | 153.70 | 150.28 | 145.44 | 144.99 |

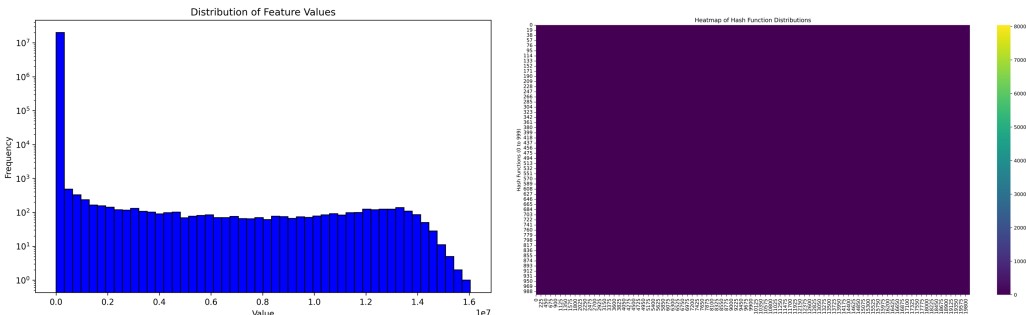

Figure 12: Density sampling response values. This result further confirms the issue of feature collapse in synthetic data.

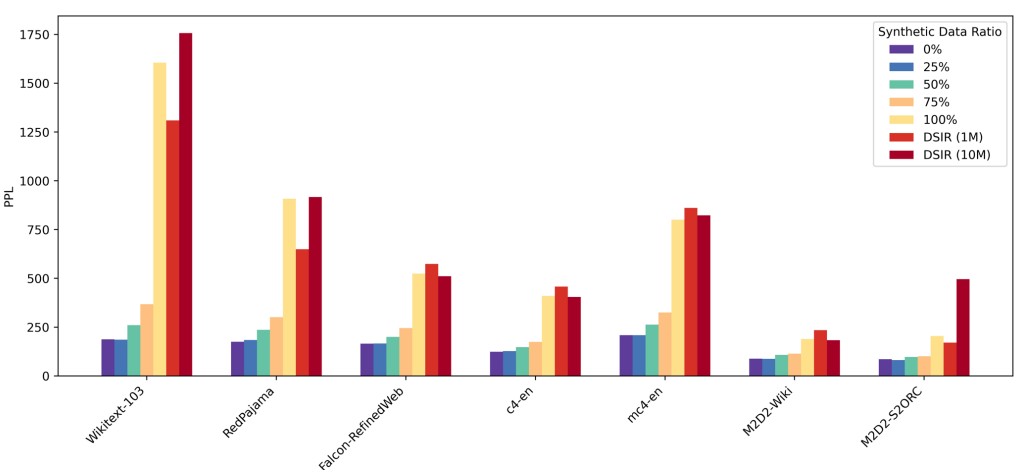

Figure 13: PPL results for OLMo-237M pretraining on selected synthetic data and data mixtures.(bar plot version for Figure 5B)

Table 13: PPL results of GPT-2 124M pretraining on mixture of human and synthetic data.

| Synthetic Data Ratio | 25% | | | | | 50% | | | | | 75% | | | | |
|---|---|---|---|---|---|---|---|---|---|---|---|---|---|---|---|
| Tokens Size | 8.4B | 16.8B | 25.2B | 33.6B | 42B | 8.4B | 16.8B | 25.2B | 33.6B | 42B | 8.4B | 16.8B | 25.2B | 33.6B | 42B |
| Epochs | 1 | 2 | 3 | 4 | 5 | 1 | 2 | 3 | 4 | 5 | 1 | 2 | 3 | 4 | 5 |
| Wikitext-103 | 45.97 | 39.87 | 37.65 | 36.91 | 36.32 | 50.29 | 43.15 | 40.46 | 39.43 | 38.65 | 58.66 | 48.75 | 45.20 | 43.42 | 42.95 |
| RedPajama | 42.28 | 37.62 | 35.72 | 34.66 | 34.24 | 46.89 | 41.42 | 39.37 | 38.21 | 37.72 | 55.72 | 49.26 | 46.27 | 44.81 | 44.30 |
| Falcon-RefinedWeb | 56.40 | 50.62 | 48.26 | 47.13 | 46.66 | 61.06 | 54.34 | 51.72 | 50.39 | 49.87 | 69.32 | 61.50 | 58.28 | 56.77 | 56.19 |
| c4-en | 48.15 | 43.14 | 40.98 | 39.91 | 39.41 | 51.79 | 46.06 | 43.90 | 42.73 | 42.23 | 58.60 | 52.22 | 49.26 | 47.87 | 47.27 |
| mc4-en | 62.46 | 56.80 | 54.35 | 53.06 | 52.71 | 70.43 | 62.48 | 59.61 | 57.66 | 57.07 | 80.37 | 71.77 | 67.90 | 65.31 | 64.82 |

Table 14: PPL results of OLMo-237M pretraining on mixture of human and synthetic data.

| Synthetic Data Ratio | 0% | 25% | 50% | 75% | 100% | DSIR (1M) | DSIR (10M) | Edu Classifier (1M) | Edu Classifier (10M) | PPL Filter (1M) | PPL Filter (10M) | Density Sampling (1M) | Density Sampling (10M) |
|---|---|---|---|---|---|---|---|---|---|---|---|---|---|
| Unique Tokens | 8.4B | 8.4B | 8.4B | 8.4B | 8.4B | 0.6B | 8.4B | 0.75B | 7.4B | 0.97B | 9B | 0.6B | 7.1B |
| Training Tokens | 8.4B | 8.4B | 8.4B | 8.4B | 8.4B | 8.4B | 8.4B | 10.5B | 7.4B | 13.68B | 9B | 8.9B | 7.1B |
| Epochs | 1 | 1 | 1 | 1 | 1 | 14 | 1 | 14 | 1 | 14 | 1 | 14 | 1 |
| Wikitext-103 | 187.36 | 185.5 | 260.08 | 367.46 | 1605.73 | 1309.53 | 1757.03 | 1111.29 | 1612.95 | 738.36 | 1193.25 | 1188.40 | 1753.89 |
| RedPajama | 175.38 | 183.93 | 236.33 | 301.09 | 907.91 | 649.36 | 916.51 | 811.14 | 1104.75 | 376.36 | 645.82 | 789.67 | 896.18 |
| Falcon-RefinedWeb | 165.17 | 166.69 | 199.68 | 245.15 | 523.93 | 573.61 | 510.96 | 522.97 | 612.72 | 344.82 | 449.86 | 501.99 | 560.92 |
| c4-en | 123.88 | 127.68 | 147.69 | 174.48 | 410.19 | 457.96 | 404.63 | 415.88 | 487.97 | 286.95 | 367.44 | 414.55 | 457.71 |
| mc4-en | 208.91 | 208.94 | 263.35 | 324.91 | 800.40 | 861.01 | 823.12 | 769.86 | 955.70 | 476.81 | 662.00 | 740.75 | 844.53 |
| M2D2-Wiki | 88.24 | 87.34 | 107.77 | 114.19 | 189.06 | 234.45 | 183.17 | 161.58 | 206.45 | 130.43 | 162.08 | 167.20 | 205.50 |
| M2D2-S2ORC | 86.15 | 81.53 | 97.61 | 100.64 | 204.22 | 170.78 | 496.40 | 145.27 | 201.52 | 117.44 | 163.38 | 131.22 | 192.97 |

Table 15: Dolma dataset statistics (v1.6), quoted from source (Soldaini et al., 2024).

| Source | Doc Type | UTF-8 bytes (GB) | Documents (millions) | Unicode words (billions) | Llama tokens (billions) |
|---|---|---|---|---|---|
| Common Crawl | web pages | 9,022 | 3,370 | 1,775 | 2,281 |
| The Stack | code | 1,043 | 210 | 260 | 411 |
| C4 | web pages | 790 | 364 | 153 | 198 |
| Reddit | social media | 339 | 377 | 72 | 89 |
| PeS2o | STEM papers | 268 | 38.8 | 50 | 70 |
| Project Gutenberg | books | 20.4 | 0.056 | 4.0 | 6.0 |
| Wikipedia, Wikibooks | encyclopedic | 16.2 | 6.2 | 3.7 | 4.3 |
| **Total** | | **11,519** | **4,367** | **2,318** | **3,059** |

