# OpenReview forum: "ToEdit: How to Synthesize Text Data to Avoid Model Collapse?"
_ICLR.cc/2025/Conference — Submitted to ICLR 2025_

### Official Review · Reviewer_1Bd4 · 2024-10-30

**Soundness:** 3
**Presentation:** 2
**Contribution:** 2
**Rating:** 6
**Confidence:** 4

**Summary:**

The paper investigates the problem of model collapse when training with synthetic data, and provides theoretical and empirical evaluations for training under different mixtures of human and synthetic data. They introduce token-level editing, which is a straightforward technique to introduce noise to human-written data by resampling a token every time a prior LLM probability assigned to the next token is larger than p=0.99. Authors show that using token-level editing in pretraining/continual pretraining/supervised finetuning they can achieve comparable performance to using standard human-authored data across a number of downstream tasks.

**Strengths:**

- Interesting setup with both theoretical and empirical evaluation.
- Empirical evaluation is extensive, covering many downstream tasks and (pretraining / continual pretraining) settings.
- Token-level editing works well in practice for continual pretraining setting, and does not damage learning for pretraining and SFT settings.

**Weaknesses:**

- Overall, although results with continual pretraining seem good, I would add way more nuance to results with pretraining and SFT. Differences in results obtained are very narrow and oftentimes the proposed method is numerically worse (by a very small margin).
- I find that some important points are not well addressed, for instance how did authors choose the p=0.99 as their threshold for resampling with token-level editing and what happens if one varies this threshold. This is especially important in light of the positive-to-mixed results (in the reviewer's opinion) obtained in the experiments.

**Questions:**

- In Figure 1, you introduce a lot of notation that you do not explain neither in a legend nor in the caption, like $\sigma$, $d$, $T$, $f_0$, $m_1$, $y_1$, $m_0$, $y_0$, etc. Please introduce all these terms in the Figure caption (or in a legend in the figure itself). If that will make things too complicated, simplify the figure by removing the notation.
- In lines 89-92, you briefly describe iterative and non-iterative model collapse. I think this needs more clear language / more detail. What are the key differences between the two? Where are they similar? Why is one more realistic than the other and why? Please focus on that.
- One stark difference in Fig.2 you do not discuss is a bimodal distribution of ppl on human data. Or is that an artifact from the binning strategy you were using (if any)?
- Not sure I fully agree with the discussion around Fig.2 (lines 190-198). I find the plots for 1M human texts and 1M synthetic texts very similar. Why did you choose to use Figures to motivate your work instead of using more objective metrics, like those that measure the similarity between probability distributions?
- I do not think that a line plot is the best setting for Fig.5B, you would probably be better off with a bar plot here. The lines hint at the wrong intuition that they have meaning, whereas only the dots have meaning.
- Applying token-level editing by using top-k sampling with an LLM as the prior when p>0.99 means you discard the tokens where the LLM is too certain. One important question here is how did you choose this number of 0.99, and what would happen if this number were different? How do we generalise any findings (or can propose any future work) based on your results?
- In line 211, you may want to explain how the DSIR sampling works and why you use it in your data analysis.
- You may have written this somewhere, but what percentage of the original tokens are edited (on average per dataset) when you use token-level editing with p=0.99?

Typos worth mentioning:
- Line 211: "important sampling"

---

> ### Author Response · Authors · 2024-11-21
> **Response - 1**
>
> We are grateful for your enthusiastic feedback and insightful comments. Below, we give detailed responses to your questions.
>
> > [Q1] : Add way more nuance to results with pretraining and SFT.
>
> **To further validate the effectiveness of our method, we incorporate more 3 general instruction tuning tasks and 2 code reasoning-related SFT tasks.** As for general instruction tuning tasks, we adopt instruction tuning datasets from [2], including CoT [3], FLAN V2 [4], and Open Assistant 1 [5]. As for code-related reasoning tasks, we utilize OSS-Instruct-75K and Evol-Instruct-110K.
>
> The experimental results demonstrate that the ∆ToEdit consistently improves performance across both general instruction tuning and code-related SFT tasks. For code-related tasks, the improvements are particularly evident in ARC-Challenge and GPQA, indicating better reasoning and code comprehension. From the trends observed, our method demonstrates partial improvements across multiple datasets, indirectly validating its effectiveness.
>
> Regarding pretraining, due to the significant time and storage cost of downloading large-scale data and performing pretraining from scratch, related experiments are still being supplemented.
>
> 1. General Instruction tuning.
>
> |                       | PIQA  | BoolQ | HellaSwag | SIQA  | Winogrande | Avg       |
> | --------------------- | ----- | ----- | --------- | ----- | ---------- | --------- |
> | CoT [3]               | 79.87 | 81.28 | 59.72     | 49.69 | 74.51      | 69.014    |
> | $\Delta ToEdit$       | 80.25 | 81.16 | 59.74     | 50.56 | 74.59      | **69.26** |
> |                       |       |       |           |       |            |           |
> | FLAN V2 [4]           | 80.79 | 84.04 | 59.98     | 51.43 | 74.66      | 70.18     |
> | $\Delta ToEdit$       | 80.69 | 85.20 | 59.99     | 52.00 | 75.37      | **70.65** |
> |                       |       |       |           |       |            |           |
> | Open Assistant 1  [5] | 79.65 | 83.18 | 60.51     | 48.52 | 74.11      | 69.194    |
> | $\Delta ToEdit$       | 79.98 | 83.91 | 60.34     | 48.31 | 74.66      | **69.44** |
>
> 2. Code SFT tasks; OSS-Instruct-75K and Evol-Instruct-110K are code-related supervised tasks, containing python, cpp, java and php languages.
>
> |                    | ARC-Challenge | GPQA  | GSM8K | MMLU  | Avg         |
> | ------------------ | ------------- | ----- | ----- | ----- | ----------- |
> | OSS-Instruct-75K   | 51.28         | 27.46 | 49.58 | 62.14 | 45.76       |
> | $\Delta ToEdit$    | 51.79         | 28.79 | 49.36 | 62.04 | **46.1328** |
> |                    |               |       |       |       |             |
> | Evol-Instruct-110K | 52.9          | 27.9  | 50.87 | 62.4  | 46.616      |
> | $\Delta ToEdit$    | 52.22         | 29.69 | 50.87 | 62.6  | **46.9216** |

---

> ### Author Response · Authors · 2024-11-21
> **Response - 2**
>
> > [Q2] :  how did authors choose the p=0.99 as their threshold for resampling with token-level editing and what happens if one varies this threshold.
>
> `how did authors choose the p=0.99 `
>
> **In our experiments, the value of $p$ was selected through a comparative analysis of hyperparameters.** As the table below shows the various performance of different $p$, we selected hyperparameter $p$ by conducting experiments on the validation set of BioMed.  As discussed in our paper, an effective synthetic dataset generation method needs to balance real data distribution and modification extent. By weighing the performance and the proportion of tokens involved, we ultimately selected a value of 0.99.
>
> | Criteria                        | PubMedqa | MQP   | RCT   | USMLE | ChemProt | Avg    |
> | :------------------------------ | -------- | ----- | ----- | ----- | -------- | ------ |
> | Resampled Tokens $p \geq 0.99$  | 64.5     | 55.73 | 30.95 | 27.65 | 14.6     | 38.686 |
> | Resampled Tokens $p \geq 0.999$ | 63.6     | 55.4  | 29.09 | 28.12 | 16.2     | 38.482 |
> | Resampled Tokens $p \leq 0.1$   | 62.4     | 51.47 | 25.6  | 29.14 | 10.0     | 35.722 |
> | Resampled Tokens $p \leq 0.01$  | 65.4     | 54.91 | 28.19 | 27.80 | 11.0     | 37.46  |
> | Resampled Tokens $p \leq 0.001$ | 64.2     | 56.39 | 35.0  | 27.80 | 12.4     | 39.158 |
>
> `what happens if one varies this threshold`
>
>  **A larger $p$ indicates fewer tokens will be resampled, while a smaller $p$ results in more tokens being resampled.**  Only moderate $p$ yield optimal results, while excessive changes degrade performance, i.e., model collapse. We present example statistics of the BioMed dataset, detailing the number of tokens and their respective percentages, as follows.
>
> | Probability Range | 0.0-0.1     | 0.1-0.2    | 0.2-0.3    | 0.3-0.4    | 0.4-0.5    | 0.5-0.6    | 0.6-0.7    | 0.7-0.8    | 0.8-0.9    | 0.9-1.0     |
> | ----------------- | ----------- | ---------- | ---------- | ---------- | ---------- | ---------- | ---------- | ---------- | ---------- | ----------- |
> | Percentage        | 34.7%       | 8.1%       | 5.4%       | 4.4%       | 3.8%       | 3.6%       | 3.7%       | 4.0%       | 5.2%       | 27.1%       |
> | Token Count       | 388,626,330 | 90,716,809 | 60,477,872 | 49,278,266 | 42,558,503 | 40,318,546 | 41,438,924 | 44,798,424 | 58,238,944 | 303,543,988 |
>
> We further supplement more experiments with more detailed ablation on hyperparameters. **The following tables present experiments on different sampling strategies and sampling sizes.** From these experiments, we ultimately selected top-k as the sampling strategy, considering both the efficiency and effectiveness of the sampling process.
>
> | Sampling Strategy | PubMedqa | medmcqa | medqa_4options |
> | ----------------- | -------- | ------- | -------------- |
> | Top-k             | 64.5     | 26.13   | 24.82          |
> | Top-p             | 63.8     | 27.11   | 25.61          |
> | Reject Sampling   | 64.5     | 28.9    | 28.2           |
>
> | Sampling Strategy | PubMedqa | medmcqa | medqa_4options |
> | ----------------- | -------- | ------- | -------------- |
> | $k=8$             | 64.5     | 26.13   | 24.82          |
> | $k=64$            | 63.8     | 28.14   | 27.34          |

---

> ### Author Response · Authors · 2024-11-21
> **Response - 3**
>
> > [Q3] : Please introduce all these terms in the Figure caption
>
> Thank you very much for pointing this out. We have updated the following points in the figure caption.
>
> The data $(x_{0},y_{0}) \sim P_{\Sigma,w,\sigma^2}$ on $\mathbb{R}^d \times \mathbb{R}$, $f_0$ is first generation trained model, and $y_1$ is synthetic data, $m_1$ is matrix to specify which need to edit. $T$ and $d$ denote the number of data points and the dimensionality, respectively. Further detailed definitions and derivations can be found in Sec. 3.
>
> > [Q4] : What are the key differences between the iterative and non-iterative model collapse? Where are they similar? Why is one more realistic than the other and why? Please focus on that.
>
> `What are the key differences ?`
>
> - From a setting perspective: the difference between the two lies in their scope: **non-iterative model collapse is not confined to training on self-generated data,** enabling it to uncover broader properties of synthetic data. For instance, in our paper, we train GPT-2 using the Cosmopedia dataset in a single generation, which was generated by Mixtral-8x7B-Instruct-v0.1. In contrast, iterative model collapse requires training on self-generated data over multiple generations.
> - From a property perspective: **the non-iterative model collapse highlights the gap between human data and general purely synthetic data**, particularly in terms of distributional properties and n-gram features. The iterative model collapse illustrates the iterative evolution of the model, resembling a self-play process. This process illustrates the gradual evolution of self-generated data without involving an analysis of the differences in nature between self-generated and real data [1,2,3].
>
> `Where are they similar? ` They both ultimately lead to model collapse, driven by the same cause—synthetic data, while they investigate different aspects of synthetic data.
>
> `Why is one more realistic than the other ?`  In common model training scenarios, we do not deliberately use self-generated data to train the model itself. The most common setting is training model on a mixture of human and synthetic data, where the synthetic data is not generated by the model itself, even we don't know where this synthetic data comes from. Moreover, numerous popular datasets, such as UltraChat [9] and OpenOrca [10], have already been used to combine synthetic and real data for training. Therefore, studying synthetic data under the non-iterative model collapse setting is more realistic.
>
> We further elaborate on this issue in the revision.

---

> ### Author Response · Authors · 2024-11-21
> **Response - 4**
>
> > [Q5] : One stark difference in Fig.2 you do not discuss is a bimodal distribution of ppl on human data. Or is that an artifact from the binning strategy you were using (if any)?
>
> Thank you for pointing this out. **The bimodal distribution of PPL on human data is likely a property of the underlying data distribution rather than an artifact of the binning strategy.** We tested this behavior using the other model and observed a similar distribution (Figure 9), which suggests it is inherent to the dataset itself. On the other hand, this phenomenon does not impact our main conclusions, as our findings remain consistent across different models and settings.
>
> > [Q6] Why did you choose to use Figures to motivate your work instead of using more objective metrics, like those that measure the similarity between probability distributions ?
>
> Visualizing data statistics offers a direct and intuitive way to highlight the differences between human and synthetic data. We think this can help readers quickly understand our key findings and research questions.
>
> `Not sure I fully agree with the discussion around Fig.2 (lines 190-198). I find the plots for 1M human texts and 1M synthetic texts very similar. `
>
> I am assuming you are referring to Fig. 4 (lines 190-198)?
>
> If so, the bottom plot in Fig.4 shows the differences in the n-grams distribution between huamn and synthetic data. If the values in the bottom plot approach zero, the two distributions are nearly identical. However, across 10,000 hash buckets, the differences in log probabilities fluctuate around $y=0$, indicating that the human texts and synthetic texts are not similar.
>
> If you have further questions, we will provide more clarification.
>
> > [Q7] : I do not think that a line plot is the best setting for Fig.5B, you would probably be better off with a bar plot here. The lines hint at the wrong intuition that they have meaning, whereas only the dots have meaning.
>
> Thank you for your advice. We have updated a bar plot for Fig.5B in our revision.

---

> ### Author Response · Authors · 2024-11-21
> **Response - 5**
>
> > [Q8] : One important question here is how did you choose this number of 0.99, and what would happen if this number were different? How do we generalise any findings (or can propose any future work) based on your results?
>
> **As we mentioned in [Q1], we conduct a series of experiments on the validation set, then select the suitable hyper-parameter $p$.**   A larger $p$ indicates fewer tokens will be resampled, while a smaller $p$ results in more tokens being resampled. As stated in line 238-244,  we need to refine the data while preserving the distribution of the real data. Based on Figure 6 and Supplementary Tables in [Q1], we select the optimal  hyperparameter using the validation set and apply it consistently across all experiments.
>
> ` propose any future work based on your results`   The discussion in our paper and above highlights: the key to improving the quality of synthetic data lies in balancing long-tail distribution preservation and optimizing the synthetic data mix ratio. **In other words, we should focus on how to generate more informative synthetic data and how to integrate it with real data for better performance.** Building on this foundation, future improvements can focus on two aspects: first, obtaining more information gain, designing more efficient generation mechanisms to inject valuable information into the synthetic data; second, optimizing methods to reduce noise during the synthesis process. This approach ensures that synthetic data retains its authenticity while enhancing its utility in practical tasks.
>
> > [Q9] : In line 211, you may want to explain how the DSIR sampling works and why you use it in your data analysis.
>
> **DSIR [11] works by estimating importance weights for each data sample to measure its relevance to the target distribution.** This involves three main steps: first, we leverage n-gram models to fit two distributions of human and synthetic data, $q_{feat}$ and $p_{feat}$, which represent the target and raw distributions, respectively. We use them to compute the likelihood ratio for each sample. Next, we calculate the importance weight for each sample $w_i = \frac{p_{feat}(z_i)}{q_{feat}(z_i)}$, which quantifies how well the sample aligns with the target distribution.  Finally, we perform importance-weighted sampling without replacement to select examples.
>
> We use DSIR in our data analysis as it allows for principled and computationally efficient selection of synthetic data points that align with the target distribution. And the importance weight also reflects the alignment between the n-gram features of synthetic data and human data. Using DSIR, we can analyze the differences between synthetic and human data across n-gram feature distributions and data matching. As shown in Fig.5, It is challenging to select synthetic data that matches human data characteristics under the significant distribution difference. To obtain high-quality synthetic data, it is essential to focus on improving the data synthesis method.

---

> ### Author Response · Authors · 2024-11-21
> **Response - 6**
>
> > [Q10] : what percentage of the original tokens are edited (on average per dataset) when you use token-level editing with p=0.99?
>
> We present the token probability statistics as percentages below. As we mentioned above, a larger $p$ indicates fewer tokens will be resampled, while a smaller $p$ results in more tokens being resampled. **Specifically, for $p=0.99$ at generation 1, 12.5% tokens are resampled.**
>
> |                   | Generation 1 (source) | Generation 2 | Generation 3 |
> | ----------------- | --------------------- | ------------ | ------------ |
> | Tokens ($p>0.99$) | 584,103               | 549,519      | 517,433      |
> | Percentage        | 12.5%                 | 11.76%       | 11.08%       |
>
> | Probability Range | 0.0-0.1     | 0.1-0.2    | 0.2-0.3    | 0.3-0.4    | 0.4-0.5    | 0.5-0.6    | 0.6-0.7    | 0.7-0.8    | 0.8-0.9    | 0.9-1.0     |
> | ----------------- | ----------- | ---------- | ---------- | ---------- | ---------- | ---------- | ---------- | ---------- | ---------- | ----------- |
> | Percentage        | 34.7%       | 8.1%       | 5.4%       | 4.4%       | 3.8%       | 3.6%       | 3.7%       | 4.0%       | 5.2%       | 27.1%       |
> | Token Count       | 388,626,330 | 90,716,809 | 60,477,872 | 49,278,266 | 42,558,503 | 40,318,546 | 41,438,924 | 44,798,424 | 58,238,944 | 303,543,988 |

---

> > ### Comment · Reviewer_1Bd4 · 2024-11-22
> >
> > I thank the authors for their rebuttal. I have updated my scores from 3 to 5 (marginally below the acceptance threshold), but decreased my score to presentation from 3 to 2. Although technically speaking the authors' experimental setup is comprehensive, I still see many points for improvement in its presentation, and struggle to see how it would impact future work.
> >
> > - There are now a number of typos in the paper. E.g., line 1106 "From perporty perspective", or line 1092 "WHAT IS DIFFERENCE BETWEEN NON-ITERIVE AND ITERVE MODEL COLLAPSE ?"
> > - In Figure 1, there are still many things to be improved. I suggest to the authors that they show that figure to colleagues who do not work on the topic (just the figure in isolation, without the accompanying paper) and ask their colleagues to explain it. This way you will see what parts are unclear and still need improvement. For example, there is $x_0, y_0$ being sampled from a very cryptic $P_{\Sigma,w,\sigma^2}$ on $\mathbb{R}^d × \mathbb{R}$. I have no idea what these variables are, and if you will not explain them, you should not include them in the Figure nor in its caption. What is $\Sigma$, $\sigma^2$, $w$, etc? Moreover, now you write in Figure 1's caption that "$y_1$ is synthetic data". Does that mean that $y_0$ is also synthetic data? If that is the case, you should clearly state that "$X$ denotes data authored by humans, whereas $Y$ data generated by a generative model", and possibly something like "the subscript $i$ denote the training iteration".
> > - Finally, I understand the importance of the findings discussed in the paper. I just do not think authors succeed in linking these findings to actionable insights that can guide researchers in their future work, which to me would make this a paper with impact. Please correct me if I am wrong, I will try to expand on this a bit further.
> >     - With the method proposed in this paper, we must assume we have data which is 100\% human authored, and we can add noise to (on average) 27.1\% of that data to generate a noisy version of that data (according to threshold $p$ selected). Authors clearly show that this noisy---or synthetic, which is the term used in the paper---data does not lead to model collapse, and can even improve performance across tasks (although many times by a small margin, and rarely also decrease performance considerably e.g. Table 1, MMLU).
> >     - However, this assumption makes the findings in this paper lead to no actionable insights, in the reviewer's opinion. The corpora we have, as the authors stated, are often a mix or human and synthetic data. If we had a mechanism that tells us which parts of some corpus are synthetic and which are human authored, we could use that to plug the method the authors propose in the "real world". However, without such a mechanism, we now know it is *possible* to have synthetic data (using the token-level editing proposed) that does not lead to model collapse, but we have no way to tell if the "synthetic data" we concretely have in an existing corpus is "good" or "bad".

---

> ### Author Response · Authors · 2024-11-22
>
> We greatly appreciate your further questions and feedback, your feedback is highly important to us, and we will do our best to address your concerns below.
>
> > [Q1] Typos
>
> Thank you for pointing out the typos. We have carefully reviewed the entire paper and corrected typos in the revision.
>
> > [Q2] In Figure 1, there are still many things to be improved.
>
> Based on your suggestions, we have made the following updates:
>
> - We have annotated all the processes and symbols at the bottom of the Figure 1 to facilitate quick reference for readers.
>   - Source Real Data: $(x_0, y_0)$
>   - Synthetic Data: $(y_1, y_2, \dots, y_n)$
>   - Number of Iterations: $i \in \{1, \dots, n\}$
>   - Test Error: $E_{\text{test}}$
>   - Trained Model: $f_i$
>   - Input Dimensions: $d$
>   - Data Size: $T$
>   - Label Noise Scalar: $\sigma$
>   - Editing Operation Matrix: $m_i$
> - We have revised the caption to quickly explain the figure in conjunction with the symbols.
>   - "Specifically, starting from real data $(x_o, y_o)$, the test error $E_{test}$ increases as $f_0$ is iteratively trained on synthetic data $(y_1, y_2, \dots, y_n)$. Our method, ToEdit, utilizes $f_0$ and an operation matrix $m_i$ to edit the data, achieving a fixed upper bound."
>
> > [Q3] I just do not think authors succeed in linking these findings to actionable insights that can guide researchers in their future work, which to me would make this a paper with impact.
>
> `we must assume we have data which is 100% human authored,`
>
> **We do not need to assume that the data is 100% human authored.**
>
> **In experiments, some datasets used in our experiments include partially synthetic data.**
>
> - Datasets used in continual pretraining (e.g., Biomed, Finance) include partially synthetic data, which has been reformatted into a reading comprehension structure [15].
>
> - OSS-Instruct-75K and Evol-Instruct-110K also contain samples synthesized by ChatGPT.
>
> In the theoretical framework, synthetic data is generated iteratively through an $n$-generation process.  (1) If the starting point is a real distribution, our method preserves most of the initial distribution to generate higher-quality data. (2) If the starting point is a mixture of synthetic and real data, the modifications are minimal, ensuring the original distribution remains largely unaffected. Therefore, applying our method in any generation $i$, we can further avoid issues, such as reduced variance and diminished diversity,  which are key factors contributing to model collapse.
>
> **In other words, whether the current data is fully real or a mix of real and synthetic, using it as an anchor to synthesize data, our method builds upon the current data distribution to achieve improvements, rather than causing model collapse.**
>
> In summary, we aim to improve the data synthesis method, specifically focusing on how to obtain higher-quality data from the existing datasets. We do not need to assume that the data at hand is 100% human-generated. Our algorithm is designed to minimize excessive distribution truncation of the original data.
>
> [15] Cheng D, Huang S, Wei F. Adapting large language models via reading comprehension[C]//The Twelfth International Conference on Learning Representations. 2023.
>
> `and struggle to see how it would impact future work.`
>
> Based on the above discussion, our approach needn't assume the original data is 100% human-authored. In contrast, our approach can be applied to optimize the current data, even if it is a mixture of real and synthetic data. From the findings and proposed method in our paper, we can influence future research in the following aspects:
>
> - **Potential applications of our work:**  (1) Data optimization. We can quickly modify and optimize the current data, using a trained language model with a single forward pass. (2) Regularization in data synthesizing process. When synthetic data becomes excessive, we can introduce real data as an anchor to balance the issues of excessive homogeneity and tail distribution cut-off in synthetic data, thereby preventing mode collapse.
> - **Lessons from our work:** The key to improving the quality of synthetic data lies in balancing long-tail distribution preservation and optimizing synthetic data approaches. In other words, we should focus on two questions: how to generate more informative synthetic data and how to effectively integrate it with real data. Building on this foundation, future improvements can focus on two aspects: first, obtaining more information gain by designing more efficient generation mechanisms to inject valuable information into the synthetic data; and second, optimizing methods to reduce noise during the synthetic process. This approach ensures that synthetic data retains its authenticity while enhancing its utility in practical tasks.

---

> > ### Comment · Reviewer_1Bd4 · 2024-11-28
> >
> > Dear authors, thank you very much for all the efforts and the extra experiments. I am increasing my marks (soundness from 2 to 3, recommendation from 5 marginally below to 6 marginally above the acceptance threshold).

---

> > > ### Author Response · Authors · 2024-11-28
> > >
> > > We truly appreciate your recognition and support! Thank you once again for the time and diligence you've dedicated to reviewing our paper.

---

### Official Review · Reviewer_PCAs · 2024-11-03

**Soundness:** 3
**Presentation:** 4
**Contribution:** 3
**Rating:** 8
**Confidence:** 3

**Summary:**

The paper explores the impact of synthetic data on language model training, particularly the phenomenon of model collapse. Model collapse occurs when models, trained extensively on synthetic data, experience performance degradation due to distributional shifts,  lack of full distributional range of human language, and over-concentrates certain linguistic features, leading to "coverage collapse." The authors propose a novel solution called Token-Level Editing  to counteract model collapse by editing human data at the token level, creating "semi-synthetic" data that retains critical human data distribution characteristics. They demonstrate, both theoretically and experimentally, that this method helps prevent model collapse by bounding test error within a finite range. Through extensive pretraining, continual pretraining, and supervised fine-tuning experiments, the results indicate that token-level editing improves the quality of synthetic data and enhances model performance across tasks.

**Strengths:**

1. The paper is well written and consistently maintains the readers interest by presenting core findings around synthetic data issues and the proposed token editing approach initially and later go on to support those claims.

2. The paper presents a novel strategy for avoiding model collapse, a critical issue as human data becomes scarce and reliance on synthetic data grows. The proposed Token-Level Editing (ToEdit) approach offers a novel way to mitigate collapse by creating semi-synthetic data, modifying only certain tokens in human data instead of relying entirely on synthetic data. This targeted editing approach preserves key distributional characteristics, reducing the risk of collapse while leveraging the benefits of synthetic data.

3. The authors support their method with a robust theoretical framework, calculating bounded test error as evidence that model performance degradation can be prevented.

4. Empirical results are provided with experiments involving pretraining, continual pretraining, and fine-tuning stages, showing that token-level editing yields consistent performance improvements across diverse learning approaches and domains, such as biomedicine, finance, and mathematics.

**Weaknesses:**

1. Token-Level Editing relies heavily on a pre-trained language model to estimate token probabilities, which may introduce biases based on the pre-existing model's characteristics. Did the authors try an ensemble of models with varied pretraining sources could be used to reduce single-model bias in the token-level editing process. This could enhance the diversity of the generated semi-synthetic data.

2. Although the study focuses on text, with recent LLMs heavily focused on solving tasks such as code generation,  I wonder if ToEdit would work on other data types (e.g., code, tabular data)? Changing tokens in code sequences might make the code prone to erros. It will be interesting to see experiments on the generalizability of ToEdit across different data domains and will help the community understand its broader applicability.

3. The experiments are limited to GPT-2 and OLMo models, which may not reflect the broader applicability of Token-Level Editing across diverse architectures. Testing on models like BERT (with bidirectional attention), T5 (with encoder-decoder architecture), and maybe more recent LLMs with sparse attention or mixture-of-experts layers could reveal whether ToEdit’s effectiveness is architecture-agnostic or if it needs adaptation for specific configurations.

**Questions:**

Questions & Suggestions:
1. It would help if the paper provided further clarification on the specific threshold values used for token-level editing (e.g., the probability threshold $p=0.99$. Explaining why this threshold was chosen and whether it varies by dataset or model could add depth to the methodology section.

2. Although common, it might still be useful to expand that PPL stands for perplexity. It will be helpful for new researchers.

3. The paper uses terms like "non-iterative model collapse" and "coverage collapse." It would be helpful broader audience to better understand these terms if it was clearly defined.

---

> ### Author Response · Authors · 2024-11-21
> **Response - 1**
>
> We sincerely appreciate your positive comments and support for our acceptance. In the following, we will address your concerns and refine our paper accordingly.
>
> > [Q1] ToEdit would work on other data types (e.g., code, tabular data)?
>
> **To further validate the effectiveness of our method, we incorporated 2 code reasoning-related SFT tasks and 3 general instruction tuning tasks.** As for general instruction tuning tasks, we adopt instruction tuning datasets from [7], including CoT [3], FLAN V2 [4], and Open Assistant 1 [5]. As for code-related reasoning tasks, we utilize OSS-Instruct-75K and Evol-Instruct-110K.
>
> **The experimental results demonstrate that the ∆ToEdit consistently improves performance across both general instruction tuning and code-related SFT tasks.** In general instruction tuning, ∆ToEdit enhances average performance across CoT, FLAN V2, and Open Assistant, with notable gains in BoolQ, SIQA, and Winogrande. For code-related tasks, the improvements are particularly evident in ARC-Challenge and GPQA, indicating better reasoning and code comprehension. These findings validate the effectiveness of the proposed modifications in enhancing both generalization and task-specific performance.
>
> 1.Code SFT tasks; OSS-Instruct-75K and Evol-Instruct-110K are code-related supervised tasks, containing python, cpp, java and php languages.
>
> |                    | ARC-Challenge | GPQA  | GSM8K | MMLU  | Avg         |
> | ------------------ | ------------- | ----- | ----- | ----- | ----------- |
> | OSS-Instruct-75K   | 51.28         | 27.46 | 49.58 | 62.14 | 45.76       |
> | $\Delta ToEdit$    | 51.79         | 28.79 | 49.36 | 62.04 | **46.1328** |
> |                    |               |       |       |       |             |
> | Evol-Instruct-110K | 52.9          | 27.9  | 50.87 | 62.4  | 46.616      |
> | $\Delta ToEdit$    | 52.22         | 29.69 | 50.87 | 62.6  | **46.9216** |
>
> 2. General Instruction tuning.
>
> |                       | PIQA  | BoolQ | HellaSwag | SIQA  | Winogrande | Avg       |
> | --------------------- | ----- | ----- | --------- | ----- | ---------- | --------- |
> | CoT [3]               | 79.87 | 81.28 | 59.72     | 49.69 | 74.51      | 69.014    |
> | $\Delta ToEdit$       | 80.25 | 81.16 | 59.74     | 50.56 | 74.59      | **69.26** |
> |                       |       |       |           |       |            |           |
> | FLAN V2 [4]           | 80.79 | 84.04 | 59.98     | 51.43 | 74.66      | 70.18     |
> | $\Delta ToEdit$       | 80.69 | 85.20 | 59.99     | 52.00 | 75.37      | **70.65** |
> |                       |       |       |           |       |            |           |
> | Open Assistant 1  [5] | 79.65 | 83.18 | 60.51     | 48.52 | 74.11      | 69.194    |
> | $\Delta ToEdit$       | 79.98 | 83.91 | 60.34     | 48.31 | 74.66      | **69.44** |

---

> ### Author Response · Authors · 2024-11-21
> **Response - 2**
>
> > [Q2] It would help if the paper provided further clarification on the specific threshold values used for token-level editing (e.g., the probability threshold $p=0.99$.
>
> **We supplement 4 experiments on hyper-parameter $p$, including: (1) ablation studies of values , (2) token percentage statistics, (3) comparisons of sampling strategies ,  and (4)  an ablation study on sampling size.**
>
> As frist table below shows different $p$ influences on BioMed, different values lead to fluctuations in data performance. The second table below presents the distribution percentages across different probability value ranges. As stated in line 241-244, we need to refine the data while preserving mainly source distribution. As shown in Figure 3,  a larger $p$ indicates fewer tokens will be resampled, while a smaller $p$ results in more tokens being resampled. Balancing performance and the preservation of data distribution, we set $p=0.99$ as threshold for our experiments. The third below shows the results of different sampling strategies.  Specifically, to control variables, we set $k=8$ for top-k sampling and $p=0.99$ for top-p sampling. We use reject sampling implementation in [11]. The results of reject sampling, top-p, and top-k are comparable. However, top-p involves a dynamic sampling range, and reject sampling requires multiple rounds of computation, leading to increased overhead. Considering computational efficiency, we chose top-k for sampling. This aligns with our original objective of maintaining minimal computational overhead. This aligns with our initial objective of minimizing computational overhead as much as possible. The final table shows the ablation study on sampling size of top-k.  The improvement achieved with larger values is relatively small. Therefore, we chose $k=8$ in our experiments.
>
> Performance impact of different $p$
>
> | Criteria                        | PubMedqa | MQP   | RCT   | USMLE | ChemProt | Avg    |
> | :------------------------------ | -------- | ----- | ----- | ----- | -------- | ------ |
> | Resampled Tokens $p \geq 0.99$  | 64.5     | 55.73 | 30.95 | 27.65 | 14.6     | 38.686 |
> | Resampled Tokens $p \geq 0.999$ | 63.6     | 55.4  | 29.09 | 28.12 | 16.2     | 38.482 |
> | Resampled Tokens $p \leq 0.1$   | 62.4     | 51.47 | 25.6  | 29.14 | 10.0     | 35.722 |
> | Resampled Tokens $p \leq 0.01$  | 65.4     | 54.91 | 28.19 | 27.80 | 11.0     | 37.46  |
> | Resampled Tokens $p \leq 0.001$ | 64.2     | 56.39 | 35.0  | 27.80 | 12.4     | 39.158 |
>
> Token distribution across different probability ranges.
>
> | Probability Range | 0.0-0.1     | 0.1-0.2    | 0.2-0.3    | 0.3-0.4    | 0.4-0.5    | 0.5-0.6    | 0.6-0.7    | 0.7-0.8    | 0.8-0.9    | 0.9-1.0     |
> | ----------------- | ----------- | ---------- | ---------- | ---------- | ---------- | ---------- | ---------- | ---------- | ---------- | ----------- |
> | Percentage        | 34.7%       | 8.1%       | 5.4%       | 4.4%       | 3.8%       | 3.6%       | 3.7%       | 4.0%       | 5.2%       | 27.1%       |
> | Token Count       | 388,626,330 | 90,716,809 | 60,477,872 | 49,278,266 | 42,558,503 | 40,318,546 | 41,438,924 | 44,798,424 | 58,238,944 | 303,543,988 |
>
> Results of different sampling strategies.
>
> | Sampling Strategy | PubMedqa | medmcqa | medqa_4options |
> | ----------------- | -------- | ------- | -------------- |
> | Top-k             | 64.5     | 26.13   | 24.82          |
> | Top-p             | 63.8     | 27.11   | 25.61          |
> | Reject Sampling   | 64.5     | 28.9    | 28.2           |
>
> Ablation study on sampling size on $k$ of top-k.
>
> | Sampling Strategy | PubMedqa | medmcqa | medqa_4options |
> | ----------------- | -------- | ------- | -------------- |
> | $k=8$             | 64.5     | 26.13   | 24.82          |
> | $k=64$            | 63.8     | 28.14   | 27.34          |

---

> ### Author Response · Authors · 2024-11-21
> **Response - 3**
>
> > [Q3] : Although common, it might still be useful to expand that PPL stands for perplexity. It will be helpful for new researchers.
>
> PPL stands for perplexity, a commonly metric in NLP to evaluate the quality of language models. It measures how well a probabilistic model predicts a given dataset, with lower values indicating better performance. Formally, the perplexity of a language model is calculated as:
>
> $\text{PPL} = 2^{-\frac{1}{N} \sum_{i=1}^{N} \log_2 P(x_i)}$
>
> Alternatively, it can also be expressed as:
>
> $\text{PPL} = \exp\left(-\frac{1}{N} \sum_{i=1}^{N} \log P(x_i)\right)$
>
> Where $N$ is the number of tokens in the dataset, $P(x_i)$ is the predicted probability of the $i$-th token. Perplexity essentially represents the exponential of the average negative log-likelihood of the predicted tokens, indicating how “perplexed” the model is when making predictions. In our implementation, we use the following code to calculate PPL:
> ```
> train_outputs = model(**model_inputs)
> neg_log_likelihood = train_outputs.loss
> ppl = np.exp(stabilized_loss)
> ```

---

> ### Author Response · Authors · 2024-11-21
> **Response - 4**
>
> > [Q4] : The paper uses terms like "non-iterative model collapse" and "coverage collapse." It would be helpful broader audience to better understand these terms if it was clearly defined.
>
> `non-iterative model collapse` We provide a detailed explanation of non-iterative model collapse as below, and offer a comparison with orignal iterative model collapse.
>
> We define “non-iterative model collapse” as the performance degradation caused by directly mixing general synthetic data without iterative training. Theoretically, without additional regularization constraints to guide data generation, the variance of the model-generated data gradually decreases during this process. The diversity of the generated data diminishes over time, ultimately leading to the collapse of the model itself.
>
> `Difference with iterative model collapse`
>
> **From a setting perspective**: the difference between the two lies in their scope: non-iterative model collapse is not confined to training on self-generated data, enabling it to uncover broader properties of synthetic data. For instance, in our paper, we train GPT-2 using the Cosmopedia dataset in a single generation, which was generated by Mixtral-8x7B-Instruct-v0.1. In contrast, iterative model collapse requires training on self-generated data over multiple generations.
>
> **From a property perspective**: the non-iterative model collapse highlights the gap between human data and general purely synthetic data, particularly in terms of distributional properties and n-gram features. The iterative model collapse illustrates the iterative evolution of the model, resembling a self-play process. This process illustrates the gradual evolution of self-generated data without involving an analysis of the differences in nature between self-generated and real data [1,2,3].
>
> They both ultimately lead to model collapse, driven by the same cause—synthetic data, while they investigate different aspects of synthetic data.In common model training scenarios, we do not deliberately use self-generated data to train the model itself.
>
> **Non-iterative model collapse is more realistic.** The most common setting is training model on mixture of human and synthetic data, where the synthetic data is not generated by the model itself, even we don't know where this synthetic data comes from. Moreover, there are already numerous popular datasets that combine synthetic and real data for training, such as UltraChat [4] and OpenOrca [5]. Therefore, studying synthetic data under the non-iterative model collapse setting is more realistic.
>
> `Coverage collapse`
>
> **“Coverage collapse” refers to a phenomenon where synthetic data is distributed within a much narrower range compared to human data, even when the data sizes are the same.** For instance, as shown in Figure 3, the PPL range of synthetic data is confined to [0, 14], whereas the PPL range of human data spans [0, 100]. Despite this disparity, the integral of the two distributions remains the same. This significant distribution gap is what we define as “coverage collapse.”
>
> > [Further Suggestion]
>
> Thank you very much for your insightful suggestions. These are highly valuable and align closely with the directions we are currently exploring. We agree that using an ensemble of models with varied pretraining sources could improve the robustness and diversity of token selection. To address this, we plan to explore ensemble-based strategies, including model voting, to enhance token selection quality and minimize single-model biases. These investigations will help determine whether ToEdit’s effectiveness requires adaptations for specific configurations. Relevant experiments are currently underway.

---

> > ### Comment · Reviewer_PCAs · 2024-12-03
> >
> > Dear authors, thank you for addressing my questions and suggestions. I had already provided a score of 8, so I will be keeping my score.

---

### Official Review · Reviewer_4XBZ · 2024-11-03

**Soundness:** 3
**Presentation:** 3
**Contribution:** 2
**Rating:** 8
**Confidence:** 3

**Summary:**

The paper addresses the problem of model collapse in language models trained on synthetic data. The authors propose Token-Level Editing (ToEdit), a method that creates semi-synthetic data by selectively editing tokens based on their probability estimates from a pre-trained model. Key contributions include:

1. Demonstration of non-iterative model collapse when mixing synthetic and human data during pre-training
2. Statistical analysis showing synthetic data suffers from distribution coverage collapse and feature over-concentration
3. A token-level editing approach that theoretically prevents model collapse while improving data quality
4. Experimental validation across pre-training, continual pre-training, and supervised fine-tuning scenarios

**Strengths:**

- Well-structured paper with clear progression
- The authors identify a critical issue in modern LLM training (model collapse with synthetic data)
- Provides thorough statistical analysis of synthetic data's limitations
- Demonstrates non-iterative model collapse, extending beyond previous work focusing on iterative collapse
- Proposed Method is computationally efficient (single forward pass)
- Tests across multiple training scenarios (pre-training, continual pre-training, fine-tuning)
- Uses various downstream tasks for validation

**Weaknesses:**

1. Parameter Sensitivity: The paper mentions using a probability threshold p=0.99 for token resampling but doesn't explore how sensitive the results are to this parameter choice. There's no ablation study showing how different threshold values affect performance. The paper doesn't explore or justify why this specific threshold works best. No ablation studies are provided to show the impact of different threshold values

2. Token Resampling Strategy (Limited Justification):
- The choice of top-k sampling with k=8 seems arbitrary
- The paper mentions they tried other sampling strategies (top-p and rejection sampling) but doesn't provide comparative results

3. Progressive Decrease in Editing
- The assumption that editing operations decrease with a fixed ratio η seems mathematically convenient but lacks empirical validation
- No real-world evidence is provided to support this pattern of decay

4. Distribution Analysis:
- The U-shaped distribution in Figure 6 needs better explanation of why it's a good indicator for token editing
- The "coverage collapse" phenomenon mentioned multiple times could be better defined and explained
- The relationship between "feature over-concentration" and model performance isn't clearly explained

4. Experimental Setup:
- Figure 5 shows results about "DSIR-Selected Data" but details about this selection process aren't well explained
- Details about the continual pre-training setup across different domains (Biomedicine, Finance, Math) are sparse

5. Results Interpretation:
- The improvements shown in Tables 1-3 could use more discussion about statistical significance
-
- Compare different sampling strategies (top-k, top-p, nucleus sampling) with ablation studies

**Questions:**

1. Can you please provide justification for choosing p =0.99 and k =8
2. Can you please provide more evidences for Progressive Decrease in Editing
3. Can you elaborate the relationship between perplexity improvements and actual model capabilities.

---

> ### Author Response · Authors · 2024-11-21
> **Response - 1**
>
> We deeply appreciate your positive comments and valuable feedback. In the following, we will address your concerns and refine our paper accordingly.
>
> > [Q1] :Parameter Sensitivity, There's no ablation study showing how different threshold values affect performance.
>
> **We supplement 4 experiments on hyper-parameter $p$, including: (1) ablation studies of values , (2) token percentage statistics, (3) comparisons of sampling strategies (see in Q2),  and (4)  an ablation study on sampling size (see in Q2) .**  As the first table below shows different $p$ influences on BioMed, different values lead to fluctuations in data performance. The second table below presents the distribution percentages across different probability value ranges. As stated in lines 241-244, we need to refine the data while preserving mainly source distribution. As shown in Figure 3, a larger p indicates fewer tokens will be resampled, while a smaller p results in more tokens being resampled. Balancing performance and the preservation of data distribution, we set p=0.99 as the threshold for our experiments.
>
> The ablation studies of $p$ on BioMed.
>
> | Criteria                        | PubMedqa | MQP   | RCT   | USMLE | ChemProt | Avg    |
> | :------------------------------ | -------- | ----- | ----- | ----- | -------- | ------ |
> | Resampled Tokens $p \geq 0.99$  | 64.5     | 55.73 | 30.95 | 27.65 | 14.6     | 38.686 |
> | Resampled Tokens $p \geq 0.999$ | 63.6     | 55.4  | 29.09 | 28.12 | 16.2     | 38.482 |
> | Resampled Tokens $p \leq 0.1$   | 62.4     | 51.47 | 25.6  | 29.14 | 10.0     | 35.722 |
> | Resampled Tokens $p \leq 0.01$  | 65.4     | 54.91 | 28.19 | 27.80 | 11.0     | 37.46  |
> | Resampled Tokens $p \leq 0.001$ | 64.2     | 56.39 | 35.0  | 27.80 | 12.4     | 39.158 |
>
> Statistics of token percentages and counts across probability ranges.
>
> | Probability Range | 0.0-0.1     | 0.1-0.2    | 0.2-0.3    | 0.3-0.4    | 0.4-0.5    | 0.5-0.6    | 0.6-0.7    | 0.7-0.8    | 0.8-0.9    | 0.9-1.0     |
> | ----------------- | ----------- | ---------- | ---------- | ---------- | ---------- | ---------- | ---------- | ---------- | ---------- | ----------- |
> | Percentage        | 34.7%       | 8.1%       | 5.4%       | 4.4%       | 3.8%       | 3.6%       | 3.7%       | 4.0%       | 5.2%       | 27.1%       |
> | Token Count       | 388,626,330 | 90,716,809 | 60,477,872 | 49,278,266 | 42,558,503 | 40,318,546 | 41,438,924 | 44,798,424 | 58,238,944 | 303,543,988 |

---

> ### Author Response · Authors · 2024-11-21
> **Response - 2**
>
> > [Q2] : Token Resampling Strategy.
>
> `The choice of top-k sampling with k=8 seems arbitrary.`  **We conduct experiments on a validation set to choose a sampling strategy and sampling size.** The first below shows the results of different sampling strategies.  Specifically, to control variables, we set $k=8$ for top-k sampling and $p=0.99$ for top-p sampling. We use reject sampling implementation in [14]. The results of reject sampling, top-p, and top-k are comparable. However, top-p involves a dynamic sampling range, and reject sampling requires multiple rounds of computation, leading to increased overhead. Considering computational efficiency, we chose top-k for sampling. This aligns with our original objective of maintaining minimal computational overhead. This aligns with our initial objective of minimizing computational overhead as much as possible.  Table 4 shows the ablation study on sampling size of top-k.  The improvement achieved with larger values is relatively small. Therefore, we chose $k=8$ in our experiments.
>
> Results of different sampling strategies.
>
> | Sampling Strategy | PubMedqa | medmcqa | medqa_4options |
> | ----------------- | -------- | ------- | -------------- |
> | Top-k             | 64.5     | 26.13   | 24.82          |
> | Top-p             | 63.8     | 27.11   | 25.61          |
> | Reject Sampling   | 64.5     | 28.9    | 28.2           |
>
> Ablation study on sampling size on $k$ of top-k.
>
> | Sampling Strategy | PubMedqa | medmcqa | medqa_4options |
> | ----------------- | -------- | ------- | -------------- |
> | $k=8$             | 64.5     | 26.13   | 24.82          |
> | $k=64$            | 63.8     | 28.14   | 27.34          |

---

> ### Author Response · Authors · 2024-11-21
> **Response - 3**
>
> > [Q3] : Progressive Decrease in Editing :  editing operations decrease with a fixed ratio η seems mathematically convenient but lacks empirical validation, No real-world evidence is provided to support this pattern of decay.
>
> **We present the percentage statistics of edited tokens in table below, demonstrating that the edited tokens  indeed exhibit a progressive decrease.** Specifically, We observe that the percentage of edited tokens (above the threshold $p>0.99$) decreases as the generation number increases. Theoretically,  this is a process of distribution shifting. When tokens ($p>0.99$) are resampled, randomness is introduced.  The sampling process can select tokens with lower probabilities. Then, tokens ($p>0.99$) are replaced, leading to a reduction of edited tokens in subsequent generations. The following table provides real-world evidence for this pattern of decay.
>
> Percentage of tokens requiring edits in the Natural-Instructions dataset. The total number of tokens is 4,671,834.
>
> |                   | Generation 1 (source) | Generation 2 | Generation 3 |
> | ----------------- | --------------------- | ------------ | ------------ |
> | Tokens ($p>0.99$) | 584,103               | 549,519      | 517,433      |
> | Percentage        | 12.5%                 | 11.76%       | 11.08%       |

---

> ### Author Response · Authors · 2024-11-21
> **Response - 4**
>
> >  [Q4] : Distribution Analysis
>
> `The U-shaped distribution in Figure 6 needs better explanation of why it's a good indicator for token editing`
>
> **From the perspective of information theory,** we can analyze the filtering potential of U-shape distribution as follows:
>
>  We utilize the U-shape distribution (Figure 6) to re-sample tokens in the high-probability region, aiming to adjust the U-shaped distribution toward a uniform distribution. By doing so, we can maximize the information entropy. According to information theory,
>
> **Lemma 1**: Let \(X\) be a discrete random variable with \(n\) possible outcomes. If the probability of each outcome is uniform, i.e., $P(x_i) = \frac{1}{n}$ for all $i \in \{1, 2, \dots, n\}$, the Shannon entropy is maximized, given by:
>
> $H(X) = -\sum_{i=1}^{n} \frac{1}{n} \log \frac{1}{n} = \log n.$
>
> This represents the maximum uncertainty achievable, implying that the dataset carries the maximum possible information content.  Loosely speaking, the uniform distribution possesses the maximum information entropy.
>
> **From the perspective of language model learning,** our method emphasizes the importance of poorly learned data.  Specifically, we resample easy tokens and encourage the model to focus on learning more challenging ones. Our method can enhance under-learned data learning by resampling high-probability tokens.
>
> `The "coverage collapse" phenomenon mentioned multiple times could be better defined and explained`
>
> “Coverage collapse” refers to a phenomenon where synthetic data is distributed within a much narrower range compared to human data, even when the data sizes are the same. For instance, as shown in Figure 3, the PPL range of synthetic data is confined to [0, 14], whereas the PPL range of human data spans [0, 100]. Despite this disparity, the integral of the two distributions remains the same. This significant distribution gap is what we define as “coverage collapse.”
>
> `"Feature over-concentration" and model performance isn't clearly explained`
>
> "Feature over-concentration" refers to the excessive repetition of n-gram features overly repeat and higher similarity in latent space.  As shown in Figure 4, 9, and 10, the feature distribution of real data differs significantly from that of synthetic data. In particular, n-grams in synthetic data are heavily concentrated in high-frequency features. This phenomenon results in low data diversity, which intuitively hinders the model’s generalization ability. Consequently, training on such low-diversity data leads to a decline in model performance.

---

> ### Author Response · Authors · 2024-11-21
> **Response - 5**
>
> > [Q5] : Experimental Setup
>
> `Figure 5 shows results about "DSIR-Selected Data" but details about this selection process aren't well explained`
>
> **DSIR [11] works by estimating importance weights for each data sample to measure its relevance to the target distribution.** This involves three main steps: first, we leverage n-gram models to fit two distributions of human and synthetic data, $q_{feat}$ and $p_{feat}$, which represent the target and raw distributions, respectively. We use them to compute the likelihood ratio for each sample. Next, we calculate the importance weight for each sample $z_i$ as $w_i = \frac{\hat{p}_{\text{feat}}(z_i)}{\hat{q}_{\text{feat}}(z_i)}$, which quantifies how well the sample aligns with the target distribution.  Finally, we perform importance-weighted sampling without replacement to select examples.
>
> We use DSIR in our data analysis as it allows for principled and computationally efficient selection of synthetic data points that align with the target distribution. And the importance weight also reflects the alignment between the n-gram features of synthetic data and human data. Using DSIR, we can analyze the differences between synthetic and human data across n-gram feature distributions and data matching. As shown in Fig.5, It is challenging to select synthetic data that matches human data characteristics under the significant distribution difference. To obtain high-quality synthetic data, it is essential to focus on improving the data synthesis method.
>
> `Details about the continual pre-training setup across different domains (Biomedicine, Finance, Math) are sparse`
>
> We follow [13] to conduct continual pre-training on Bio, Finance, and Math domains. Specifically, PubMed Abstracts from the Pile are utilized as the pre-training corpora for the biomedicine  domain. For the finance domain, financial news data covering over 7,000 stocks from May 2022 to May 2023 is collected using the FinGPT framework. We continue pre-training OLMo-1B and LLaMA-3-8B on each domain. For implementation, we utilized the official training framework for OLMo-1B, leveraging Fully Sharded Data Parallel (FSDP) for continual pretraining. For LLaMA, we adopted the LLaMA-Factory framework to carry out the continual pretraining process.

---

> ### Author Response · Authors · 2024-11-21
> **Response - 6**
>
> > [Q6] : Results Interpretation
>
> `The improvements shown in Tables 1-3 could use more discussion about statistical significance`
>
> **To further validate the effectiveness of our method, we incorporate more 3 general instruction tuning tasks and 2 code reasoning-related SFT tasks.** As for general instruction tuning tasks, we adopt instruction tuning datasets from [2], including CoT [3], FLAN V2 [4], and Open Assistant 1 [5]. As for code-related reasoning tasks, we utilize OSS-Instruct-75K and Evol-Instruct-110K.
>
> The supplement tables below demonstrate that the ∆ToEdit consistently improves performance across both general instruction tuning and code-related SFT tasks. For code-related tasks, the improvements are particularly evident in ARC-Challenge and GPQA, indicating better reasoning and code comprehension. From the trends observed, our method demonstrates partial improvements across multiple datasets, indirectly validating its effectiveness.
>
> **More discussion of Tables 1, 2, and 3**:  Experiments in our paper demonstrate the effectiveness of our method across the three critical stages of language model training: continual pre-training, pre-training, and fine-tuning. Our method enhances model performance on downstream tasks without increasing data size, showcasing its ability to tap into the potential of existing data. Additionally, these results confirm that semi-synthetic data serves as a viable approach for generating high-quality data and improving model generalization.
>
> As for Table 1, our method achieves consistent improvements over the baseline data for both OLMo-1B and LLaMA-3-8B models. For example, in the Biomedicine domain, OLMo-1B improves from 38.83 to 40.89, while LLaMA-3-8B increases from 56.04 to 56.48. These improvements demonstrate that our ToEdit strategy effectively enhances learning across diverse domains.
>
> As for Table 2, it further supports the robustness of our approach in general pre-training tasks. The average performance of OLMo-1B increases from 32.75 to 33.11, reflecting improved generalization capabilities. While the improvement is modest, the consistent trend across tasks like PIQA, BoolQ, and ARC-c highlights the broader applicability of our method.
>
> As for Table 3, it illustrates the benefits of our method in fine-tuning tasks. The average performance of LLaMA-3-8B improves from 69.34 to 69.70, with notable gains in tasks like Winogrande and SIQA. Similarly, in following table, ToEdit improves other data such as FLAN V2 and Open Assistant 1, with average performance increasing from 70.18 to 70.65 for FLAN V2. These improvements, demonstrate the adaptability of our method to instruction-tuning tasks.
>
> 1. General Instruction tuning.
>
> |                       | PIQA  | BoolQ | HellaSwag | SIQA  | Winogrande | Avg       |
> | --------------------- | ----- | ----- | --------- | ----- | ---------- | --------- |
> | CoT [3]               | 79.87 | 81.28 | 59.72     | 49.69 | 74.51      | 69.014    |
> | $\Delta ToEdit$       | 80.25 | 81.16 | 59.74     | 50.56 | 74.59      | **69.26** |
> |                       |       |       |           |       |            |           |
> | FLAN V2 [4]           | 80.79 | 84.04 | 59.98     | 51.43 | 74.66      | 70.18     |
> | $\Delta ToEdit$       | 80.69 | 85.20 | 59.99     | 52.00 | 75.37      | **70.65** |
> |                       |       |       |           |       |            |           |
> | Open Assistant 1  [5] | 79.65 | 83.18 | 60.51     | 48.52 | 74.11      | 69.194    |
> | $\Delta ToEdit$       | 79.98 | 83.91 | 60.34     | 48.31 | 74.66      | **69.44** |
>
> 2. Code SFT tasks; OSS-Instruct-75K and Evol-Instruct-110K are code-related supervised tasks, containing python, cpp, java and php languages.
>
> |                    | ARC-Challenge | GPQA  | GSM8K | MMLU  | Avg         |
> | ------------------ | ------------- | ----- | ----- | ----- | ----------- |
> | OSS-Instruct-75K   | 51.28         | 27.46 | 49.58 | 62.14 | 45.76       |
> | $\Delta ToEdit$    | 51.79         | 28.79 | 49.36 | 62.04 | **46.1328** |
> |                    |               |       |       |       |             |
> | Evol-Instruct-110K | 52.9          | 27.9  | 50.87 | 62.4  | 46.616      |
> | $\Delta ToEdit$    | 52.22         | 29.69 | 50.87 | 62.6  | **46.9216** |
>
> > [Q7] : Can you please provide justification for choosing p =0.99 and k =8
>
> We have supplemented additional experiments and conducted a detailed analysis; please refer to  [Q2].

---

> ### Author Response · Authors · 2024-11-21
> **Response - 7**
>
> > [Q8] : Can you elaborate the relationship between perplexity improvements and actual model capabilities.
>
> **We further include 22 validation datasets from the Pile and 7 general understanding downstream tasks, to analyze the relationship between perplexity improvements and actual model capabilities.** The testing framework follows [4]. Specifically,  as shown in first table below, the PPL decreases as the proportion of purely synthetic data decreases. In second table , the performance on downstream tasks similarly exhibits a gradual improvement with the decline in synthetic data. Although the improvement in PPL does not directly translate to an equivalent increase in downstream task accuracy.  **It can be observed from the trend that improvements in PPL are positively correlated with enhancements in downstream task performance.** The relationship between PPL improvement and downstream performance gain depends on more complicated factors, like instruction tuning methods and base model capabilities.
>
> PPL evaluation results on 22 validation using the testing framework in [1].
>
> |                    | ArXiv | BookCorpus2 | Books3 | DM_Mathematics | Enron_Emails | EuroParl | FreeLaw | GitHub | Gutenberg_(PG-19) | HackerNews | NIH_ExPorter |
> | ------------------ | ----- | ----------- | ------ | -------------- | ------------ | -------- | ------- | ------ | ----------------- | ---------- | ------------ |
> | Human Data         | 22.26 | 25.39       | 22.87  | 10.84          | 23.50        | 30.73    | 12.04   | 4.15   | 16.88             | 32.54      | 23.53        |
> | 25% Synthetic Data | 21.86 | 26.32       | 23.87  | 11.05          | 24.85        | 35.02    | 12.84   | 4.35   | 17.99             | 33.80      | 23.76        |
> | 50% Synthetic Data | 22.50 | 28.01       | 25.75  | 10.84          | 26.56        | 41.99    | 14.02   | 4.67   | 19.70             | 36.12      | 24.61        |
> | 75% Synthetic Data | 24.35 | 31.19       | 28.98  | 11.81          | 30.30        | 56.32    | 16.03   | 5.30   | 22.75             | 40.44      | 26.19        |
> | Synthetic Data     | 35.60 | 43.72       | 47.72  | 17.25          | 66.97        | 129.75   | 29.62   | 12.00  | 50.14             | 87.95      | 39.48        |
>
> |                    | OpenSubtitles | OpenWebText2 | PhilPapers | Pile-CC | PubMed_Abs | PubMed_Cen | StackEx | Ubuntu_IRC | USPTO | Wikipedia | YouSub | Avg       |
> | ------------------ | ------------- | ------------ | ---------- | ------- | ---------- | ---------- | ------- | ---------- | ----- | --------- | ------ | --------- |
> | Human Data         | 28.08         | 25.77        | 33.56      | 26.78   | 18.97      | 15.49      | 10.81   | 20.86      | 19.32 | 24.31     | 21.54  | **21.37** |
> | 25% Synthetic Data | 29.25         | 26.94        | 34.63      | 27.83   | 19.55      | 15.38      | 11.03   | 22.32      | 19.58 | 25.88     | 22.63  | **22.31** |
> | 50% Synthetic Data | 31.00         | 28.76        | 37.48      | 29.36   | 20.51      | 15.89      | 11.54   | 23.53      | 20.51 | 27.57     | 24.91  | **23.90** |
> | 75% Synthetic Data | 34.18         | 32.04        | 42.39      | 32.17   | 22.33      | 16.92      | 12.55   | 26.54      | 22.21 | 30.68     | 28.98  | **27.03** |
> | Synthetic Data     | 57.83         | 53.94        | 78.18      | 54.69   | 34.82      | 23.87      | 20.47   | 51.78      | 37.24 | 46.12     | 65.49  | **49.30** |
>
> Downstream tasks evaluation using the testing framework in [1].
>
> |                    | TruthfulQA | LogiQA | Wino. | PIQA  | ARC-E | BoolQ | OBQA | avg             |
> | ------------------ | ---------- | ------ | ----- | ----- | ----- | ----- | ---- | --------------- |
> | Human Data         | 32.68      | 23.03  | 51.3  | 64.42 | 44.4  | 60.98 | 15   | **41.68714286** |
> | 25% Synthetic Data | 27.91      | 21.37  | 50.12 | 63.93 | 43.94 | 62.29 | 15.4 | **40.70857143** |
> | 50% Synthetic Data | 30.84      | 22.58  | 52.41 | 63.33 | 44.02 | 62.14 | 16   | **41.61714286** |
> | 75% Synthetic Data | 29.5       | 22.65  | 49.8  | 63.44 | 44.53 | 61.56 | 17.2 | **41.24**       |
> | Synthetic Data     | 28.89      | 22.58  | 49.72 | 63    | 46.3  | 54.53 | 16.8 | **40.26**       |

---

> > ### Comment · Reviewer_4XBZ · 2024-11-26
> > **Thank you for the response.**
> >
> > Thank you for all the efforts the authors have put into the rebuttal. This is commendable work. I have increased my scores. Please include all your additional experiments in appendix of the paper.

---

> > > ### Author Response · Authors · 2024-11-26
> > >
> > > Thank you for your acknowledgment! We will include all additional experiments and discussions in the revision.

---

### Official Review · Reviewer_wNZi · 2024-11-05

**Soundness:** 2
**Presentation:** 3
**Contribution:** 1
**Rating:** 3
**Confidence:** 4

**Summary:**

This paper investigates the role of synthetic data for LM pretraining.

1. The main observation/finding is:

> The mixture of synthetic data and real data is worse than real data only, which is evaluated with perplexity. (trained with Dolma + x% of Cosmopedia, and eval with wikitext-103, Redpajama, RefineWeb, C4-en)

2. A new data synthesis method, TOEDIT, which resamples tokens with lower perplexity using a language model. The authors claim TOEDIT outperforms real data in continual pretraining and SFT scenarios.

**Strengths:**

The analysis of Cosmopedia synthetic data is comprehensive and provides valuable insights for future research

The theoretical framework is interesting and well-developed

The paper addresses an important question in the field about the utility of synthetic data in pretraining

**Weaknesses:**

**Major Concerns**

1. Significant Literature Gap and Contradictory Findings
- The paper overlooks a crucial published work [1] that directly contradicts its main findings
- [1] demonstrates that mixing synthetic and real data improves perplexity across various domains in the Pile and enhances downstream task performance
- This fundamental contradiction needs to be addressed and explained

Figure2 in [1] shows that by mixing the synthetic data with the real C4 data, LM can get lower perplexity on various domain corpus of the Pile. [1] also shows the introduce of synthetic data gave better results on downstream tasks. However, this submission claims the mixture of data is worse than real data only, by evaluating the perplexity. (I suspect the evaluation is incorrect which lead to this contrast observation. My 2nd concern below will detail more)

2. Flawed Experimental Design for Main Claims

The pretraining real data is Dolma, which consist of wikipedia, C4. Hence, the more real data you use, the lower perplexity on Wikitext-103 and C4-en you should expect. So it is domain mismatching problem, instead of saying synthetic data is not useful.
On the other hand, the synthetic pretraining data Cosmpopedia was build from LLM's rephrase of web-crawled data, which means Cosmopedia's data distribution differs significantly from web-crawled data, making the comparison with RedPajama and RefineWeb potentially unfair.  I would suggest to use proper out of domain text for perplexity evaluation, and also introduce the downstream task evaluation.  I would suggest to follow the evaluation framework from [1] for better comparison

3. TOEDIT Methodology Concerns
- The non-autoregressive token replacement approach may compromise text coherence and grammatical correctness
- No discussion of how grammatical consistency is maintained during token resampling

4. Weak Empirical Results (Baseline performance is concerningly low)
- MMLU: baseline (26.56) vs. proposed (23.63) vs. random (25)
- WinoGrande and Hellaswag results near random guessing. Improvements are marginal (1.3, -0.08) and not statistically significant


**Minor Concern**

1.  The conclusion drawn from "75% words under 0.6 probability" (Lines 265-267) ignores fundamental linguistic entropy
- English typically has 10-11 bits per word entropy
- The observed probability distribution may be natural rather than indicating filtering potential


[1][Rephrasing the Web: A Recipe for Compute and Data-Efficient Language Modeling](https://aclanthology.org/2024.acl-long.757) (Maini et al., ACL 2024)

**Questions:**

N/A

---

> ### Author Response · Authors · 2024-11-21
> **Response - 1.1**
>
> We sincerely thank you for your critical feedback and valuable suggestions. Below, we will strive to address your concerns and refine our paper accordingly.
>
> > [Q1] : Significant Literature Gap and Contradictory Findings
>
> `This fundamental contradiction needs to be addressed and explained`
>
> **We should emphasize that the conclusions in [1] do not contradict ours; on the contrary, the findings in [1] are consistent with ours.**
>
> **Cosmopedia [12] is produced by pure synthetic data method (prompting LLMs), but both *Rephrasing the Web*[1] and ours are semi-synthetic data methods (reformatting  and token replacing).  The former cut-offs long-tail distribution [7], while the latter preserves and adjusts it.**
>
> Specifically, both *Rephrasing the Web* [1] and token-level editing (ours) aim to refine data while preserving the original source distribution, producing semi-synthetic data. In contrast, purely synthetic data in Cosmopedia lacks the long-tail distribution (Figure 3) and overly concentrates on n-gram features (Figure 4). Ultimately, semi-synthetic data enhances training performance, whereas purely synthetic data results in model collapse. In other words, replacing the whole real sample with synthetic data can damage the performance.
>
> We provide a detailed comparison of Cosmopedia, *Rephrasing the Web* [1] and ours in the below table. The primary distinction between Cosmopedia, [1], and our approach lies in the degree to which the original human data distribution is preserved.
>
> | Method                     | Data Type      | Approach                                                     | Result                               |
> | -------------------------- | -------------- | ------------------------------------------------------------ | ------------------------------------ |
> | **Cosmopedia**             | Pure synthetic | Using a prompt to induce data from LLMs.                     | Reveal non-iterative model collapse. |
> | **Rephrasing the Web [1]** | Semi-synthetic | Using a prompt and source content to guide LLMs to reformat source content. | Improve training performance.        |
> | **ToEdit (Ours)**          | Semi-synthetic | Using the distribution of source content estimated by LLMs (single forward pass) to replace tokens. | Improve training performance.        |
>
> `I would suggest to follow the evaluation framework from [1] for better comparison`
>
> **We have followed your suggestion to conduct evaluation framework from [1].  The results are consistent with our findings.** Specifically,  the PPL increases as the proportion of purely synthetic data grows, while the performance on downstream tasks similarly exhibits a gradual decline with the increase in synthetic data.
>
> PPL evaluation results on 22 validation using the testing framework in [1].
>
> |                    | ArXiv | BookCorpus2 | Books3 | DM_Mathematics | Enron_Emails | EuroParl | FreeLaw | GitHub | Gutenberg_(PG-19) | HackerNews | NIH_ExPorter |
> | ------------------ | ----- | ----------- | ------ | -------------- | ------------ | -------- | ------- | ------ | ----------------- | ---------- | ------------ |
> | Human Data         | 22.26 | 25.39       | 22.87  | 10.84          | 23.50        | 30.73    | 12.04   | 4.15   | 16.88             | 32.54      | 23.53        |
> | 25% Synthetic Data | 21.86 | 26.32       | 23.87  | 11.05          | 24.85        | 35.02    | 12.84   | 4.35   | 17.99             | 33.80      | 23.76        |
> | 50% Synthetic Data | 22.50 | 28.01       | 25.75  | 10.84          | 26.56        | 41.99    | 14.02   | 4.67   | 19.70             | 36.12      | 24.61        |
> | 75% Synthetic Data | 24.35 | 31.19       | 28.98  | 11.81          | 30.30        | 56.32    | 16.03   | 5.30   | 22.75             | 40.44      | 26.19        |
> | Synthetic Data     | 35.60 | 43.72       | 47.72  | 17.25          | 66.97        | 129.75   | 29.62   | 12.00  | 50.14             | 87.95      | 39.48        |

---

> ### Author Response · Authors · 2024-11-21
> **Response - 1.2**
>
> |                    | OpenSubtitles | OpenWebText2 | PhilPapers | Pile-CC | PubMed_Abs | PubMed_Cen | StackEx | Ubuntu_IRC | USPTO | Wikipedia | YouSub | Avg       |
> | ------------------ | ------------- | ------------ | ---------- | ------- | ---------------- | -------------- | ------------- | ---------- | ----------------- | -------------- | ---------------- | --------- |
> | Human Data         | 28.08         | 25.77        | 33.56      | 26.78   | 18.97            | 15.49          | 10.81         | 20.86      | 19.32             | 24.31          | 21.54            | **21.37** |
> | 25% Synthetic Data | 29.25         | 26.94        | 34.63      | 27.83   | 19.55            | 15.38          | 11.03         | 22.32      | 19.58             | 25.88          | 22.63            | **22.31** |
> | 50% Synthetic Data | 31.00         | 28.76        | 37.48      | 29.36   | 20.51            | 15.89          | 11.54         | 23.53      | 20.51             | 27.57          | 24.91            | **23.90** |
> | 75% Synthetic Data | 34.18         | 32.04        | 42.39      | 32.17   | 22.33            | 16.92          | 12.55         | 26.54      | 22.21             | 30.68          | 28.98            | **27.03** |
> | Synthetic Data     | 57.83         | 53.94        | 78.18      | 54.69   | 34.82            | 23.87          | 20.47         | 51.78      | 37.24             | 46.12          | 65.49            | **49.30** |
>
> Downstream tasks evaluation using the testing framework in [1].
>
> |                    | TruthfulQA | LogiQA | Wino. | PIQA  | ARC-E | BoolQ | OBQA | avg             |
> | ------------------ | ---------- | ------ | ----- | ----- | ----- | ----- | ---- | --------------- |
> | Human Data         | 32.68      | 23.03  | 51.3  | 64.42 | 44.4  | 60.98 | 15   | **41.68714286** |
> | 25% Synthetic Data | 27.91      | 21.37  | 50.12 | 63.93 | 43.94 | 62.29 | 15.4 | **40.70857143** |
> | 50% Synthetic Data | 30.84      | 22.58  | 52.41 | 63.33 | 44.02 | 62.14 | 16   | **41.61714286** |
> | 75% Synthetic Data | 29.5       | 22.65  | 49.8  | 63.44 | 44.53 | 61.56 | 17.2 | **41.24**       |
> | Synthetic Data     | 28.89      | 22.58  | 49.72 | 63    | 46.3  | 54.53 | 16.8 | **40.26**       |

---

> ### Author Response · Authors · 2024-11-21
> **Response - 2**
>
> > [Q2] Flawed Experimental Design for Main Claims
>
> **We provide the domain statistics for Dolma, demonstrating that It is not domain mismatching problem.** We supplement the testing results followed your suggestion above, which are consistent with the findings reported in our paper.
>
>  `It is domain mismatching problem ` We can't fully agree on this point, it is not domain mismatching problem. **As shown in below table, C4 and wikipedia account for approximately 6.86% and 0.14%, respectively, of this dataset.** Specifically, Dolma (v1.6) comprises 7 subdomains: C4, Wikipedia, Wikibooks, Common Crawl, Reddit, PeS2o, Project Gutenberg, and The Stack. This small portion of in-domain data has minimal impact on PPL testing.
>
> `Cosmopedia's data distribution differs significantly from web-crawled data, making the comparison with RedPajama and RefineWeb potentially unfair.` From a pretraining perspective, Cosmopedia was generated by Mixtral-8x7B-Instruct-v0.1, which was likely trained on the same open-source datasets as well. Moreover, Cosmopedia is currently the largest publicly available open-source general synthetic dataset, created by diverse prompts. It comprises 7 subdomains: web_samples, stanford_edu, wikihow, openstax, khanacademy, and automathtext.
>
> Additionally, we are not claiming that synthetic data is ineffective; rather, we emphasize that different synthetic data generation methods lead to varying outcomes. Therefore, we compared Cosmopedia and other data mixture using the same general PPL testing. To provide a more thorough explanation, we apply the testing framework from [1], which includes 22 validation datasets and 7 downstream tasks. The additional experimental results are consistent with the findings reported in our paper.
>
> Dolma dataset statistics (v1.6) (Quoted from [here](https://huggingface.co/datasets/allenai/dolma))
>
> | Source               | Doc Type     | UTF-8 bytes (GB) | Documents (millions) | Unicode words (billions) | Llama tokens (billions) |
> | -------------------- | ------------ | ---------------- | -------------------- | ------------------------ | ----------------------- |
> | Common Crawl         | web pages    | 9,022            | 3,370                | 1,775                    | 2,281                   |
> | The Stack            | code         | 1,043            | 210                  | 260                      | 411                     |
> | C4                   | web pages    | 790              | 364                  | 153                      | 198                     |
> | Reddit               | social media | 339              | 377                  | 72                       | 89                      |
> | PeS2o                | STEM papers  | 268              | 38.8                 | 50                       | 70                      |
> | Project Gutenberg    | books        | 20.4             | 0.056                | 4.0                      | 6.0                     |
> | Wikipedia, Wikibooks | encyclopedic | 16.2             | 6.2                  | 3.7                      | 4.3                     |
> | Total                |              | 11,519           | 4,367                | 2,318                    | 3,059                   |

---

> ### Author Response · Authors · 2024-11-21
> **Response - 3.1**
>
> > [Q3] Methodology Concerns: The non-autoregressive token replacement approach may compromise text coherence and grammatical correctness
>
> Thank you very much for pointing out such an important point. We will address this in detail.
>
> Since data synthesis is a prerequisite for model training, we aim to ensure that the cost of data synthesis does not exceed the cost of training itself. When designing data synthesis algorithms, we must balance synthesis efficiency and effectiveness, considering both autoregressive and non-autoregressive approaches. Autoregressive methods leverage the LMs to generate coherent text sequentially. In contrast, non-autoregressive methods resample individual tokens based on their probability distributions.
>
> Specifically, our ToEdit modifies data using the probability distribution generated in a single forward pass. For instance, if the generated sequence length is 1024, the computational cost of autoregressive methods would be 1024 times higher than ours. This efficiency advantage is why our method can run effectively on GPUs like the 3090 or 4090 series.
>
> However, we acknowledge that non-autoregressive methods may introduce  the semantic inconsistency problem. Resampled tokens may not fit seamlessly into a given sentence.  To address this issue, we determine the sampling threshold through validation experiments, aiming to reduce aggressive replacements. We perform sampling in high-probability regions, focusing on tokens that are relatively easier to predict. And we manually verify and tune threshold on a validation set before applying it. A larger $p$ indicates fewer tokens will be resampled, while a smaller $p$ results in more tokens being resampled.
>
> We supplement more experiments and discussion about hyper-parameter $p$. As table below shows different $p$ influence performance on BioMed, different values lead to the fluctuations in data performance. The second below presents the distribution percentages across different probability value ranges. Balancing the sampling rate, token proportion distribution, and performance, we selected $p=0.99$.  The final two tables respectively present the results of different sampling strategies and the ablation study on hyperparameters.
>
> Performance impact of different $p$
>
> | Criteria                        | PubMedqa | MQP   | RCT   | USMLE | ChemProt | Avg    |
> | :------------------------------ | -------- | ----- | ----- | ----- | -------- | ------ |
> | Resampled Tokens $p \geq 0.99$  | 64.5     | 55.73 | 30.95 | 27.65 | 14.6     | 38.686 |
> | Resampled Tokens $p \geq 0.999$ | 63.6     | 55.4  | 29.09 | 28.12 | 16.2     | 38.482 |
> | Resampled Tokens $p \leq 0.1$   | 62.4     | 51.47 | 25.6  | 29.14 | 10.0     | 35.722 |
> | Resampled Tokens $p \leq 0.01$  | 65.4     | 54.91 | 28.19 | 27.80 | 11.0     | 37.46  |
> | Resampled Tokens $p \leq 0.001$ | 64.2     | 56.39 | 35.0  | 27.80 | 12.4     | 39.158 |
>
> Statistics of token percentages and counts across probability ranges.
>
> | Probability Range | 0.0-0.1     | 0.1-0.2    | 0.2-0.3    | 0.3-0.4    | 0.4-0.5    | 0.5-0.6    | 0.6-0.7    | 0.7-0.8    | 0.8-0.9    | 0.9-1.0     |
> | ----------------- | ----------- | ---------- | ---------- | ---------- | ---------- | ---------- | ---------- | ---------- | ---------- | ----------- |
> | Percentage        | 34.7%       | 8.1%       | 5.4%       | 4.4%       | 3.8%       | 3.6%       | 3.7%       | 4.0%       | 5.2%       | 27.1%       |
> | Token Count       | 388,626,330 | 90,716,809 | 60,477,872 | 49,278,266 | 42,558,503 | 40,318,546 | 41,438,924 | 44,798,424 | 58,238,944 | 303,543,988 |

---

> ### Author Response · Authors · 2024-11-21
> **Response - 3.2**
>
> Results of different sampling strategies.
>
> | Sampling Strategy | PubMedqa | medmcqa | medqa_4options |
> | ----------------- | -------- | ------- | -------------- |
> | Top-k             | 64.5     | 26.13   | 24.82          |
> | Top-p             | 63.8     | 27.11   | 25.61          |
> | Reject Sampling   | 64.5     | 28.9    | 28.2           |
>
> Ablation study on sampling size on $k$ of top-k.
>
> | Sampling Strategy | PubMedqa | medmcqa | medqa_4options |
> | ----------------- | -------- | ------- | -------------- |
> | $k=8$             | 64.5     | 26.13   | 24.82          |
> | $k=64$            | 63.8     | 28.14   | 27.34          |

---

> ### Author Response · Authors · 2024-11-21
> **Response - 4**
>
> > [Q4] Weak Empirical Results
>
> `Baseline performance is concerningly low.` The baselines model (OLMo-1B) is a relatively weaker model of reasoning, which is trained on open-source corpora. However, it provides more open-source data and code, facilitating us to conduct research.
>
> **To further validate the effectiveness of our method, we incorporate more 3 general instruction tuning tasks and 2 code reasoning-related SFT tasks.** As for general instruction tuning tasks, we adopt instruction tuning datasets from [2], including CoT [3], FLAN V2 [4] and Open Assistant 1[5]. As for code-related reasoning tasks, we utilize OSS-Instruct-75K and Evol-Instruct-110K.
>
> The experimental results demonstrate that the ∆ToEdit consistently improves performance across both general instruction tuning and code-related SFT tasks. In general instruction tuning, ∆ToEdit enhances average performance across CoT, FLAN V2, and Open Assistant, with gains in SIQA, and Winogrande. For code-related tasks, the improvements are particularly evident in ARC-Challenge and GPQA, indicating better reasoning and code comprehension. These findings validate the effectiveness of the proposed methods in enhancing both generalization and task-specific performance.
>
> 3 more tasks of general Instruction tuning.
>
> |                      | PIQA  | BoolQ | HellaSwag | SIQA  | Winogrande | Avg       |
> | -------------------- | ----- | ----- | --------- | ----- | ---------- | --------- |
> | CoT [3]              | 79.87 | 81.28 | 59.72     | 49.69 | 74.51      | 69.014    |
> | $\Delta ToEdit$      | 80.25 | 81.16 | 59.74     | 50.56 | 74.59      | **69.26** |
> |                      |       |       |           |       |            |           |
> | FLAN V2 [4]          | 80.79 | 84.04 | 59.98     | 51.43 | 74.66      | 70.18     |
> | $\Delta ToEdit$      | 80.69 | 85.20 | 59.99     | 52.00 | 75.37      | **70.65** |
> |                      |       |       |           |       |            |           |
> | Open Assistant 1 [5] | 79.65 | 83.18 | 60.51     | 48.52 | 74.11      | 69.194    |
> | $\Delta ToEdit$      | 79.98 | 83.91 | 60.34     | 48.31 | 74.66      | **69.44** |
>
> 2 more Code SFT tasks; OSS-Instruct-75K and Evol-Instruct-110K are code-related supervised tasks, containing python, cpp, java and php languages.
>
> |                    | ARC-Challenge | GPQA  | GSM8K | MMLU  | Avg         |
> | ------------------ | ------------- | ----- | ----- | ----- | ----------- |
> | OSS-Instruct-75K   | 51.28         | 27.46 | 49.58 | 62.14 | 45.76       |
> | $\Delta ToEdit$    | 51.79         | 28.79 | 49.36 | 62.04 | **46.1328** |
> |                    |               |       |       |       |             |
> | Evol-Instruct-110K | 52.9          | 27.9  | 50.87 | 62.4  | 46.616      |
> | $\Delta ToEdit$    | 52.22         | 29.69 | 50.87 | 62.6  | **46.9216** |

---

> ### Author Response · Authors · 2024-11-21
> **Response - 5**
>
> > [Q5] The observed probability distribution may be natural rather than indicating filtering potential.
>
> We appreciate you for providing an excellent reference.  We provide the following discussion for `U-shape distribution indicate filtering potential`:
>
> From the perspective of language model learning, our method emphasizes the importance of poorly learned data, as phenomena like linguistic entropy you provided.  Specifically, Our method can enhance under-learned data learning by resampling high-probability tokens.
>
> Furthermore, we can analyze the filtering potential of U-shape distribution through the lens of information theory. According to information theory:
>
> **Lemma 1**: Let \(X\) be a discrete random variable with \(n\) possible outcomes. If the probability of each outcome is uniform, i.e., $P(x_i) = \frac{1}{n}$ for all $i \in \{1, 2, \dots, n\}$, the Shannon entropy is maximized, given by:
>
> $H(X) = -\sum_{i=1}^{n} \frac{1}{n} \log \frac{1}{n} = \log n.$
>
> This represents the maximum uncertainty achievable, implying that the dataset carries the maximum possible information content.  Loosely speaking, the uniform distribution possesses the maximum information entropy. To leverage this property, we utilize the U-shape distribution (Figure 6) to re-sample tokens in the high-probability region, aiming to adjust the U-shaped distribution toward a uniform distribution. By doing so, we can maximize the information entropy.

---

> ### Author Response · Authors · 2024-11-21
> **Reference**
>
> [1] Maini P, Seto S, Bai H, et al. Rephrasing the web: A recipe for compute and data-efficient language modeling[J]. arXiv preprint arXiv:2401.16380, 2024.
>
> [2] Xia M, Malladi S, Gururangan S, et al. Less: Selecting influential data for targeted instruction tuning[J]. arXiv preprint arXiv:2402.04333, 2024.
>
> [3] Wei J, Wang X, Schuurmans D, et al. Chain-of-thought prompting elicits reasoning in large language models[J]. Advances in neural information processing systems, 2022, 35: 24824-24837.
>
> [4] Longpre S, Hou L, Vu T, et al. The flan collection: Designing data and methods for effective instruction tuning[C]//International Conference on Machine Learning. PMLR, 2023: 22631-22648.
>
> [5] Köpf A, Kilcher Y, von Rütte D, et al. Openassistant conversations-democratizing large language model alignment. CoRR, abs/2304.07327, 2023. doi: 10.48550[J]. arXiv preprint arXiv.2304.07327.
>
> [6] Ilia Shumailov, Zakhar Shumaylov, Yiren Zhao, Nicolas Papernot, Ross Anderson, and Yarin Gal. Ai models collapse when trained on recursively generated data. Nature, 631(8022):755–759, 2024.
>
> [7] Elvis Dohmatob, Yunzhen Feng, and Julia Kempe. Model collapse demystified: The case of regression. arXiv preprint arXiv:2402.07712, 2024a.
>
> [8] Elvis Dohmatob, Yunzhen Feng, Pu Yang, Francois Charton, and Julia Kempe. A tale of tails: Model collapse as a change of scaling laws. arXiv preprint arXiv:2402.07043, 2024b.
>
> [9]Ding N, Chen Y, Xu B, et al. Enhancing chat language models by scaling high-quality instructional conversations[J]. arXiv preprint arXiv:2305.14233, 2023.
>
> [10] Lian W, Goodson B, Pentland E. OpenOrca: An Open Dataset of GPT Augmented FLAN Reasoning Traces[EB/OL].(2023)
>
> [11] Xie S M, Santurkar S, Ma T, et al. Data selection for language models via importance resampling[J]. Advances in Neural Information Processing Systems, 2023, 36: 34201-34227.
>
> [12] Loubna Ben Allal, Anton Lozhkov, Guilherme Penedo, Thomas Wolf, and Leandro von Werra. Cosmopedia, 2024. URL https://huggingface.co/datasets/HuggingFaceTB/ cosmopedia.
>
> [13] Cheng D, Huang S, Wei F. Adapting large language models via reading comprehension[C]//The Twelfth International Conference on Learning Representations. 2023.
>
> [14] Liu T, Zhao Y, Joshi R, et al. Statistical rejection sampling improves preference optimization[J]. arXiv preprint arXiv:2309.06657, 2023.

---

> > ### Author Response · Authors · 2024-11-28
> >
> > Dear Reviewer wNZi,
> >
> > We are authors of Submission 11413, *ToEdit: How to Synthesize Text Data to Avoid Model Collapse?*
> >
> > We have followed your suggestions to :
> >
> > **(1) conduct more evaluation from [1],** including 22 validation sets and 7 downstream tasks, which finally aligns with our original findings;
> >
> > **(2) supplement more ablation on hyperparameter p,** including various values of p, percentage statistics, sampling strategies, and sampling sizes;
> >
> > **(3) further validate our method** by including 3 general instruction tuning tasks and 2 code reasoning SFT tasks;
> >
> > **(4) demonstrate that it is not a domain mismatch problem** by using Domain Statistics;
> >
> > **(5) explain filtering potential** from information theory and model learning perspectives;
> >
> > **(6) compare thoroughly with current literature** demonstrating which doesn't contradict our findings.
> >
> > All results and discussions are updated in Appendix E,F, and Table 4,5,6,7,8,10.
> >
> > Thank you once again for the time and attention you've dedicated to reviewing our paper.
> >
> > We are eager to know whether we have addressed your concerns or if any further discussions are needed. Your insights are invaluable for refining our work.
> >
> > Best, Authors

---

> ### Comment · Reviewer_wNZi · 2024-11-30
> **Feedback of rebuttal -1**
>
> Thanks for your response, and I really appreciate the authors' additional experiments and clarifications. However, these don't address my original concerns about:
>
> - The claim in Line 83 contradicting previous work
> - Data leakage in Figure 2's evaluation and domain issues
> - The problematic and insufficiently explained non-autoregressive method for generating better synthetic data
> - The lack of statistical significance in the evaluation results (In the updated manuscripts, both Table 2 and Table 3 show very small improvements with no mean/variance reported)
>
> So I would keep my initial rejection position.
>
> 1. The claim in Line 83 is challenged by previous work WRAP[1], which wasn't discussed in the initial version. Moreover, this claim is now also challenged by this submission's new experiments:
>
>
> - The new results show that different synthetic datasets yield different conclusions about performance impact: Cosmopedia degrades model performance, while WRAP[1] and ToEDIT improve it. This challenges the submission's fundamental claim in line 83 that "we find that directly mixing general synthetic data, without iterative training, leads to performance degradation."
> - The definitions of semi-synthetic and pure-synthetic lack scientific rigor. In fact, Cosmopedia, WRAP, and TOEDIT synthesize data conditioned on different levels of real data:
>
>   - Cosmopedia expands real data and generates data without real data
>   - WRAP generates paraphrases of real data
>   - TOEDIT performs token editing on real data
> These three methods clearly modify real data at large, medium, and small scales respectively. The concept of pure synthetic data is ambiguous, as WRAP is also fully generated by an LLM - couldn't it be considered pure synthetic? Additionally, Figure 1 uses "complete synthetic data" - do you mean "pure synthetic data"?
>
>
> 2. The experiments in Figure 2 remain the primary issue warranting rejection:
>
> The authors overlooked that C4 is a subset of CommonCrawl. The table in Response-2 clearly shows that CommonCrawl is the largest portion of Dolma. Therefore, training with Dolma and evaluating with C4 is clearly an in-domain evaluation.
> C4 and Wikipedia are part of Dolma, which is the pretraining data. Testing with C4 and Wikipedia should be avoided. This isn't an in-domain versus out-of-domain issue - it's a test data leakage issue.
>
> As shown in WRAP[1], equations (1) and (2) demonstrate that it's unfair to compare the perplexity of models trained on synthetic data with those trained on real data, as the latter are optimized to minimize perplexity over real data. Thus, they should naturally achieve lower perplexity than models trained on a mixture of real and synthetic data.
>
> Consequently, I remain concerned about all results and conclusions from Figure 2, which I consider a significant issue in ML research methodology.
>
>
> 3. Regarding non-autoregressive token replacement:
>
>
> - While I agree this is computationally efficient,
> - The new results contradict the selection of p=0.99. According to this table, p=0.1 performs best.
> - It's unclear why replacing words that the model is highly confident about (which are likely common words) would be beneficial. What do these tokens get replaced with? A key concern is that pronouns and articles typically have high probability - why would replacing them make sense? Could you explain what tokens with p=0.99 probability would be replaced with? In my understanding, if one words get >0.99 probability, the rest choices are all lower than 0.01 which means they are improper to be here.
>
>
> 4. For the results in Response-4 and the updated draft, could you report means and variances? The improvements remain very small and inconsistent.
>
>
> Finally, I am sorry I late in responding as it has been took me some time to read all these new results and discussion. But luckily the discussion period has been extended and I will response quickly in the next several day.

---

> > ### Comment · Reviewer_wNZi · 2024-11-30
> > **Feedback of rebuttal - 2**
> >
> > if possible, can you show some examples of the synthetic data with different p? For example, without replacement, after replaced, and replaced with different p. I think this is very important to understand why the data quality gets improved but this is missed in both the initial draft and the revised draft. (correct me if they are there but I missed them. I am sorry if that's the case)

---

> > > ### Author Response · Authors · 2024-12-01
> > > **Follow-Up Response - 1**
> > >
> > > We deeply appreciate receiving your thorough and detailed feedback. We will do our utmost to address your concerns.
> > >
> > > > [Q1] The claim in Line 83 contradicting previous work.
> > >
> > > We acknowledge your insightful observation and understanding --`These three methods clearly modify real data at large, medium, and small scales respectively`.  To avoid ambiguity of concepts and correct contradiction, we will provide the scientific rigor explanation and definition of synthetic data. Accordingly, we will revise line 83 and all related statements in our paper.
> > >
> > > - line 83 $\rightarrow$ Specifically, we find that directly mixing "pure synthetic data$^1$", without iterative training, leads to performance degradation.
> > >   -  "pure synthetic data$^1$": This data is obtained by expanding a prompt using a language model. In contrast, semi-synthetic data (e.g., rephrasing in [1]) improves pretraining performance.
> > >
> > > `The definitions of semi-synthetic and pure-synthetic lack scientific rigor.`
> > >
> > > **Definition of Synthetic Data**
> > >
> > > Synthetic data ($D_s$) can be categorized based on its relationship with the distributions of a language model ($P_{LM}$) and real data ($P_{data}$) during the generation process, quantified as $d = \text{KL}( ·|| P_{\text{data}})$,
> > >
> > > $$
> > > D_s =
> > > \begin{cases}
> > > D_s^{pure} \sim P_{\text{LM}} , & \text{if }  \text{KL}(P_{\text{LM}} || P_{\text{data}}) > \epsilon, \\\\
> > > D_s^{semi}\sim P_{\text{semi}}, & \text{if }  \text{KL}(P_{\text{semi}} || P_{\text{data}}) \leq \epsilon, \\\\
> > > \end{cases}
> > > $$
> > >
> > > where:
> > >
> > > -  **Pure Synthetic Data $D_s^{pure}$**: Generated entirely from the language model $(D_s^{pure} \sim P_{\text{LM}}),$ with a KL divergence $\text{KL}(P_{\text{LM}} \| P_{\text{data}})$ exceeding a threshold $\epsilon$. This implies a significant deviation of the language model’s distribution from the real data distribution.
> > >
> > > - **Semi-Synthetic Data $D_s^{semi}$**. Derived from limited modifications to real data ($P_{\text{data}}$), ensuring that the resulting distribution ($P_{\text{semi}}$) has a KL divergence $\text{KL}(P_{\text{semi}} || P_{\text{data}})$ bounded by $\epsilon$. This reflects a closer alignment of semi-synthetic data with real data.
> > >
> > > From the generation process:
> > >
> > > - **Pure Synthetic Data $D_s^{pure}$** : This data is induced by a language model through prompts and does not modify real data, resulting in low overlap content with real data.
> > >   - Example: `Cosmopedia expands real data and generates data without real data.`
> > > - **Semi-Synthetic Data $D_s^{semi}$**:  This data is generated by directly modifying real data, such as paraphrasing or token-level editing. It derives from transformations of real data.
> > >   - Example:  `WRAP[1] generates paraphrases of real data. TOEDIT (ours) performs token editing on real data.`
> > >
> > > `Figure 1 uses "complete synthetic data" - do you mean "pure synthetic data"?`
> > >
> > > Your understanding is correct. We will revise all related statements in our paper to align with the clearer definition provided above.

---

> > > > ### Author Response · Authors · 2024-12-01
> > > > **Follow-Up Response - 2**
> > > >
> > > > > [Q2] : Data leakage in Figure 2's evaluation and domain issues
> > > >
> > > > Our evaluation sets are more than C4 and Wikipedia.
> > > >
> > > > **We try to mitigate the potential impact of data leakage by conducting an extensive evaluation of a total of 34 test sets in our paper, which enables us to obtain more accurate results.**
> > > >
> > > > The test sets and results are listed below:
> > > >
> > > > - Figure 2 and 7 (PPL test sets): wikitext, RedPajama, Falcon-RefineWed, c4-en and mc4-en.
> > > > - Table 5 (PPL test sets): ArXiv, BookCorpus2, Books3, DM_Mathematics, Enron_Emails, EuroParl, FreeLaw, GitHub, Gutenberg_(PG-19), HackerNews, NIH_ExPorter, OpenSubtitles, OpenWebText2, PhilPapers, Pile-CC, PubMed_Abstracts, PubMed_Central, StackExchange, Ubuntu_IRC, USPTO_Backgrounds, Wikipedia_(en), YoutubeSubtitles.
> > > > - Table 6 (downstream tasks): TruthfulQA, LogiQA, Wino, PIQA, ARC-E, BoolQ and OBQA
> > > >
> > > > Discussion on the potential impact of data leakage in our test sets:
> > > >
> > > > - Results in Figure 2: We use standard test sets from [Paloma](https://huggingface.co/datasets/allenai/paloma), the official benchmark for Dolma as described in [15]. Paloma-bench is specifically designed for PPL evaluation, particularly for Dolma. According to the official statement from Dolma paper[15], the Dolma team has already performed data de-contamination to prevent data leakage.
> > > > - Results in Table 5,6: We followed your suggestion to conduct additional evaluations on other PPL test sets from the Pile and general understanding downstream tasks, consistent with those used in [1]. While data leakage remains a possibility, the addition of 22 extra test sets and downstream tasks may help mitigate this risk and enable us to obtain more accurate results.
> > > >
> > > > In summary, our approach is to expand the scope of test sets to minimize the potential risks of data leakage as effectively as possible.
> > > >
> > > > If you are still concerned about data leakage, please feel free to discuss further deeply with us.
> > > >
> > > > -----
> > > >
> > > > **The rest of the responses (case studies, mean/variance) are on the way. We kindly ask for a few more days.**

---

> > > > > ### Comment · Reviewer_wNZi · 2024-12-01
> > > > > **Feedback of rebuttal - 3**
> > > > >
> > > > > Thank you for your partial response. I'll address what I've read so far and will respond to your future posts once I receive them.
> > > > >
> > > > > While "Follow-Up Response - 2" addresses my concerns about data leakage, it doesn't resolve the issues with Figure 2. Additionally, I would suggest to cite Paloma.
> > > > >
> > > > > To clarify my concern about the domain issue: I disagree with the current revised version's experiments and statements in Figure 2. Specifically, I don't agree that Figure 2's experiments support the claim that "Non-Iterative Model Collapse occurs when training language models from scratch on AI-synthesized data or a mixture of human and synthetic data leads to performance degradation." This degradation could be attributed to domain mismatch rather than the synthetic nature of the data. All four subfigures in Figure 2 still present domain mismatch scenarios:
> > > > >
> > > > > - In-domain testing: pretraining with real data which consists of CC and Wikipedia, testing with data from CC and Wikipedia
> > > > > - Out-of-domain testing: pretraining with synthetic data or mixed synthetic data, testing with data from CC and Wikipedia
> > > > >
> > > > > Therefore, it's expected that models trained on more real data would achieve lower perplexity on these test sets.
> > > > >
> > > > > While I appreciate the authors' addition of downstream evaluations as I suggested to support Figure 2's statement, there are two main issues:
> > > > >
> > > > > - The results aren't convincing. Models trained on 25% synthetic data achieve 40.71 averaged scores, while those trained on 50% synthetic data perform similarly to real data (41.62 vs. 41.69). Moreover, training with 50% or 75% synthetic data outperforms training with 25% synthetic data. This trend clearly contradicts what's shown in Figure 2.
> > > > > - Given the small differences in downstream evaluation, mean and variance should be reported.
> > > > > - Although new experiments were added, the original experiments in Figure 2 should be corrected and replaced rather than retained. As mentioned above, readers might struggle to determine whether the degradation stems from using synthetic data or is simply due to expected domain mismatch evaluation.
> > > > >
> > > > > One additional comment: despite including many additional perplexity evaluation datasets (now 34 in total), the authors should have selected them more carefully. Some datasets, such as Gutenberg (PG-19), StackEx, pile-CC, and Arxiv, share the same domain as Dolma. Given Dolma's size, to ensure the comparison focuses on synthetic versus real data rather than in-domain versus out-of-domain effects, I suggest emphasizing downstream evaluation. Alternatively, using only a subset of Dolma (like C4) as the real data and evaluating on the PILE could work. Either solution could replace the existing Figure 2.

---

> > > > > > ### Author Response · Authors · 2024-12-02
> > > > > > **Follow-Up Response - 3**
> > > > > >
> > > > > > We sincerely appreciate your timely feedback and clarification. We further address your concerns as below.
> > > > > >
> > > > > > > [Q3] While "Follow-Up Response - 2" addresses my concerns about data leakage, it doesn't resolve the issues with Figure 2.
> > > > > >
> > > > > > `Given Dolma's size, to ensure the comparison focuses on synthetic versus real data rather than in-domain versus out-of-domain effects, I suggest emphasizing downstream evaluation.`
> > > > > >
> > > > > > **We will follow your suggestion to (1) replace Figure 2 with downstream task evaluation and (2) add more results to support our claim.**
> > > > > >
> > > > > > **We further expand downstream tasks evaluation on the OLMo models. The results below also validate and support our claim.**  The average performance of the OLMo models gradually decreases as incorporating pure synthetic data (rigorous concept definition refers to ["Follow-Up Response - 1"](https://openreview.net/forum?id=mVCcWCjeEz¬eId=0lSaZQQzPr) ). As you have noted, while there are some fluctuations in GPT-2 results in Table 5, the overall trend still indicates that incorporating purely synthetic data leads to performance degradation. The experiments on OLMo below further support our original claim.
> > > > > >
> > > > > > 1. **Below are the new supplementary results of pretraining OLMo-237M.  ± indicates the standard error.** (detailed settings are provided in Appendix D).
> > > > > >
> > > > > > |                    | TruthfulQA (%) | LogiQA (%)    | Wino. (%)     | PIQA (%)      | ARC-E (%)      | OBQA (%)      | Avg. (%) |
> > > > > > | ------------------ | -------------- | ------------- | ------------- | ------------- | -------------- | ------------- | -------- |
> > > > > > | Human Data         | 26.81 ± 1.550  | 21.06 ± 1.028 | 52.01 ± 1.404 | 56.69 ± 1.156 | 31.73 ± 0.9550 | 13.80 ± 1.543 | 33.68    |
> > > > > > | 25% Synthetic Data | 26.44 ± 1.543  | 21.25 ± 1.032 | 52.64 ± 1.403 | 57.02 ± 1.155 | 31.78 ± 0.9552 | 12.40 ± 1.475 | 33.59    |
> > > > > > | 50% Synthetic Data | 25.95 ± 1.534  | 20.04 ± 1.099 | 52.25 ± 1.408 | 56.64 ± 1.126 | 31.82 ± 0.9557 | 12.80 ± 1.495 | 33.25    |
> > > > > > | 75% Synthetic Data | 25.34 ± 1.522  | 20.87 ± 1.025 | 50.43 ± 1.405 | 55.60 ± 1.159 | 32.74 ± 0.9629 | 12.00 ± 1.454 | 32.83    |
> > > > > > | Synthetic Data     | 23.01 ± 1.473  | 20.29 ± 1.014 | 49.33 ± 1.405 | 55.93 ± 1.158 | 33.33 ± 0.9673 | 14.20 ± 1.562 | 32.68    |
> > > > > >
> > > > > > `Additionally, I would suggest to cite Paloma.`
> > > > > >
> > > > > > We will add the citation and incorporate all discussion of data source in Appendix.
> > > > > >
> > > > > > **More updated table with standard error as follows (more updated table will be presented in revision):**
> > > > > >
> > > > > > 2. General Instruction tuning tasks
> > > > > >
> > > > > > |                      | PIQA(%)        | BoolQ(%)       | HellaSwag(%)   | SIQA(%)        | Winogrande(%) | Avg(%)    |
> > > > > > | -------------------- | -------------- | -------------- | -------------- | -------------- | ------------- | --------- |
> > > > > > | CoT [3]              | 79.87 ± 0.9355 | 81.28 ± 0.6821 | 59.72 ± 0.4894 | 49.69  ± 1.131 | 74.51 ± 1.224 | 69.014    |
> > > > > > | $\Delta ToEdit$      | 80.25 ± 0.9140 | 81.16 ± 0.6838 | 59.74 ± 0.4832 | 50.56  ± 1.127 | 74.59 ± 1.223 | **69.26** |
> > > > > > |                      |                |                |                |                |               |           |
> > > > > > | FLAN V2 [4]          | 80.79 ± 0.9190 | 84.04 ± 0.6406 | 59.98 ± 0.4589 | 51.43 ± 1.130  | 74.66 ± 1.222 | 70.18     |
> > > > > > | $\Delta ToEdit$      | 80.69 ± 0.9210 | 85.20 ± 0.6210 | 59.99 ± 0.4869 | 52.00 ± 1.020  | 75.37 ± 1.210 | **70.65** |
> > > > > > |                      |                |                |                |                |               |           |
> > > > > > | Open Assistant 1 [5] | 79.65 ± 0.939  | 83.18 ± 0.654  | 60.51 ± 0.487  | 48.52 ± 1.130  | 74.11 ± 1.23  | 69.194    |
> > > > > > | $\Delta ToEdit$      | 79.98 ± 0.933  | 83.91 ± 0.642  | 60.34 ± 0.488  | 48.31 ± 1.121  | 74.66 ± 1.22  | **69.44** |
> > > > > >
> > > > > > 3. Code SFT tasks;
> > > > > >
> > > > > > |                    | ARC-Challenge(%) | GPQA(%)       | GSM8K(%)      | MMLU(%)        | Avg(%)      |
> > > > > > | ------------------ | ---------------- | ------------- | ------------- | -------------- | ----------- |
> > > > > > | OSS-Instruct-75K   | 51.28 ± 1.453    | 27.46 ± 2.110 | 49.58 ± 1.257 | 62.14 ± 0.3818 | 45.76       |
> > > > > > | $\Delta ToEdit$    | 51.79 ±  1.460   | 28.79 ± 2.141 | 49.36 ± 1.377 | 62.04 ± 0.3823 | **46.1328** |
> > > > > > |                    |                  |               |               |                |             |
> > > > > > | Evol-Instruct-110K | 52.9 ±  1.458    | 27.9 ± 2.121  | 50.87 ± 1.322 | 62.4 ± 0.3828  | 46.616      |
> > > > > > | $\Delta ToEdit$    | 52.22 ± 1.459    | 29.69 ± 2.160 | 50.87 ± 1.347 | 62.6 ± 0.3827  | **46.9216** |

---

> > > > > > > ### Author Response · Authors · 2024-12-02
> > > > > > > **Follow-Up Response - 4**
> > > > > > >
> > > > > > > > [Q4] It's unclear why replacing words that the model is highly confident about (which are likely common words) would be beneficial.
> > > > > > >
> > > > > > > **Intuitively, we replace relatively simpler tokens (high probability) with harder ones (low probability), thereby increasing the learning difficulty of the sample.**
> > > > > > >
> > > > > > > **Case studies**:
> > > > > > >
> > > > > > > The 3  tables below show the cases from CoT, Flan V2, and Magicoder-Evol-Instruct-110K datasets. In these cases, our method tends to replace common words with challenging ones. For example, "people" → "individuals", this change enhances the lexical complexity and the semantic depth. The same example includes: "count" → "calculate",  "mean" → "imply" and so on. This aligns with our goal: adjusting data distribution towards a more informative direction.
> > > > > > >
> > > > > > > **Theoretical explanation：**
> > > > > > >
> > > > > > > - **Motivation:** Inspired by Information theory (detailed discussion in [Reponse 5](https://openreview.net/forum?id=mVCcWCjeEz&noteId=7RuuW5jtav) and Appendix F.4), the uniform distribution possesses the maximum information entropy. We try to adjust the U-shaped distribution (Figure 6) toward a uniform distribution. Theoretically, by re-sampling tokens in the high probability region, we can maximize the information entropy of the datasets.
> > > > > > >
> > > > > > > - **Implementation:** In practice, we also need to consider whether the sampled tokens are appropriate. As in real-world scenarios, the language model does not fit perfectly and can only serve as a proxy distribution. Therefore, we conduct ablation experiments to select the appropriate p.
> > > > > > >
> > > > > > > From the perspective of distribution shift, we present the progressive percentage changes below (Table 11), indicating distribution shifting. More discussion can be found in Appendix F.
> > > > > > >
> > > > > > > `Why do you choose p=0.99 instead of p=0.01?` In other words, why do we re-sample tokens from the high-probability region?
> > > > > > >
> > > > > > > **To prevent the loss of long-tail information, we opt to re-sample within the high-probability region.** In the data synthesizing process, long-tail samples are often lost caused by the truncation inherent in LM sampling [16]. The continual loss of long-tail information during data synthesis leads to model collapse [6]. More discussion about how we choose the hyper-parameter p can refer to Appendix E.
> > > > > > >
> > > > > > > |                   | Generation 1 (source) | Generation 2 | Generation 3 |
> > > > > > > | ----------------- | --------------------- | ------------ | ------------ |
> > > > > > > | Tokens ($p>0.99$) | 584,103               | 549,519      | 517,433      |
> > > > > > > | Percentage        | 12.5%                 | 11.76%       | 11.08%       |
> > > > > > >
> > > > > > > Case 1: code reasoning sample in Magicoder-Evol-Instruct-110K:
> > > > > > >
> > > > > > > | **Before** (source)                                          | **After** (edited)                                           |                        |
> > > > > > > | ------------------------------------------------------------ | ------------------------------------------------------------ | ---------------------- |
> > > > > > > | Construct a function using PHP language that applies lexical analysis on a provided text string to analyze the individual, non-repeated words elements present. | Construct a function using PHP language that applies lexical analysis on a provided text string to quantify unique words. | "analyze" → "quantify" |
> > > > > > > | Test with provided string, `$str = 'Greetings, Planet Earth!'`. | Test with provided string, `$str = 'Greetings, Planet Earth!'`. | No changes.            |
> > > > > > > | Implements `wordCount` to remove punctuation, convert text to lowercase, split into words, and count unique words. | Implements `wordCount` to remove punctuation, convert text to lowercase, split into words, and calculate unique words. | "count" → "calculate"  |
> > > > > > > | Returns `{'greetings': 1, 'planet': 1, 'earth': 1}`.         | Returns `{'greetings': 1, 'planet': 1, 'earth': 1}`.         | No changes.            |

---

> > > > > > > > ### Author Response · Authors · 2024-12-02
> > > > > > > > **Follow-Up Response - 5**
> > > > > > > >
> > > > > > > > Case 2: instruction tuning sample in Flan V2：
> > > > > > > >
> > > > > > > > | **Before** (source)                                          | **After** (edited)                                           |                         |
> > > > > > > > | ------------------------------------------------------------ | ------------------------------------------------------------ | ----------------------- |
> > > > > > > > | Here is a premise: Washington Post reports two attempts by private companies to grow embryos--a practice banned among federally funded researchers but allowed in the private sector. | Here is a premise: Washington Post documents two attempts by private companies to grow embryos--a practice banned among federally funded researchers but allowed in the private sector. | "reports" → "documents" |
> > > > > > > > | Here is a hypothesis: Washington Post has never made a report on the subject of embryos. | Here is a hypothesis: Washington Post has never published a report on the subject of embryos. | "made" → "published"    |
> > > > > > > > | Here are the options: Pick your answer from: - yes. - it is not possible to tell. - no. | Here are the options: Pick your answer from: - yes. - it is not possible to tell. - no. | No changes.             |
> > > > > > > > | Is it possible to conclude that if the premise is true, then so is the hypothesis? | Is it possible to conclude that if the premise is true, then so is the hypothesis? | No changes.             |
> > > > > > > > | "no"                                                         | "no"                                                         | No changes.             |
> > > > > > > >
> > > > > > > > Case 3: instruction tuning sample in CoT：
> > > > > > > >
> > > > > > > > | **Before** (source)                                          | **After** (edited)                                           |  |
> > > > > > > > | ------------------------------------------------------------ | ------------------------------------------------------------ | ------------------------ |
> > > > > > > > | Can we conclude from "A man in a green tie talking to two woman over drinks." that "The people are at a bar."? | Can we conclude from "A man in a green tie talking to two woman over drinks." that "The individuals are at a bar."? | "people" → "individuals" |
> > > > > > > > | Options:                                                     | Options:                                                     | No changes.              |
> > > > > > > > | - yes                                                        | - yes                                                        | No changes.              |
> > > > > > > > | - no                                                         | - no                                                         | No changes.              |
> > > > > > > > | - it is not possible to tell.                                | - it is not possible to tell.                                | No changes.              |
> > > > > > > > | Explanation: A man in a green tie talking to two woman over drinks does not mean they are at a bar. | Explanation: A man in a green tie talking to two woman over drinks does not imply they are at a bar. | "mean" → "imply"         |
> > > > > > > >
> > > > > > > > [16] Dohmatob E, Feng Y, Yang P, et al. A tale of tails: Model collapse as a change of scaling laws[J]. arXiv preprint arXiv:2402.07043, 2024.

---

> ### Comment · Reviewer_wNZi · 2024-12-02
> **Feedback of rebuttal - 4**
>
> Thanks for your examples.
> 1. Can you answer my question about pronoun and articles?
> 2. Can you answer my question about what's the exact resampling strategy TOEDIT is using? Also, I think this is an important detail but why it is missing from the submission?
> 3. Can you answer my question about how p=0.99 was selected? According to the ablation results during rebuttal, p=0.99 is not the best. You proposed a hypothesis  about keeping long-tail property, but it is contradicted to the experiment results.
> 4. The examples are all whole word replacement, but in practice, the replacement would happen for subwords. For example, given a word A, if it is split into subwords B and C. C's probability would be very high when conditioned on B but it is weird to replace C but keep B unchanged. how does TOEDIT handle this case?
> 5. Can you also provide a sequence of subwords and the probabilities of each subword for any of the examples you posted here?

---

> ### Author Response · Authors · 2024-12-03
> **Follow-Up Response - 6**
>
> We would like to kindly bring your attention to the responses posted:
>
> - **We have presented the resampling strategy at line 424 in Sec. 4.** Discussions and ablation experiments of the resampling strategy are presented in Appendix E and Table 7,8. We also discussed with other reviewers in [Reponse -a](https://openreview.net/forum?id=mVCcWCjeEz&noteId=mKtNwfCyrM) and [Reponse - b](https://openreview.net/forum?id=mVCcWCjeEz&noteId=d9aF52MEjw) . The same reponse were also presented to you in [Reponse-3.2](https://openreview.net/forum?id=mVCcWCjeEz&noteId=QcJKIpEReb) .
> - **We have discussed why we choose $p=0.99$  empirically and theoretically in [follow-up reponse 4](https://openreview.net/forum?id=mVCcWCjeEz&noteId=mdpeUWoZnt) and Appendix E.**
> - " A hypothesis about keeping long-tail property" is not new. **We have mentioned this motivation repeatedly in the Introduction (line 052),  Sec 2.2,  Sec 3.1 Method, Appendix F.7, and Figure 3.**
>
> ---
>
> We truly acknowledge and respect your comments. Our paper focuses on "How to synthesize text data to avoid model collapse ?" The model collapse problem is our primary concentration. Around this topic:
>
> - We have demonstrated that our method (1) theoretically avoids model collapse, with a fixed upper bound proved in Section 3 and Appendix B, and (2) empirically enhances performance in pre-training, continual pre-training, and SFT, as shown in Tables 1, 2, and 3.
> - We also present ablation experiments, including different $p$ values, sampling strategies, and percentage statistics (Tables 4, 7, 8, and 9), along with case studies ([Follow-up Response-4](https://openreview.net/forum?id=mVCcWCjeEz¬eId=mdpeUWoZnt)).
> - Further detailed discussions in the Appendix includes the differences between iterative and non-iterative model collapse, filtering potential, coverage collapse, the DSIR mechanism, potential applications, and a rigorous definition of synthetic data ([Follow-up Response-1](https://openreview.net/forum?id=mVCcWCjeEz&noteId=0lSaZQQzPr)).
>
> As for the part of speech (POS) of token within samples (e.g., pronouns or articles), we treat all words equally and do not implement specific designs for different POS. This does not affect the effectiveness of our method. Below, we present our source code, where you can directly refer to the implementation details. Since the extended discussion phase nears its close, we will update the analysis for probabilities of each subword and a supplementary discussion for POS in Appendix.
>
> We will include all additional results in the revision.
>
> Thank you once again for your invaluable suggestion.
>
> ```
> def resampling(prob_dict, num_samples=1, beta=1):
>         words_candicate = list(prob_dict.keys())
>         prob_scores = np.array(list(prob_dict.values()))
>
>         adjusted_probs = softmax(prob_scores / beta)
>         accepted_token = np.random.choice(words_candicate, num_samples, p=adjusted_probs)
>
>         return accepted_token
> ```

---

### Author Response · Authors · 2024-11-25
**Summary of Rebuttal**

Dear Reviewers,

Thank you once again for your valuable comments and suggestions on our paper. We have carefully revised our manuscript and conducted additional experiments based on your feedback. For the reviewers’ convenience, **we have highlighted the changes in the revised manuscript in blue.** We present a summary of the modifications as follows:

1. **More validation of findings (@Reviewer wNZi and 4XBZ):**  We have supplemented below experiments, which aligns with our original findings. Additionally, we provide a detailed comparison with other methods in Table 10 and Appendix F.7.
   - 22 PPL validation sets and 7 downstream tasks on GPT-2 and OLMo models in Table 5 and 6.
2. **Ablation Study on Hyper-Parameter $p$ (@Reviewer wNZi, 4XBZ, PCAs, and 1Bd4)**: We have supplemented our analysis with (1) various values of $p$, (2) percentage statistics, (3) sampling strategies, and (4) sampling sizes, as detailed in Tables 4, 7, 8, 9, and Appendix E.
3. **Additional Empirical Results (@Reviewer wNZi, 1Bd4, 4XBZ, and PCAs):** We include 3 general instruction tuning tasks and 2 code reasoning SFT tasks in Table 3 to further validate our method.
4. **Domain Statistics for Dolma (@Reviewer wNZi):** We present the domain statistics for Dolma in Table 15, demonstrating that it is not a domain mismatch problem.
5. **Evidence of Progressive Decrease in Editing (@Reviewer 4XBZ and 1Bd4):** We have supplemented the evidence of the progressive decrease in editing, as presented in Table 11 and Appendix F.6.
6. **Explanation of U-Shape Distribution Filtering Potential (@Reviewer wNZi and 4XBZ):** We analyze the filtering potential of the U-shape distribution through the perspectives of information theory and model learning, as detailed in Appendix F.4.
7. **Explanation of Experimental Setup and DSIR (@Reviewer 4XBZ and PCAs):** We provide more detailed descriptions of the experimental setups in Appendix D and how DSIR works in Appendix F.3.
8. **Definitions of Non-Iterative Model Collapse and Coverage Collapse (@Reviewer PCAs and 1Bd4):** We provide comprehensive definitions of non-iterative model collapse and coverage collapse in Appendices F.1 and F.2, respectively.
9. **Improvements of Figure 1 (@Reviewer 1Bd4)**:  We have annotated all the processes and symbols at the bottom of Figure 1 to facilitate quick reference for readers.
10. **Impact on Future Work (@Reviewer 1Bd4):** We discuss the potential applications and influence on future research in Appendix G. And, we explain that our method needn't assume the initial data is 100% human-authored in Appendix F.8.

Finally, we sincerely thank all reviewers and ACs for their critical feedback and valuable suggestions.

---

### Meta-Review · Area_Chair_rfu7 · 2024-12-21

**Metareview:**

The paper proposes to mitigate model collapse by creating "semi-synthetic" data. ToEdit resamples tokens in human data based on probabilities from a pre-trained model, with a threshold parameter controlling the extent of modification. The paper provides a multi-faceted investigation, including statistical analysis of synthetic data, a theoretical framework for ToEdit, and empirical evaluation.

Reviewer wNZi highlighted a critical flaw in the experiments presented in Figure 2, arguing that the observed performance degradation could be attributed to domain mismatch rather than the inherent properties of synthetic data. The authors addressed this by adding more downstream evaluations. The rationale behind the final parameter selection could be more thoroughly explained in the revised version. Reviewer 1Bd4 also expressed concerns about the practical implications of the paper's findings, arguing that the assumption of having 100% human-authored data to start with limits the applicability of ToEdit in real-world scenarios where data is often a mix of human and synthetic content. A more detailed explanation of the DSIR sampling method and its role in the data analysis can be added.

While the paper focuses on a timely problem, the reviewers argue that the paper has fundamental flaws, especially Reviewer wNZi with high confidence. Considering the weaknesses detract from the paper's overall quality and impact, the submission can be improved by overcoming the critical concerns raised by reviewers.

**Additional Comments On Reviewer Discussion:**

The assumption of a progressive decrease in editing operations with a fixed ratio in the theoretical model lacked strong empirical validation in the original submission. The authors added some supporting evidence in the rebuttal, but further analysis might be beneficial. The initial distinction between terms lacked scientific rigor, as pointed out by Reviewer wNZi. Although addressed in the rebuttal with a more formal definition based on KL divergence, it required careful consideration in the revised manuscript. Reviewer wNZi also highlighted a critical flaw in the experiments presented in Figure 2, arguing that the observed performance degradation could be attributed to domain mismatch rather than the inherent properties of synthetic data. The authors addressed this by adding more downstream evaluations.

---

### Decision · Program_Chairs · 2025-01-22

Reject